EMBO
Molecular Medicine

# NOTCH1 intracellular domain stabilization by MDM2 plays a major role in NSCLC response to platinum

Sara Bernardo[1], Lisa Brunet[1,14], Quentin Dominique Thomas [1,14], David Bracquemond[1], Céline Bouclier[1], Marie Colomb[1], Maicol Mancini[1], Eric Fabbrizio [1,2], Alba Santos[2,3,4], Sylvia-Fenosoa Rasamizafy [1], Amina-Milissa Maacha[1], Anais Giry[1], Emilie Bousquet-Mur[1], Laura Papon[1], Marion Goussard[1], Christophe Fremin[1], Andrea Pasquier [5], María Rodríguez [6,7], Camille Travert[1,8], Jean-Louis Pujol[1,8], Laetitia K Linares[9], Lisa Heron-Milhavet [10], Alexandre Djiane [10], Irene Ferrer[2,3,4], Luis Paz-Ares [3,4,11], Xavier Quantin[1], Luis M Montuenga [4,5,12,13], Hélène Tourriere [1,2✉] & Antonio Maraver [1✉]

## Abstract

**Despite major advances in the clinical management of non-small cell lung carcinoma (NSCLC), most patients treated with first-line platinum-based chemotherapy combined with immune checkpoint inhibitors will relapse, which constitutes an unmet medical need. Here, we found that various DNA damage inducers increase the levels of Notch Intracellular Domain (NICD), the active form of NOTCH1. Mechanistically, we revealed that, upon platinum treatment, the expression levels of both MDM2 and NICD were increased and that MDM2 stabilised NICD through ubiquitination. Using NSCLC patient-derived xenografts displaying intrinsic carboplatin resistance, we demonstrated that combining carboplatin with a γ-secretase inhibitor, which hinders NICD generation, significantly improves survival and reduces tumour growth compared with carboplatin monotherapy. Furthermore, in patients with NSCLC who received platinum-based chemotherapy, the level of MDM2 expression in the tumour correlated with poor progression-free survival, which further validates the key role of MDM2 in response to platinum compounds. Our findings present a new therapeutic opportunity for patients with NSCLC, the most common form of lung cancer.**

**Keywords** NSCLC; Platinum Chemotherapy; Resistance; MDM2; Notch
**Subject Categories** Cancer; Post-translational Modifications & Proteolysis; Respiratory System

## Introduction

Lung cancer is the most lethal cancer in the world with 1.8 million deaths per year (Organization, 2021). The main subtype is non-small cell lung carcinoma (NSCLC) that accounts for around 85% of all lung cancer cases. The three major genomic alterations in NSCLC are oncogenic mutations in the *KRAS* and *EGFR* genes, and *ALK* fusions (Skoulidis and Heymach, 2019). The survival of patients who are eligible for targeted therapy, for instance patients with NSCLC harboring EGFR or ALK genetic alterations, has been greatly improved, but most patients will develop resistance at later stages (Girard, 2022; Lin et al, 2021). Of note, conventional platinum-based chemotherapy is the main alternative for the patients who develop resistance to targeted therapy (Hendriks et al, 2023b). Moreover, the combination of platinum-based therapy and immune checkpoint inhibitors is proposed as first-line therapy to patients with the main oncogenic alteration in NSCLC, KRAS (Reck et al, 2021), and also to most patients who are not eligible for targeted therapies. Therefore, platinum chemotherapy, alone or in combination, is still the standard of care for ~60-70% of patients with metastatic NSCLC (Gandhi et al, 2018; Paz-Ares et al, 2018). However, some patients will develop resistance and other patients will never respond due to intrinsic resistance. Hence, understanding the basis of the response/absence of response to platinum is a major clinical question that needs to be addressed.

Several resistance mechanisms to platinum salts have been proposed, such as decreased cellular drug import, increased cellular detoxification, or enhanced DNA damage repair, to name only a few (Huang et al, 2021). Regarding enhanced DNA damage repair, platinum salts form inter- and intra-strand adducts in the DNA

[1]Oncogenic Pathways in Lung Cancer. Institut de Recherche en Cancérologie de Montpellier (IRCM), Univ Montpellier, Institut Régional du Cancer de Montpellier (ICM), INSERM, Montpellier, Cedex 5 34298, France. [2]CNRS, Paris, France. [3]Unidad de Investigación Clínica de Cáncer de Pulmón, Instituto de Investigación Hospital 12 de Octubre-CNIO, Madrid 28029-28041, Spain. [4]Consorcio de Investigación Biomédica en Red de Cáncer (CIBERONC), Madrid 28029, Spain. [5]Solid Tumors Program, CIMA-CCUN-University of Navarra, Pamplona, Spain. [6]Department of Thoracic Surgery, Clínica Universidad de Navarra, Pamplona, Spain. [7]CCUN-University of Navarra, Pamplona, Spain. [8]Montpellier Academic Hospital, Hôpital Arnaud de Villeneuve, Montpellier, Cedex 5 34090, France. [9]Metabolism and Sarcomas. Institut de Recherche en Cancérologie de Montpellier (IRCM), Univ Montpellier, Institut Régional du Cancer de Montpellier (ICM), INSERM, Montpellier, Cedex 5 34298, France. [10]Epithelial Growth and Cancer. Institut de Recherche en Cancérologie de Montpellier (IRCM), Univ Montpellier, Institut Régional du Cancer de Montpellier (ICM), INSERM, Montpellier, Cedex 5 34298, France. [11]Medical School, Universidad Complutense, Madrid 28040, Spain. [12]Department of Pathology, Anatomy and Physiology, School of Medicine, University of Navarra, Pamplona, Spain. [13]Navarra Health Research Institute (IDISNA), Pamplona, Spain. [14]These authors contributed equally: Lisa Brunet, Quentin Dominique Thomas. ✉E-mail: helene.tourriere@inserm.fr; antonio.maraver@inserm.fr

double helix that eventually lead to double-strand breaks (DSBs), if not properly repaired. DNA DSBs are the most deleterious lesions because they generate genomic instability and trigger the DNA Damage Response (DDR) (Aguilera and Gomez-Gonzalez, 2008). The main DDR sensors are the ataxia telangiectasia and Rad3 related (ATR) and ataxia-telangiectasia mutated (ATM) checkpoint protein kinases that initiate a signaling cascade to activate cellular programs with the aim of maintaining genome integrity. Depending on the DNA damage extent, the DDR might promote cell cycle arrest to allow DNA repair or might induce cell apoptosis and/or senescence if DNA damage is extreme (Krenning et al, 2019).

Activation of the Notch signaling pathway also has been implicated in platinum treatment resistance in different cancers, including lung cancer where this treatment induces Notch activity through a largely unknown mechanism (Liu et al, 2013). NOTCH is a transmembrane receptor activated by the interaction with transmembrane ligands present on the membrane of neighboring cells. There are four different Notch receptors (NOTCH1-4) and five ligands [JAGGED 1 and 2, and Delta-like (DLL) 1, 3 and 4] in humans. Ligand binding to NOTCH extracellular domain induces a multistep proteolytic cleavage of the receptor. The last step is promoted by the γ-secretase complex, leading to the release of the active form of the receptor called NOTCH Intracellular Domain (NICD). Therefore, γ-secretase inhibitors, which abolish NICD formation, represent a promising strategy to defeat cancer (McCaw et al, 2021; Ran et al, 2017). After nuclear translocation, in the canonical Notch pathway, NICD binds to the transcription factor RBPJ, allowing the displacement of corepressors and the recruitment of coactivators, such as MAML and p300. These factors activate a transcription program that includes among other, HES/HEY family members, Deltex-1 or MYC, depending on the cell type and context (Bray, 2016). We previously demonstrated that the Notch pathway plays a major role in KRAS-driven and in EGFR-driven NSCLC biology as well as in resistance to EGFR-targeted therapy (Bousquet Mur et al, 2020; Maraver et al, 2012). As Notch activity is increased in lung cancer cells upon platinum treatment (Liu et al, 2013), we hypothesized that Notch might play a major role also in the resistance to such treatment.

Here, we confirmed that the Notch pathway is activated upon platinum-based therapy in vitro and also expanded to the in vivo setting. Moreover, we found that the DDR promoted such activation. Specifically, we found that upon Carboplatin treatment, ATM increased MDM2 levels that in turn enhanced NICD stability through non-degradative ubiquitination. We also demonstrated in vivo, using a NSCLC patient-derived xenograft (PDX) displaying intrinsic resistance to Carboplatin, that inhibition of MDM2 or of the Notch pathway increases survival upon Carboplatin treatment. Lastly, we observed that in patients with NSCLC treated with platinum-based chemotherapy, MDM2 levels negatively correlated with progression-free survival (PFS). Altogether, our findings describe a DNA damage-ATM-MDM2-NICD axis induced by Carboplatin treatment and open a new therapeutic opportunity for patients with NSCLC.

# Results

## DNA damage induces NICD stabilization

The Notch pathway is upregulated in response to platinum treatment in lung cancer cells (Liu et al, 2013), but the underlying molecular mechanisms are largely unknown. Therefore, first we tested whether the Notch pathway was activated only by platinum (Carboplatin) or also by other DNA damage inducers currently used for cancer treatment (in particular Irinotecan, a topoisomerase II inhibitor, and γ irradiation). As the tumor suppressor p53 plays a major role in the DDR (Abuetabh et al, 2022), we used two human KRAS mutant-driven NSCLC cell lines: A549 (harboring wild-type p53) and H358 (harboring a homozygous deletion of p53). We exposed A549 and H358 cells to the three genotoxic agents and monitored DDR and Notch pathway activation by measuring the expression levels of phosphorylated (active) ATM and NICD, respectively. All three treatments led to DDR and Notch pathway activation (Figs. 1A and EV1A). This effect was p53-independent because NICD level was similarly increased in both H358 and A549 cells (Figs. 1A and EV1A).

We then focused on Carboplatin, the molecule clinically relevant for NSCLC treatment. We first confirmed that DNA DSBs were produced upon incubation with Carboplatin, as indicated by the formation of γH2AX and 53BP1 foci (Figs. 1B and EV1B). Following γ−secretase-mediated cleavage of the NOTCH transmembrane domain, NICD is translocated to the nucleus to exert its transcriptional program (Bray, 2016). Therefore, to confirm Notch pathway activation upon Carboplatin treatment in our cellular models, we used subcellular fractionation and analyzed NICD levels in the different subcellular compartments. NICD strongly accumulated in the chromatin fraction of platinum-treated H358 and A549 cells (thus p53-independent effect), validating the increased Notch activity upon DNA damage in our system (Figs. 1C and EV1C). Taken together, our data demonstrate that in H358 and A549 cells, Carboplatin induced both DDR and the Notch pathway, validating our experimental system to understand how the Notch pathway is increased upon platinum treatment.

NICD levels can be increased through enhanced NOTCH1 processing on the membrane promoted by increased ligand and/or full-length NOTCH1 expression, or by NICD stabilization (Bray and Gomez-Lamarca, 2018). To determine by which mechanism NICD levels were increased upon DNA damage induction, we analyzed the protein expression of all Notch ligands (i.e., JAGGED1, JAGGED2, DLL1, DLL3 and DLL4) and full-length NOTCH1 after Carboplatin incubation. In both H358 and A549 cells, phosphorylated ATM and NICD were consistently induced by Carboplatin treatment, but this was not the case for full-length NOTCH1 or its ligands (Figs. 1D and EV1D). This suggested that the increased NCID expression could be explained by enhanced stability. To test this hypothesis, we incubated both cell lines with Carboplatin or vehicle for 48 h and added Cycloheximide (a translation inhibitor) at different time-points before the end of the exposure to Carboplatin. Importantly, NICD stability was increased in cells incubated with Carboplatin compared with control in both H358 (Fig. 1E,F) and A549 cells (Fig. EV1E), thus independently of p53 expression.

Altogether, our results indicated that DNA damage increases NICD stability.

## NICD stabilization upon DNA damage is ATM-dependent

As we found that three different genotoxic agents increased NICD levels, we hypothesized that the DDR could promote the NICD increased stability upon Carboplatin treatment. As the DDR is

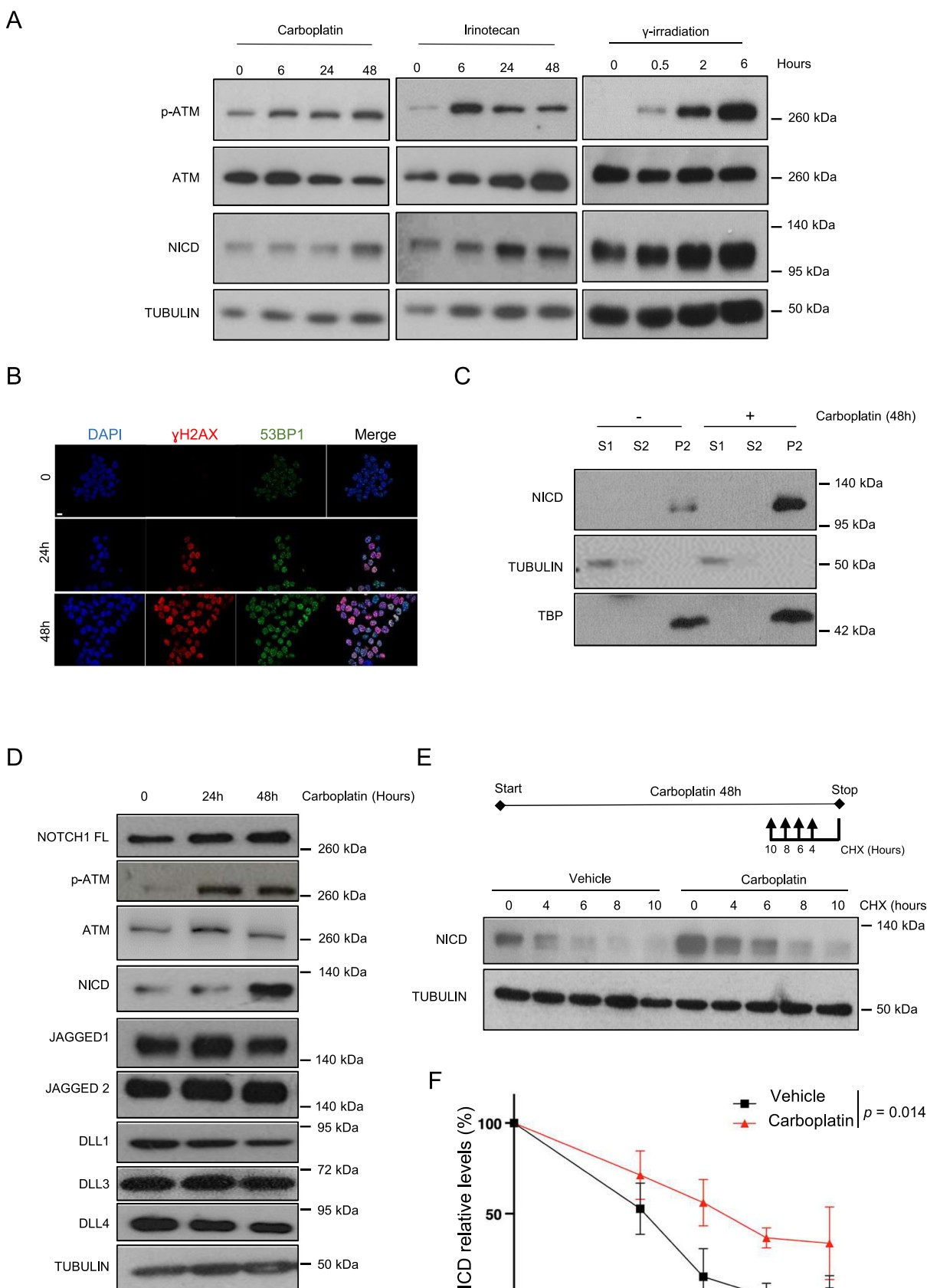

**Figure 1.  DNA damage induces NICD stabilization.**

(A) Western blotting of the indicated proteins in H358 cells incubated with 100 µM Carboplatin ($n = 2$), 25 µM irinotecan ($n = 2$), or γ-irradiation (4 Gy) ($n = 1$) for the indicated times. p-ATM, phosphorylated (activated) ATM. (B) Immunofluorescence analysis of γH2AX (in red) and 53BP1 (in green) expression in H358 cells after 24 or 48 h of incubation with 100 µM Carboplatin ($n = 1$). Nuclei were stained with DAPI (blue). Scale bar: 10 µm. (C) Western blotting of the indicated proteins after cellular fractionation of H358 cells incubated ($+$) or not ($-$) with 100 µM Carboplatin for 48 h ($n = 3$). S1 cytoplasmic fraction, S2 nucleoplasmic fraction, P2 chromatin fraction, TUBULIN loading control for S1, TBP loading control for P2. (D) Western blotting of the indicated proteins in H358 cells incubated with 100 µM Carboplatin for the indicated times ($n = 1$). (E) Western blotting of the indicated proteins in H358 cells incubated or not with 100 µM Carboplatin for 48 h and with 50 µg/µl Cycloheximide (CHX) for the indicated times before the end of Carboplatin treatment ($n = 3$). Upper panel: Schematic representation of the experiment design. (F) Graph showing the mean ± SEM of three different experiments performed as described in (E) and the statistics between curves was determined using a modified Chi-square-based method. NICD level was normalized to TUBULIN at each time point and then to untreated sample (lane 0) that was set at 100%. Hours correspond to Cycloheximide treatment. In the legend, $n$ indicates the number of biological replicates. Source data are available online for this figure.

orchestrated by two main DNA damage sensor kinases (ATM and ATR), we used specific inhibitors to test their possible role in NICD stabilization upon DNA damage. We pre-incubated H358 and A549 cells with KU55933 (ATM inhibitor) and Ceralasertib (ATR inhibitor) (Foote et al, 2018) for 2 h before Carboplatin addition. ATM and ATR were efficiently inhibited, as indicated by the decreased levels of their phosphorylated forms (Figs. 2A and EV2A). ATR inhibition did not decrease NICD accumulation upon DNA damage. Conversely, NICD levels in H358 and A549 cells incubated with Carboplatin were strongly decreased by ATM inhibition (Figs. 2A and EV2A). To confirm that ATM controlled NICD stability upon Carboplatin treatment, we performed Cycloheximide pulse-chase experiments in the presence of Carboplatin and the ATM inhibitor. As before NICD levels were stabilized for 8 h in cells incubated with only Carboplatin. Conversely, NICD half-life was strongly reduced in cells co-incubated with the ATM inhibitor and was barely detectable after 4 h (Figs. 2B,C and EV2B).

Our findings suggest that ATM is required for increasing NICD stability upon DNA damage in a p53-independent manner.

## NICD is ubiquitinylated upon DNA damage and MDM2 is required for NICD stabilization

To finely control Notch signaling, NICD is regulated by various post-translational modifications, including ubiquitination that promotes NICD degradation (Lai, 2002). Therefore, to determine whether Carboplatin induces NICD ubiquitination, we used a well-characterized system in which ectopic expression of His6-tagged ubiquitin allows the purification of de novo ubiquitinated proteins (Treier et al, 1994). We generated a NOTCH1 delta extracellular domain construct (NOTCH1-deltaEx) (see "Methods" for details) to express a ligand-independent but γ-secretase-dependent NOTCH1 protein. This allowed increasing NICD expression in cells, but not in a supraphysiological manner as it is frequently the case upon direct NICD overexpression. We co-transfected the NOTCH1-deltaEx and the His6-tagged ubiquitin plasmids in 293 T cells and then incubated them with Carboplatin as before. Due to the increased NICD expression following DNA damage induction, we expected lower NICD ubiquitination, but surprisingly, it was strongly increased (Fig. 3A).

A previous work showed that the E3 ligase MDM2 interacts with and ubiquitinates NICD, leading to hyperactivation of the Notch pathway and not to NICD degradation (Pettersson et al, 2013). Although this study was performed in the absence of any type of DNA damage, MDM2 is an important DDR downstream actor and furthermore, it displays p53-independent functions (Bouska and

Eischen, 2009), making it a good E3 ligase candidate for DNA damage-induced NICD ubiquitination. Thus, we first analyzed MDM2 protein levels in H358 and A549 cells after incubation with Carboplatin. Intriguingly, in both cell lines (i.e., p53-independent), MDM2 expression increased upon DNA damage induction, but decreased when cells were co-incubated with the ATM inhibitor, mirroring the patterns of phosphorylated ATM and NICD (Figs. 3B and EV2C).

As these data suggested that MDM2 could be responsible for NICD increased stability upon DNA damage, we then analyzed the effect of MDM2 loss of function in this setting. In accordance with our previous data, in control cells (siNT-treated), NICD protein levels increased upon Carboplatin treatment. Conversely, in siMDM2-treated H358 and A549 cells, NICD expression was not increased after incubation with Carboplatin (Figs. 3C and EV2D), validating our hypothesis. To further demonstrate that MDM2 promoted NICD stability upon Carboplatin treatment, we monitored NICD half-life in siMDM2- and siNT-treated cells incubated with Cycloheximide and Carboplatin. In both siMDM2-treated H358 and A549 cells, NICD stabilization was strongly decreased after Carboplatin incubation (Figs. 3D,E and EV2E).

Altogether, our data show that upon DNA damage, NICD is ubiquitinated and MDM2 is required to increase its stability independently of p53.

## MDM2 promotes NICD stabilization through its ubiquitination

Then, to determine whether MDM2 was sufficient to stabilize NICD, we ectopically expressed wild-type MDM2 and NOTCH1-deltaEx in human 293T cells. To demonstrate that MDM2 E3 ligase activity was required for such stabilization, we expressed an E3 ligase-dead MDM2 mutant (C464 A point mutation in the RING domain) (Riscal et al, 2016). Moreover, to further link ATM to MDM2 in NICD stability regulation, we also expressed the MDM2 S395A mutant, in which ATM-induced phosphorylation in this residue is abolished (Maya et al, 2001). Interestingly, in the absence of DNA damage, ectopic expression of wild-type MDM2 was sufficient to increase NICD levels in the chromatin fraction (Fig. 4A). Of note, we did not observe this effect with the two MDM2 mutants, indicating that both E3 ligase activity and S395 phosphorylation by ATM are required for the MDM2-mediated NICD increase.

To formally prove that NICD is a MDM2 E3 ligase substrate and also that the S395 residue plays a critical role in the MDM2-mediated ubiquitination, we ectopically expressed NOTCH1-

**A**

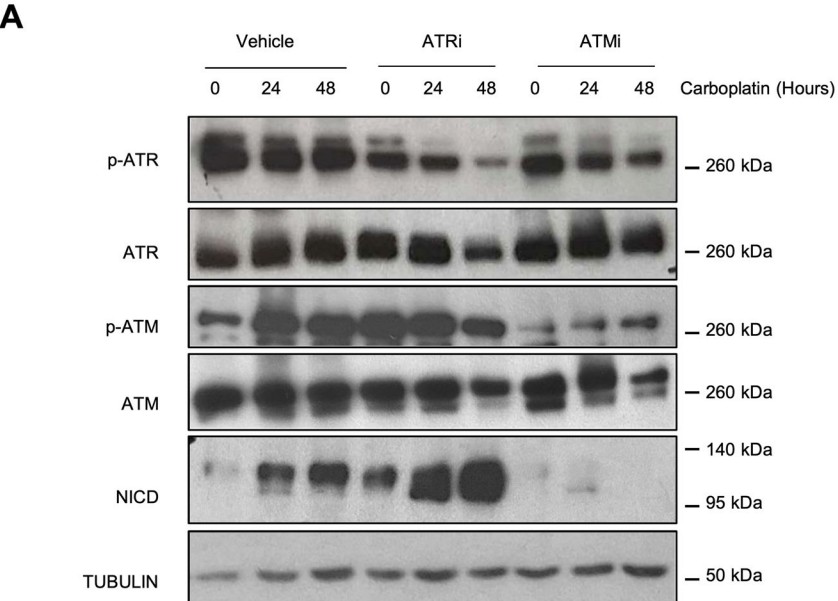

**B**

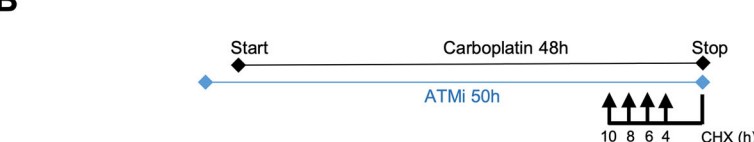

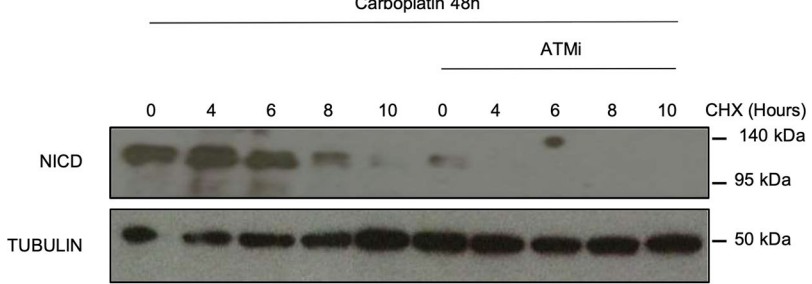

**C**

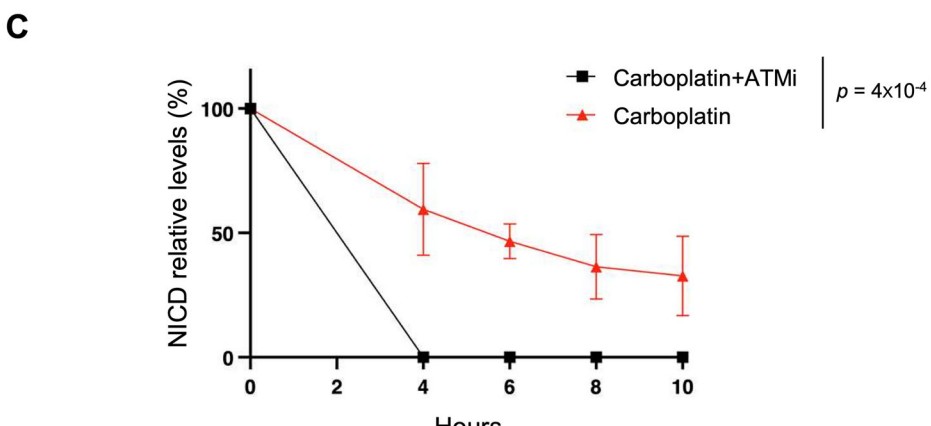

◄ **Figure 2. NICD stabilization upon DNA damage is ATM-dependent.**

(A) H358 cells were incubated with 100 µM Carboplatin and with/without 10 µM KU-55933 (ATM inhibitor) or 5 µM Ceralasertib (ATR inhibitor) for the indicated time followed by immunoblotting to detect the expression of the indicated proteins ($n = 2$). P phosphorylated. (B) H358 cells were incubated with 100 µM Carboplatin for 48 h and with or without 10 µM KU-55933 (ATM inhibitor). 50 µg/µl Cycloheximide (CHX) was added for the indicated time before the end of Carboplatin treatment ($n = 3$). Inhibitors were added 2 h before Carboplatin addition. Upper panel: Schematic representation of the experimental design. (C) Graph showing the mean ± SEM of three different experiments performed as described in (B) and the statistics between curves was determined using a modified Chi-square-based method. NICD level was normalized to TUBULIN at each time point and then to untreated sample (lane 0) that was set at 100%. Hours correspond to Cycloheximide treatment. In the legend, $n$ indicates the number of biological replicates. Source data are available online for this figure.

deltaEx, His6-tagged ubiquitin, and wild-type MDM2, MDM2 C464A or MDM2 S395A in 293T cells and purified ubiquitinated proteins. In the absence of ectopic wild-type MDM2, NICD was weakly ubiquitinated (Fig. 4B, line 2 in the right panel). Importantly, NICD ubiquitination was strongly increased in cells transfected with wild-type MDM2 (Fig. 4B, line 3 in the middle panel), but not in cells transfected with the ligase-dead MDM2 C464A mutant or the MDM2 S395A mutant that cannot respond to ATM phosphorylation (Fig. 4B, lines 4 and 5 in the right panel). As a positive control, we showed that ubiquitination of p53 (the best-known target of MDM2 E3 ligase activity) was increased by ectopic expression of wild-type MDM2 (Fig. 4B, line 3 compared with line 2 in right panel) but not of the ligase-dead MDM2 464A mutant (Fig. 4B, line 4 in right panel). Furthermore, as previously described (Khosravi et al, 1999; Maya et al, 2001), MDM2 S395A increased p53 ubiquitination compared with wild-type MDM2 (Fig. 4B, line 5 in the right panel), further supporting the hypothesis that ATM-mediated MDM2 S395 phosphorylation is specifically implicated in NICD ubiquitination regulation.

To assess whether the MDM2 and NICD interaction shown previously (Pettersson et al, 2013) occurs in our setting and whether the MDM2 E3 ligase function or the ATM-mediated phosphorylation of S395 are implicated in this interaction, we performed co-immunoprecipitation experiments using 293T cells that ectopically express NOTCH1-deltaEx and wild-type MDM2, MDM2 C464A or MDM2 S395A. We observed that MDM2 and NICD were co-precipitated independently of MDM2 mutational status (Fig. EV3), confirming their interaction in our experimental setting. Although MDM2 E3 ligase activity and MDM2 S395 phosphorylation were not required for this interaction, they are crucial for MDM2-induced NICD ubiquitination and stabilization.

In the same seminal study (Pettersson et al, 2013), the authors used the ubiquitin K48R mutant. This lysine is required for ubiquitin chain generation and recognition by the proteosome (Petroski and Deshaies, 2005). Indeed, Petterson and colleagues showed that MDM2 induces NICD ubiquitination even in the presence of the K48R mutant, indicating that this ubiquitination is not degradative. Therefore, to confirm that MDM2-mediated ubiquitination of NCID in our experimental settings was neither degradative, we transfected cells with NOTCH1-deltaEx, His6-tagged wild-type ubiquitin or ubiquitin K48R, and wild-type MDM2. Unlike MDM2, several E3 ligases (e.g., FBX7, Itch or Sel-10) ubiquitinate NICD and induce its degradation to fine tune Notch activity (Lai, 2002) (Welcker and Clurman, 2008) and accordingly, NICD protein levels accumulated in the presence of K48R mutant compared to wild-type ubiquitin (Fig. 4C, left panel). Furthermore, again we observed that wild-type MDM2 ectopic expression induced NICD ubiquitination when co-expressed with

ubiquitin wild-type, and this effect was increased in cells expressing the ubiquitin K48R mutant (Fig. 4C, middle and right panels). This confirmed that MDM2-mediated NICD ubiquitination does not promote NICD degradation.

Altogether, our findings indicate that upon DNA damage, MDM2 induces the ubiquitination of NICD to increase its protein stability.

## Inhibition of the MDM2-NICD axis enhances platinum effectiveness and increases survival in vivo

Then, to prove that the MDM2-mediated NICD stability increase upon DNA damage protects cancer cells against platinum treatment, we generated a stable H358 cell line in which NOTCH1-deltaEx expression can be induced by doxycycline (see "Methods"). First, we confirmed that doxycycline effectively increased NICD levels in our de novo generated cell line (Fig. 5A). Interestingly, we showed that NICD expression induction was sufficient to diminish Carboplatin effectiveness compared with parental H358 cells (Fig. 5A).

Based on this encouraging result, we investigated whether Carboplatin effect could be enhanced by inhibiting the MDM2-NICD axis in vivo. To increase the clinical relevance of our approach, we used a NSCLC PDX that displays intrinsic resistance to Carboplatin (Ferrer et al, 2018). We let the tumors grow to 200 mm³ before randomization into six treatment groups ($n = 8$–9 mice/group): vehicle; Carboplatin; Carboplatin with SP141, an MDM2 inhibitor with proven therapeutic benefit in preclinical in vivo models of breast cancer (Wang et al, 2014); Carboplatin with dibenzazepine (DBZ), a γ-secretase inhibitor that abolishes NICD formation and that is effective in NSCLC PDXs in vivo (Bousquet Mur et al, 2020); SP141 with DBZ; and the combination of the three drugs. We administered SP141 and DBZ for five consecutive days per week and Carboplatin once per week (in the middle of each 5-day cycle) (see treatment schedule in Fig. EV4A). We noticed that the SP141 and DBZ combination had a deleterious effect on body weight and we gave drug holidays to all groups when some mice displayed strong weight loss (weight was monitored throughout the treatment) (Fig. EV4B). Some animals never recovered the weight loss and we euthanized them for ethical reasons. Specifically, we euthanized 3/9 mice in the SP141 + DBZ group. Finally, in the triple combination group, addition of Carboplatin further enhanced the weight loss and we euthanized 8/9 mice. Therefore, we did not consider this group for analysis.

We measured tumor growth in all mice until the first animal in each group reached the end point (tumor volume of 1500 mm³): day 25 for the vehicle group, day 42 for the Carboplatin+DBZ group, and day 32 for the other groups. This indicated that tumor

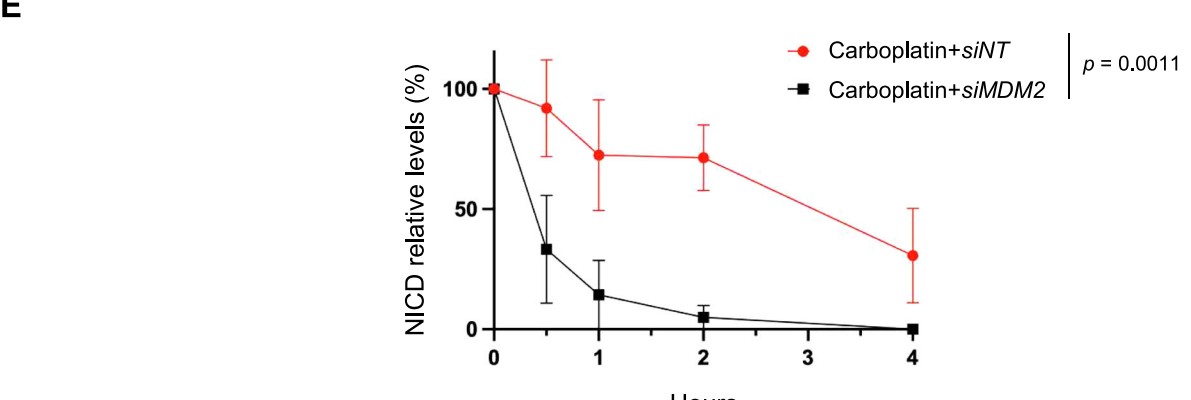

growth was much lower in the Carboplatin+DBZ group compared with the Carboplatin alone group (Fig. 5B). The survival analysis showed that Carboplatin had only a small and non-significant effect compared with vehicle (Fig. 5C), confirming in our experimental setting that this PDX model is intrinsically resistant to platinum

treatment as previously described (Ferrer et al, 2018). Remarkably, survival was strongly increased in the Carboplatin+DBZ (NICD inhibitor) group compared with the vehicle and Carboplatin alone groups (Fig. 5C). Interestingly, the only other group that showed increased survival compared to vehicle was the Carboplatin plus

**Figure 3.  NICD is ubiquitinylated upon DNA damage and its stabilization is MDM2-dependent.**

(A) 293 T cells were transfected to express NOTCH1-DeltaEx, 6His-ubiquitin (UB-HIS) or empty vector (pcDNA), as indicated, and incubated (+) or not (−) with Carboplatin for 48 h. Histidine pull-down was performed and NICD ubiquitination was analyzed by western blotting ($n = 3$). INPUT, total cell extracts. (B) Western blotting of the indicated proteins in H358 cells incubated with 100 μM Carboplatin with or without 10 μM KU-55933 (ATM inhibitor) for the indicated time ($n = 3$). (C) Western blotting of the indicated proteins in H358 cells in which *MDM2* was silenced with a *siRNA* against MDM2 (*siMDM2*) or not (non-targeting *siRNA*; *siNT*) ($n = 2$). At 6 h post-transfection, 100 μM Carboplatin was added to the cells for 48 h. (D) H358 cells transfected with *siMDM2* or *siNT* were incubated with 100 μM Carboplatin for 48 h and with 50 μg/μl of Cycloheximide (CHX) for the indicated time before the end of Carboplatin treatment ($n = 3$). Upper panel: schematic representation of the experimental design. (E) Graph showing the mean ± SEM of three different experiments performed as described in (D) and the statistics between curves was determined using a modified Chi-square-based method. NICD level was normalized to TUBULIN at each time point and then to the level in the untreated sample (lane 0) set to 100%. Hours correspond to Cycloheximide treatment. In the legend, *n* indicates the number of biological replicates. Source data are available online for this figure.

MDM2 inhibitor (SP141) treatment group. The median survival was 31, 40.5, 38.5, 44 and 56 days for the vehicle, Carboplatin, SP141 + DBZ, Carboplatin+SP141 and Carboplatin+DBZ groups, respectively (Fig. 5C).

To confirm proper therapeutic target inhibition, we treated other cohorts of nude mice ($n = 3$ mice/group) harboring same NSCLC PDX xenografts with SP141 and/or DBZ for 4 days and once with Carboplatin (acute treatment). At 24 h after the Carboplatin injection (D2 in Fig. EV4A) and 6 h after the last treatment with SP141 and/or DBZ, we assessed γH2AX, cleaved caspase 3 (CC3), HES1 (a downstream target of NICD) and Ki67 expression levels in PDX-derived tumors by immunohistochemistry (IHC) and calculated their histoscore (H-score) (Hirsch et al, 2003). Among all the groups that received Carboplatin (alone or with SP141 or DBZ), the γH2AX H-score was significantly increased only in the Carboplatin+DBZ group compared with vehicle, indicating that Notch inhibition increased DNA damage (Figs. 5D and EV4C). Similarly, the CC3 H-score was increased only in the Carboplatin+DBZ group, whereas Ki67 H-score was similar in all groups. This indicated that the effect of the Carboplatin+DBZ combination was due to apoptosis and not to decreased proliferation (Figs. 5D, EV4C and EV5A). Lastly HES1 H-score was drastically decreased in the Carboplatin+DBZ and SP141 + DBZ groups (Figs. EV4C and EV5A), indicating a proper target inhibition of NICD by DBZ. However, HES1 H-score in the Carboplatin group was not different from that of the vehicle group. This was intriguing because in the absence of Carboplatin, HES1 is a suitable IHC read-out of Notch pathway activation in both KRAS- and EGFR-driven NSCLC (Bousquet Mur et al, 2020; Maraver et al, 2012). We concluded that HES1 is not a reliable read-out of Notch activity increase promoted by DNA damage, but it is still a good marker of Notch inhibition.

To demonstrate Notch pathway activation upon Carboplatin treatment in vivo, and as NICD expression cannot be detected by IHC, we determined NICD expression in tumors by western blotting and found that in accordance with our in vitro data, it was increased in the Carboplatin group (Figs. 5E and EV5B). Conversely, NICD expression was slightly decreased in the Carboplatin+SP141 group compared with the Carboplatin group, and very strongly diminished in the Carboplatin+DBZ group (Figs. 5E and EV5B). Western blot analysis of HES1 expression confirmed the IHC data (no increase upon Carboplatin treatment). On the other hand, the expression of HEY1, another bHLH transcription factor downstream of NICD (Weber et al, 2014), was increased by Carboplatin treatment (Figs. 5E and EV5B), further confirming the Notch pathway activation by Carboplatin in vivo. Of note, co-

treatment of Carboplatin with SP141 or DBZ abolished HEY1 expression increase. Lastly, Carboplatin increased MDM2 protein levels (Figs. 5E and EV5B), confirming our in vitro data, whereas Notch inhibition (DBZ) decreased this effect, indicating that a feed forward loop could exist between NICD and MDM2.

Taken together, our in vivo preclinical data establish that MDM2-NICD axis inhibition enhances Carboplatin therapeutic effect in NSCLC.

## MDM2 expression correlates with poor progression-free survival in patients with NSCLC treated with platinum compounds

To confirm the clinical relevance of the MDM2-NICD axis in NSCLC, we analyzed primary tumor samples from a cohort of 41 patients with NSCLC following treatment with platinum alone or in combination with other treatments (see "Methods" for details). As we showed that HES1 was an unreliable readout of the Notch pathway activation in the presence of DNA damage, we only analyzed MDM2 expression (using the H-score). We ranked patients based on MDM2 expression on tumor cells, from the highest to the lowest H-score, and separated them into four quartiles. We compared progression-free survival (PFS) in the groups with the lowest (i.e., the lowest quartile) and the highest MDM2 expression (i.e., the highest quartile). The median PFS was 29.5 months in the lowest MDM2 expression group and 5.4 months in the highest MDM2 expression group (hazard ratio 5.359, 95% confidence interval, 1.695–16.94, $P = 0.004$) (Fig. 6A). This robust difference suggests that MDM2 plays a major role in the response to platinum-based chemotherapy in patients with NSCLC.

Lastly, we assessed NICD protein levels in a smaller cohort of patients with NSCLC and available frozen tumor tissue after surgery. All had received platinum-based neoadjuvant chemotherapy before surgery and were classified as chemosensitive ($n = 3$) and chemo-resistant ($n = 5$) in function of the tumor cell clearance in adjacent lymph nodes. MDM2 and NICD expression levels were strongly and similarly increased in four of the five chemo-resistant patients, suggesting that the MDM2 control of NICD stability observed in NSCLC human cell lines may occur also in patients with NSCLC. Conversely, we could not detect MDM2 and NICD in the three chemosensitive patients (Fig. 6B). These data strongly suggest that the MDM2-NICD axis plays an important role in the response to platinum in patients with NSCLC.

Our clinical data confirmed and expanded the mechanistic findings showing that the MDM2-NICD axis plays a major role in the DDR and Carboplatin effectiveness in NSCLC.

**A**

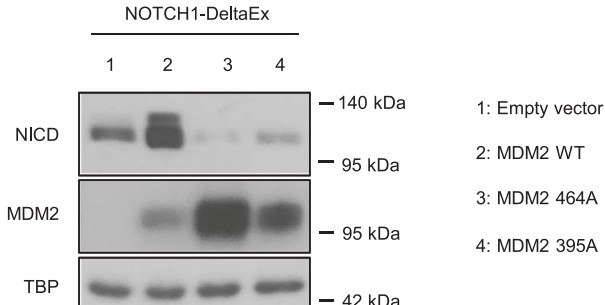

NICD

MDM2

TBP

1: Empty vector

2: MDM2 WT

3: MDM2 464A

4: MDM2 395A

Chromatin fraction

**B**

INPUT

PULLDOWN
HIS-Ub

**C**

INPUT

PULLDOWN
HIS-Ub

◀ **Figure 4. MDM2 E3-ligase activity is required for NICD ubiquitination.**

(A) 293 T cells were transfected with NOTCH1-DeltaEx and then co-transfected with empty vector (pcDNA), MDM2 wild-type (WT), MDM2 464 A or MDM2 395 A for 48 h. Then, NICD levels in the chromatin fraction were analyzed by western blotting after cell fractionation (n = 3). TBP: loading control for the chromatin fraction. (B) 293T cells were transfected with NOTCH1-DeltaEx and then co-transfected with 6xtag-His-ubiquitin (UB-HIS), empty vector (pcDNA), MDM2 WT, MDM2 464 A or MDM2 395 A for 48 h followed by histidine affinity pull-down (pulldown HIS-Ub) and western blotting (n = 1). INPUT: total cell extracts. TBP was used as control loading in INPUT. Graph shows the quantification of NICD signal in each pulldown fraction relative to the UB-HIS signal in the same fraction. Then, NICD signal in each lane was normalized to the signal in lane 2 (empty vector) set at 100%. For p53, the signal in MDM-395A-expressing cells was normalized to that in lane 3 (MDM2 WT) set at 100%. Lane 1 is the pulldown negative control (without ubiquitin). (C) 293 T cells were transfected with NOTCH1-DeltaEx and then co-transfected with 6xtag-His-ubiquitin (UB-HIS) wild-type (WT) or K48R mutant, in combination with empty vector (pcDNA) or MDM2 WT for 48 h, followed by histidine affinity pull-down (pulldown HIS-Ub) and western blotting (n = 2). INPUT: total cell extracts. TBP was used as loading control in INPUT. Graph shows the quantification of the NICD signal in the pull-down fraction relative to the UB-HIS signal in the same fraction. Then, NICD signal in each lane was normalized to that in lane 3 (MDM2 WT) set at 100%. Lane 1 is the pull-down negative control (without ubiquitin expression). In the legend, n indicates the number of biological replicates. Source data are available online for this figure.

## Discussion

In the last years, massive research efforts have been made to find new therapeutic options for NSCLC, the deadliest cancer worldwide (Organization, 2021). These efforts unveiled unprecedented and innovative treatments, such as immune checkpoint- and KRAS$^{G12C}$-inhibitors. Nevertheless, platinum-based chemotherapy remains the most used treatment for patients from early stages to metastatic lung cancer (Hendriks et al, 2023a; Hendriks et al, 2023b), and hence overcoming resistance against this therapeutic option is an unmet medical need. To develop new drug combinations that tackle the appearance of resistance, it is crucial to understand the molecular changes occurring in lung cancer cells upon treatment with platinum-based drugs.

In the current study, we first discovered that DNA damage activates the Notch pathway. We then showed that ATM, one of the main downstream sensors of DDR, is implicated in Notch signaling activation. As previous studies showed that Notch inhibits ATM (Vermezovic et al, 2015), we hypothesized that in physiological conditions, the ATM-NOTCH crosstalk may establish a negative feedback loop to regulate ATM activation upon DDR. Then, we showed that upon incubation with DNA damage-inducing agents, NICD ubiquitination increases concomitantly with its stability. This was a counterintuitive observation because it is normally accepted that ubiquitination promotes NICD degradation (Lai, 2002). This result led us to MDM2 because, to the best of our knowledge, it is the only E3 ubiquitin ligase reported to interact with and activate NICD (Pettersson et al, 2013). We found that MDM2-mediated stabilization of NCID is p53-independent, although it is widely known that MDM2 also binds to p53 and promotes its ubiquitination and degradation to tightly control its activity. This is in agreement with recent studies showing that some MDM2 roles in cancer are p53-independent (Arena et al, 2018; Cisse et al, 2020; Riscal et al, 2016) and also with in vivo studies demonstrating that cisplatin effect in lung cancer cells is p53-independent (Oliver et al, 2010).

One key observation in our work is that MDM2 accumulated upon DNA damage induction in an ATM-mediated manner. Previous studies showed that in mice, upon DDR, ATM phosphorylates MDM2 at S394 (S395 in human) (Gannon et al, 2012). In accordance, we observed that MDM2 S395 is also crucial for NICD ubiquitination and increased expression, unveiling a previously undescribed ATM-MDM2-NICD axis that operates upon Carboplatin treatment. These findings are also in agreement with a seminal study showing that MDM2 ubiquitinates NICD, but not for degradation (Pettersson et al, 2013). Still, at this stage we do not know: (i) whether MDM2 promotes NICD mono- or poly-ubiquitination; (ii) which of the 17 lysine residues in NICD is (are) targeted by MDM2, and (iii) whether the MDM2-targetd lysine(s) differ(s) from those recognized by E3 ligases that ubiquitinate NICD for degradation (Lai, 2002) (Welcker and Clurman, 2008). All these questions are currently studied in the laboratory. Moreover, NICD is tightly controlled at the post-transcriptional level, not only through ubiquitination, but also through, for instance, phosphorylation (Braunreiter and Cole, 2019), hydroxylation (Ferrante et al, 2022), acetylation (Guarani et al, 2011) and methylation (Hein et al, 2015). Therefore, it is not surprising to detect NICD bands of different sizes. In our experience, these different bands (sometimes >2) are not associated with any particular condition (e.g., treatment or cell line), and we plan to study this issue in detail using proteomics. In any case, we showed that MDM2 stabilizes NICD in the absence of DNA damage, and this finding could partially explain why MDM2 amplification is an independent factor of poor prognosis in patients with NSCLC (Dworakowska et al, 2004). Moreover, by analyzing NSCLC samples from a subset of patients treated with platinum-based chemotherapy, we found that NICD and MDM2 expression levels were positively correlated and that the MDM2-NICD axis was activated only in non-responder patients. Lastly, in another cohort of patients with NSCLC and treated with platinum-based chemotherapy, we found that MDM2 expression level was inversely correlated with PFS. Therefore, our data strongly suggest that high MDM2 expression, by amplification or other mechanisms, will affect the response to platinum-based chemotherapy in NSCLC by stabilizing NICD.

As preclinical in vivo model, we used nude mice xenografted with a NSCLC PDX that displays intrinsic resistance against platinum. Accordingly, in these mice, treatment with Carboplatin alone did not increase γ-H2AX expression in tumor cells compared with vehicle, unlike what observed after treatment with Carboplatin +DBZ (Notch inhibitor). Therefore, although at this stage we cannot rule out that other mechanisms besides DNA damage are applying in our experimental system, our data are in accordance with previous studies on ovarian and cervical cancers showing that NICD inhibition enhances DNA damage upon platinum treatment (Li et al, 2019; McAuliffe et al, 2012). Moreover, a study on colorectal cancer found that NICD inhibition increases radiation-induced DNA damage (Zhang et al, 2018). In addition, using a *D. melanogaster* ovarian tumor model, a previous study showed that NICD ectopic expression promotes the upregulation of double-

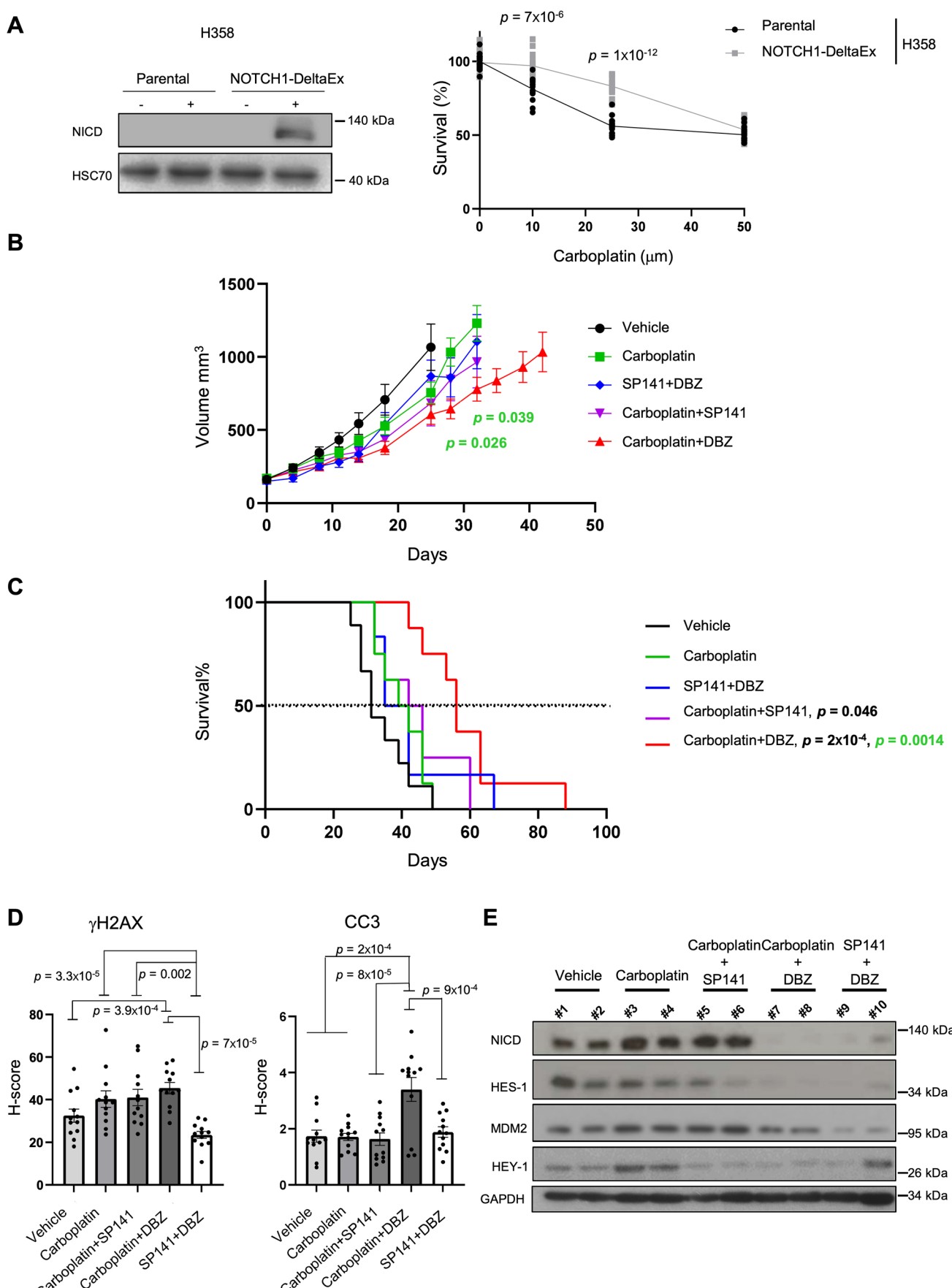

◄ **Figure 5. Inhibition of the MDM2-NICD axis enhances platinum effectiveness and increases survival in vivo.**

(A) Parental and NOTCH1-DeltaEx H358 cells were incubated with doxycycline (0.01 μg/ml) to induce NICD expression (western blot in left panel) and different concentrations of Carboplatin (0, 10, 25 and 50 μM) for 3 days. Then, cell viability was measured ($n = 3$). Graph shows the mean ± SEM of three different experiments. $P$ values were obtained by multiple unpaired $t$ test. HSC70 was used as loading control. (B) Growth of PDX TP57 tumors in nude mice treated with vehicle ($n = 9$), Carboplatin ($n = 8$), SP141 (MDM2 inhibitor) + DBZ (Notch inhibitor) ($n = 6$), Carboplatin + SP141 ($n = 8$), or Carboplatin + DBZ ($n = 8$). The $y$ axis shows the tumor volume measured with a caliper. Data are the mean ± SEM. Green $P$ values indicate significant differences between the Carboplatin group and the Carboplatin + DBZ group (two-way ANOVA followed by Tukey's multiple comparison test). (C) Survival analysis in the same animals described in (B). The $y$ axis shows the percentage of surviving animals and the $x$ axis the days after treatment. Survival curves were compared with the log rank (Mantel–Cox) test and significant differences were represented by black bold $P$ values for the treated groups compared with the vehicle group and green $P$ value for treated groups compared with the Carboplatin group. (D) IHC analysis of γH2AX and CC3 expression levels in PDX TP57 xenografts from nude mice treated with vehicle, Carboplatin alone, Carboplatin + SP141, Carboplatin + DBZ, or SP141 + DBZ ($n = 12$ measures from three different tumors for all groups). Data are the mean ± SEM and they correspond to the analysis of 4 fields (original magnification, ×20) per tumor. $P$ values were obtained by one-way ANOVA followed by Tukey's multiple comparison test. (E) The indicated proteins were assessed by western blotting in the same mouse groups as in (D) ($n = 2$ different tumors/group). In the legend, $n$ indicates the number of biological replicates. Source data are available online for this figure.

strand break repair genes (Jevitt et al, 2021). We are investigating whether a similar molecular mechanism could occur in mammalian cells.

Overall, our data suggest that targeting the MDM2-NICD axis could enhance Carboplatin therapeutic effect in patients with NSCLC. An important question now is how to identify the patients who could benefit from Notch inhibition. To this aim, at our cancer center (ICM), we are currently conducting a clinical trial to monitor Notch activity in KRAS[G12C]- and EGFR-mutated patients with NSCLC using HES1 as a read-out for the Notch pathway (Alcina 2 cohort 11; NCT04025541). Therefore, our clinical research could pave the way for future clinical trials to test the effect of Notch inhibitors in patients with NSCLC treated with targeted therapy. By contrast, in patients with NSCLC treated with Carboplatin, we would need to use another surrogate marker of Notch activation, for instance HEY1. Indeed, here we found that HES1 does not mirror Notch increased expression after Carboplatin treatment, unlike what observed in patients with EGFR mutated patients and with EGFR targeted therapy (Bousquet Mur et al, 2020). Additional work is required to understand the reason behind this unusual observation, but we anticipate that it might be related to the uncoupling of HES1 and NICD signaling upon DNA damage. For instance, as HES1, but not NICD, is present in the Fanconi anemia core complex (Tremblay et al, 2008), we could hypothesize that other pathways regulate HES1 expression upon DNA damage.

Overall, our data propose a new therapeutic opportunity for patients who do not respond or become resistant to platinum-based chemotherapy, the most common treatment for lung cancer, the leading cause of death by cancer worldwide (Organization, 2021).

# Methods

### Reagents and tools table

| Reagent/resource | Reference or source | Identifier or catalog number |
|---|---|---|
| **Experimental models** | | |
| HEK-293 cells (*H. sapiens*) | ATCC | CRL-1573 |
| H358 (*H. sapiens*) | ATCC | CRL-5807 |

| Reagent/resource | Reference or source | Identifier or catalog number |
|---|---|---|
| A549 (*H. sapiens*) | ATCC | CCL-185 |
| Athymic Nude-Foxn 1 | Charles River | Crl:NU(NCr)-Foxn1<nu> |
| **Recombinant DNA** | | |
| psPAX2 | Addgene | 12260 |
| PVSV G vector | Addgene | 138479 |
| pCLX-R4-DEST-R2 | Addgene | 45956 |
| MDM2 wt and mutants | Riscal et al (2016) | N/A |
| HIS-Ubiquitin | Treier et al (1994) | N/A |
| NOTCH1-DeltaEx | This study | N/A |
| **Antibodies** | | |
| Anti-rabbit | Cell Signaling Technology | 7077 |
| Anti-mouse | Cell Signaling Technology | 7076 |
| Phosphorylated ATR | Cell Signaling Technology | 2853 |
| ATR | Cell Signaling Technology | 2790S |
| JAG1 | Cell Signaling Technology | 70109S |
| JAG2 | Cell Signaling Technology | 2210T |
| DLL3 | Cell Signaling Technology | 78110S |
| DLL4 | Cell Signaling Technology | 96406S |
| NOTCH1 | Cell Signaling Technology | 3608S |
| GAPDH | Cell Signaling Technology | 2118S |
| DLL1 | Antivia Biosciences | QC16739-41705 |
| MDM2 | Merck Millipore | 4B2C1.11 |
| MDM2 | Merck Millipore | 4B11 |
| Tubulin | Merck Millipore | T9026 |
| p53 | Santa Cruz Technology | SC-126 |
| TFIID/TATA Binding Protein/TBP (1TB18) | Santa Cruz Technology | SC-56794 |
| HSC70 | Santa Cruz Technology | SC-7298 |
| HEY1 | Proteintech | 19929 |
| polyhistidine | Sigma-Aldrich | H1029 |
| γH2AX | Merck Millipore | |
| 53BP1 | Merck Millipore | MAB3802 |

| Reagent/resource | Reference or source | Identifier or catalog number |
|---|---|---|
| Anti-rabbit Alexa Fluor 488 | Thermo Fisher Scientific | A11008 |
| Anti-mouse Cy3 | Thermo Fisher Scientific | A10521 |
| NICD | Cell Signaling Technology | 4147 |
| Phosphorylated ATM | Cell Signaling Technology | 13050 |
| ATM | Cell Signaling Technology | 2873S |
| H3 | Cell Signaling Technology | 59715S |
| Anti-NOTCH1 | Abcam | Ab27526 |
| Anti-HES1 clone D6P2U | Cell Signaling Technology | 11988 |
| Anti-MDM2 clone IF2 | Merck Millipore | MABE340 |
| **Oligonucleotides and other sequence-based reagents** | | |
| Smartpool siRNA MDM2 | Thermo Scientific Dharmacon | L-003279-00-0005 |
| **Chemicals, enzymes and other reagents** | | |
| Cycloheximide | SIGMA ALDRICH CHIMIE | C7698-1G |
| Acetic acid | SIGMA ALDRICH | 8.18755 |
| Irinotecan | Fresenius Kabi | N/A |
| JetPEI | Polyplus | POL101000053 |
| polybrene | BPS Bio Science | 78939 |
| RPMI 1640 | Thermo Fisher Scientific | SLM 240 |
| puromycin | Invivogen | ANT-PR-1 |
| Trichloroacetic acid | Sigma-Aldrich | T9159-500g |
| Triton X100 | Sigma-Aldrich | X100 |
| Lysing Matrix D tubes | MP Biomedical | 1169130CF |
| Pierce BCA protein Assay kits | Thermo Fisher Scientific | 23225 |
| DAPI | Thermo Fisher Scientific | 62248 |
| Fetal bovine Serum | MERCK Millipore | F7524-500ML - LOT 0001656690 |
| Penicillin Streptomycin | Thermo Fischer Scientific | 15140-122 |
| Carboplatin | ACCORD HEALTHCARE | N/A |
| ATM inhibitor | Selleckchem | KU-55933 |
| ATR inhibitor | Sellechckem | AZD-6738 (ceralazertib) |
| Jet Optimus | Polyplus | POL101000006 |
| Dharmafect 1 | Thermo Scientific Dharmacon | T-2001-03 |
| Sulforodamine B | Merck Millipore | 230162-5 g |
| Phosphatase inhibitor | Sigma-Aldrich | P5726 |
| Protease inhibitor cocktail | Sigma-Aldrich | P8340 |
| ECL plus | Ozyme | OZYB001-5000 |
| DAKO Fluorescent Mounting Medium | Agilent Technologies | S3023 |
| HisPur Ni-NTA Resin | Thermo Fisher Scientific | R901-01 |
| Cytiva Disposable PD-10 Columns | Fisher Scientific | Cytiva 17-0435-01 |
| Micrococcal nucleaese | Biolabs | M0247S |

| Reagent/resource | Reference or source | Identifier or catalog number |
|---|---|---|
| Dynabeads protein G magnetic beads | Thermo Fisher Scientific | 10003D |
| Dibenzazepine (DBZ) | Syncom | SIC-020042 |
| SP141 | SynBio3 platform | N/A |
| Imidazole | Sigma-Aldrich | T62593 |
| Guanidium chloride | Sigma-Aldrich | 50940 |
| iodoacetamide | Sigma-Aldrich | I1149 |
| **Software** | | |
| Qpath 4.1 | https://doi.org/10.1038/s41598-017-17204-5 | |
| GraphPad Prism 10.6.0 | https://www.graphpad.com | |
| ImageJ | ImageJ | |
| biorender | BioRender.com | |
| **Other** | | |
| PHERAstar FSX microplate reader | BMG LABTECH | |
| Xstrahl Xenx preclinical trradiator | Xstrahl | |
| Upright ZEISS AXIO Imager M2 | Apotome2 | |

## Cell culture and proliferation assay

The H358 ($KRAS^{G12C}$ mutation, homozygous p53 deletion), A549 ($KRAS^{G12D}$ mutation, wild-type p53) and 293T cell lines were obtained from the ATCC repository and authenticated. Cells were cultured in RPMI 1640 medium containing 10% fetal calf serum (FCS) and 10% antibiotics, in a humidified atmosphere of 5% $CO_2$ at 37 °C. They were tested regularly for mycoplasma contamination. Cycloheximide (Sigma) was added at 50 μg/ml at the indicated time points. Carboplatin (ACCORD Healthcare, France) was used at 100 μM for 48 h. KU55933 (ATM inhibitor) and Ceralasertib (ATR inhibitor) (Selleckchem) were used at 10 μM and at 5 μM, respectively, and added 2 h before Carboplatin treatment. Irinotecan (Fresenius Kabi) was used at 25 μM for 48 h. For γ-irradiation, cells were exposed to a single dose of 4 grays (Gy), at a dose rate of 2.7 Gy/min using a Xstrahl XenX preclinical irradiator. The H358 Notch1-DeltaEx cell line was generated by transduction of H358 cells with lentiviral NOTCH1-DeltaEx particles. For lentivirus production, 293T cells were co-transfected using JetPEI (Polyplus) with the NOTCH1-DeltaEx plasmid, psPAX2, and pVSV-G vectors. Conditioned medium containing viral particles was collected three days after transfection, cleared by centrifugation, supplemented with 5 μg/ml polybrene, and added to H358 cells at 37 °C, 5% $CO_2$, for 2 days. Afterwards, cells were selected in 1 μg/ml puromycin (Sigma) for 1 week. This new cell line and the H358 parental line were used for proliferation assays in which 8000 cells were plated in a 96-well plate with 50 μl of RPMI 1640 containing 10% FCS. Carboplatin was added at the indicated concentrations for 3 days. Then, plates were rinsed three times with 1× PBS, fixed with 10% ice-cold trichloroacetic acid (Sigma) for 10 min, rinsed

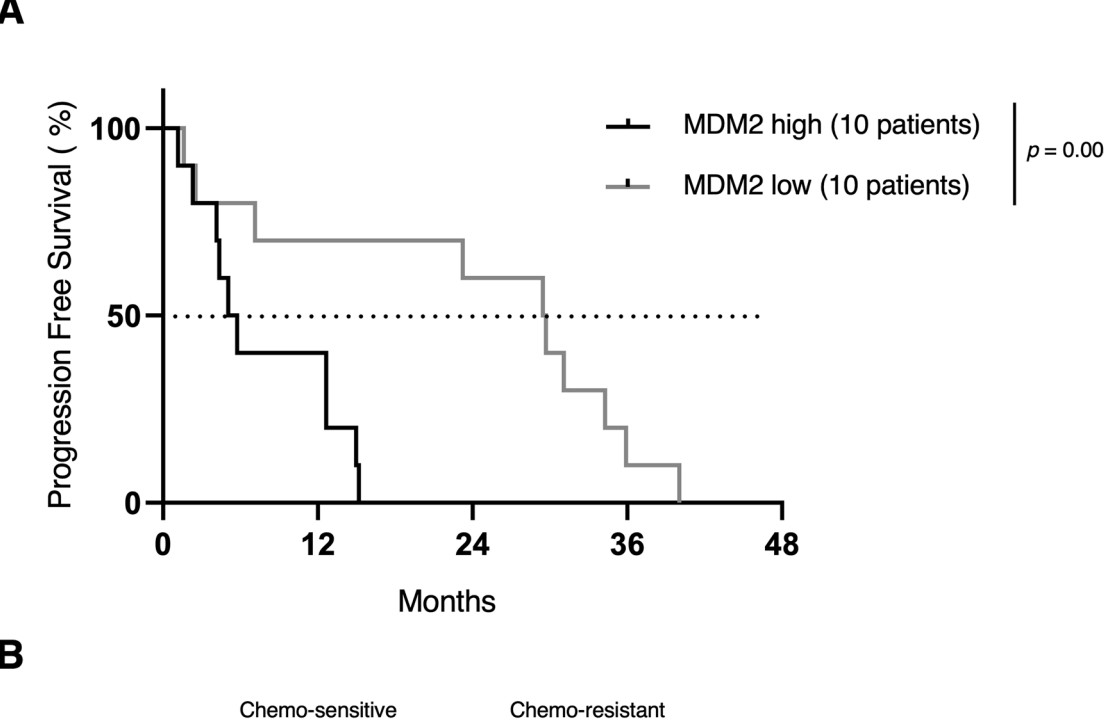

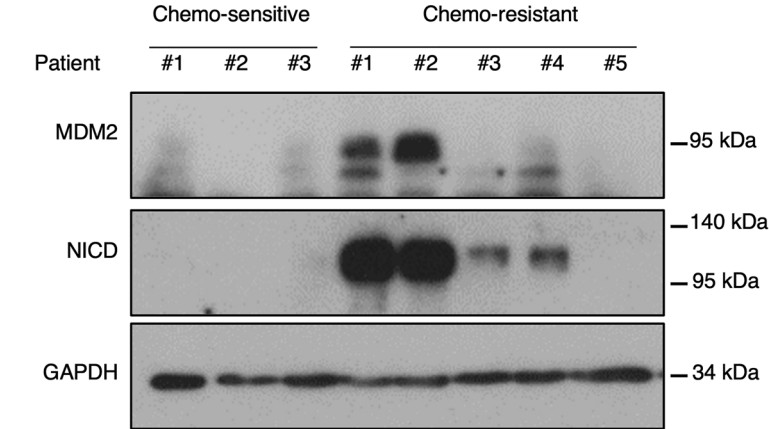

**Figure 6. MDM2 expression correlates with poor progression-free survival in patients with NSCLC treated with platinum-based chemotherapy.**

(A) Progression-free survival of patients with NSCLC treated with platinum-based chemotherapy ($n = 41$) in function of MDM2 expression in the tumor assessed by IHC: highest quartile ($n = 10$) versus lowest quartile ($n = 10$) of MDM2 expression. Survival curves were compared with the log rank (Mantel–Cox) test. (B) Immunoblotting of the indicated proteins in tumor samples from three chemosensitive and five chemo-resistant patients with NSCLC treated with neoadjuvant platinum-based chemotherapy before surgery. Source data are available online for this figure.

three times with 1× PBS, and then stained with 0.04% sulforho-damine dissolved in 1% acetic acid (Sigma) for 30 min. Then, wells were rinsed again three times with 1% acetic acid and the stain was solubilized with 10 mM Tris pH 10.5 for 30 min. The absorbance was read at 540 nM with a PHERAstar FSX microplate reader (BMG LABTECH). Data are presented as fold-change versus untreated cells.

**Protein extraction and western blotting**

Cell extracts were prepared by incubating cells in lysis buffer [50 mM Tris–HCl buffer, pH 8.0, supplemented with 150 mM NaCl, 5 mM EDTA, 0.5% DOCNa, 1% SDS, 1% TritonX-100,

phosphatase and protease inhibitor cocktail (Sigma, USA, #P5726 and #P8340)] on ice for 30 min. For tumor protein extraction, samples were homogenized in lysis buffer in Lysing Matrix D tubes (MP Biomedicals), then incubated on ice for 15 min and centrifuged at 15,000 rpm for 20 min. Protein concentrations were measured with the Pierce TM BCA Protein Assay Kit (Thermo-Fisher Scientific, USA). Proteins were separated by SDS-PAGE, transferred to PVDF membranes, and analyzed using the following antibodies against: NICD (#4147, 1:500), phosphorylated ATM (#13050, 1:1000), ATM (#2873S, 1:1000), phosphorylated ATR (#2853, 1:1000), ATR (#2790S, 1:1000), JAG1 (70109S, 1:1000), JAG2 (2210T, 1:1000), DLL3 (78110S 1:1000), DLL4 (96406S, 1:1000), full length NOTCH1 (3608S, 1:1000), GAPDH (2118S,

1:5000), and H3 (59715S, 1:5000) (all from Cell Signaling Technology, Inc) and also DLL1 (QC16739-41705, Antiva Biosciences, 1:1000), MDM2 (#4B2C1.11 and #4B11 Merck Millipore, 1:2000), TUBULIN (#T9026, Sigma, 1:10,000), poly-histidine (H1029 Sigma, 1:1000), p53 (sc-126 Santa Cruz Technology, 1:1000), TBP (TFIID sc-56794, Santa Cruz Technology, 1:500), HSC70 (sc-7298 Santa Cruz Technology, 1:1000), and HEY1 (#19929 Proteintech 1:1000). Secondary antibodies were horseradish peroxidase-linked anti-rabbit (#7077, Cell Signaling Technology, 1:10,000 dilution), or anti-mouse (#7076, Cell Signaling Technology, 1:10,000 dilution) IgG. Antibody binding was detected by chemiluminescence using the ECL detection system (GE Healthcare) or ECL Plus (for NICD) (GE Healthcare). Each membrane is cut to allow incubation of multiple antibodies on the same Western blot. Only membranes incubated with ATM/P-ATM or ATR/P-ATR were stripped between these two antibodies. Western blot quantifications were done with the ImageJ software.

## Generation of NOTCH1-DeltaEx and cell transfection

The NOTCH1-DeltaEx plasmid was constructed by adding the NOTCH transmembrane domain into a NICD plasmid, kindly provided by Dr Bijan Sobhian (Yatim et al, 2012). The whole sequence was cloned into the Gateway entry vector pDONR221 using specific primers containing the AgeI and MluI restriction sites combined with attB-specific sites. Using the Gateway system (Invitrogen), the NOTCH1-DeltaEx construct was then cloned into the Gateway destination vector pCLX-R4-DEST-R2, a kind gift from Dr. Patrick Salmon (Addgene plasmid # 45956).

The wild-type MDM2, MDM2$^{C464A}$, MDM2$^{395A}$, and His6-tagged ubiquitin constructs (wild-type and K48R) were provided by Dr Laetitia Linares, author of this study. The wild-type MDM2 and MDM2$^{C464A}$ constructs (Riscal et al, 2016) and the His6-tagged ubiquitin constructs (Treier et al, 1994) had been previously described. Then, 2µg of each plasmid was transfected in 293T cells with JEToptimus (POL101000006, Polyplus) according to the manufacturer's instructions. After 48 h, cells were collected to perform chromatin fractionation, immunoprecipation, or ubiquitination assays.

For *MDM2* silencing, the siGENOME SMARTpool against *MDM2* (L-003279-00-0005) was obtained from Thermo Scientific Dharmacon RNAi Technologies and transfected in H358 and A549 cells according to the manufacturer's instructions. After 6 h, cells were incubated with Carboplatin for 48 h.

## Immunofluorescence

Cells were grown on 12 mm glass coverslips and treated as described. Then, cells were washed in PBS, fixed in 2% formaldehyde at room temperature for 15 min, and permeabilized in 0.5% Triton X-100 in PBS for 10 min. Cells were incubated in blocking buffer for 1 h, followed by primary antibodies against γH2AX (the same antibody used for western blotting) and 53BP1 (1:1000, MAB3802 Merck). Cells were washed and incubated with secondary anti-rabbit-Alexa Fluor 488 (1:500, Thermo Fisher Scientific) and anti-mouse-Cy3 (1:500, Thermo Fisher Scientific) antibodies at room temperature for 1 h. After washing, cells were incubated with DAPI (1 µg/ml) and mounted onto glass slides

using DAKO Fluorescent Mounting Medium (Agilent Technologies S3023). Images were acquired using an Upright ZEISS AXIO Imager M2 with Apotome 2.

## Chromatin fractionation

Chromatin fractionation was performed as previously described (Smits et al, 2006). $3 \times 10^6$ cells were washed in PBS and resuspended in 200 µl of solution A (10 mM HEPES pH 7.9, 10 mM KCl, 1.5 mM MgCl$_2$, 0.34 M sucrose, 10% glycerol, 1 mM DTT, protease and phosphatase inhibitors). Triton X-100 was added to a final concentration of 0.1%, cells were incubated on ice for 5 min, and the cytoplasmic (S1) and nuclear fractions (P1) were harvested by centrifugation at $1300 \times g$ for 4 min. Isolated nuclei were washed in solution A, lysed in 150 µl of solution B (3 mM EDTA, 0.2 mM EGTA, 1 mM DTT, protease and phosphatase inhibitors), and incubated on ice for 10 min. The soluble nuclear (S2) and chromatin fractions were harvested by centrifugation at $1700 \times g$ for 4 min. Isolated chromatin (P2) was then washed in solution B, spun down at $10,000 \times g$, and resuspended in 150 µl 1× Laemmli buffer.

## Ubiquitination analysis

293T cells were transfected with the indicated plasmids and recovered by trypsinization; 10% of all harvested cells was used to prepare the total cell extract for western blot analysis, as described above. The remaining cells were resuspended in 10 ml guanidine denaturing buffer (6 M guanidine chloride, 20 mM Tris, 0.01 M imidazole, 0.5 mM DTT, 0.1% iodoacetamide, 0.5% Triton X-100) and incubated at 4 °C under agitation for 1 h. Extracts were incubated with 50 µl of Ni-NTA Agarose beads/sample (HisPur™ Ni-NTA Resin, ThermoScientific) at 4 °C for 2 h under agitation to promote histidine binding to the nickel beads. Then, the bead/protein complexes were gravity-filtered on a sintered column (Fisher Scientific, Cytiva 17-0435-01), washed in 5 ml guanidine denaturing buffer followed by washing first with 20 mM imidazole buffer (2× PBS, 20 mM imidazole, 0.25% Triton X-100) and then with 50 mM imidazole buffer (2× PBS, 50 mM imidazole, 0.25% Triton X-100). Beads were recovered in 50 µl of 4× Laemmli buffer, denatured at 95 °C for 10 min and analyzed by immunoblotting.

## Immunoprecipitation

Extracts were prepared as described in the Chromatin fractionation method, but only the isolated chromatin was used. Chromatin was resuspended in micrococcal nuclease buffer, incubated with micrococcal nuclease (M0247S, Biolabs) at 37 °C for 10 min, and resuspended in the same volume of lysis buffer (20 mM Tris pH 7.5, 150 mM NaCl, 5% glycerol, 0.5% NP40, 0.2 mM EDTA). For immunoprecipitation, extracts were diluted three times in dilution buffer (20 mM Tris pH 7.5, 100 mM NaCl, 0.2 mM EDTA) and incubated with 5µl of anti-NOTCH1 antibody (Ab27526, Abcam) at 4 °C, with agitation, overnight. This was followed by incubation with 50 µl of Dynabeads™ Protein G magnetic beads (Thermo Fisher Scientific) for 1 h to bind the immunoprecipitated complexes to the beads. After several washes in wash buffer (20 mM Tris pH 7.5, 150 mM NaCl, 0.25% NP40, 0.2 mM EDTA), the bead-antibody-protein complexes were resuspended in 50 µl of 4×

Laemmli buffer and denatured at 95 °C for 10 min. The supernatants containing the immunoprecipitated proteins were separated on SDS-PAGE gels and analyzed by immunoblotting as described above.

## Mice

All in vivo experiments were performed conducted in compliance with the French regulations and ethical guidelines for experimental animal studies in an accredited establishment (Agreement No. E34-172-27). Immunodeficient mice are housed in a SOPF status area where the cages are positioned in a ventilated rack. By using this type of rack, each cage can be considered a microbiological unit. Health checks are conducted according to FELASA recommendations, alternating between the analysis of ventilation unit filters and sentinel mice. The PDX TP57 was generated in Luis Paz-Ares' laboratory (one of the study authors) at the Instituto de Biomedicina de Sevilla (IBIS). The PDX was from a NSCLC sample that harbored KRAS (G12A) and p53 (P151R) mutations and displayed intrinsic resistance to Carboplatin (Ferrer et al, 2018). A 0.5 mm³ piece was implanted (ethics approval: 2021092310364690 #33183 v4) into the right flank of 6-week-old athymic nude female mice (Crl:NU(NCr)-Foxn1<nu >) from Charles River. When tumor volume reached 160 mm³, mice were randomized in six groups (9 mice for the vehicle, SP141+Carboplatin and SP141+ DBZ+Carboplatin groups, and 8 mice for the Carboplatin, DBZ +Carboplatin and SP141 + DBZ groups) and drug treatments were started. Dibenzazepine (DBZ), obtained from Syncom (#SIC-020042, The Netherlands), was administered at 2.2 mg/kg/day 5 days per week (Monday–Friday) by intraperitoneal (ip) injection as before (Bousquet Mur et al, 2020). Carboplatin (10 mg/ml, ACCORD Healthcare, France) was administered at 50 mg/kg/day 1 day per week (Wednesday) by ip, as already described (Oliver et al, 2010). SP141 was synthesized by the SynBio3 platform (Gilles Subra's team, Montpellier) and was administered at 30 mg/kg/day 5 days per week intraperitoneally as before (Cisse et al, 2020; Wang et al, 2014). Tumor growth was monitored twice per week using a caliper and mice were euthanized when tumors reached 1500 mm³. The health status of mice was monitored by body weight measurement twice per week and by daily clinical examination. In case of weight loss >20%, gel diet was provided. If after 3–4 days, health was not improved, the concerned mice were euthanized.

For the acute experiment, when the tumor reached 400 mm³, mice were treated with SP141 and DBZ for 4 days (Monday to Thursday) and with Carboplatin on Wednesday. Six hours after the last treatment, mice were sacrificed and tumors were collected: half tumor was snap-frozen and the other half was fixed in 4% paraformaldehyde for immunoblotting and immunohistochemistry (IHC) analysis, respectively.

Animal procedures were performed according to protocols approved by the French national committee of animal care.

## Immunohistochemistry

PDX tumors were fixed, embedded in paraffin and stained with hematoxylin and eosin or used for IHC. IHC of HES1, Ki67, CC3 and γ-H2AX was performed by the RHEM platform as previously described (Bousquet Mur et al, 2020; Cisse et al, 2020). For patient samples, immunochemistry was done by ICM Translational

Research Unit (N° CORT: ICM-CORT-2022-15) with anti-MDM-2 (clone IF2, # MABE340, Millipore, 1:500 dilution) antibody.

For each tumor, four ×10 magnification fields were scored using the Qpath software, providing a total of 12 measures per treated group. Protein expression was evaluated according to the staining intensity and the percentage of positive cells using the H-score (Hirsch et al, 2003). Staining intensity was scored as negative (0), weak (1), moderate (2), or strong (3) and the percentage of positive cells was reported for each staining intensity. The H-score ranged from 0 to 300 and was calculated by taking into account the percentage of positive tumor cells and the staining intensity (Hirsch et al, 2003).

## Patients and ethical considerations

All the experiments are conformed to the principles from the WMA Declaration of Helsinki and the Department of Health and Human Services Belmont Report. For the PFS analysis (Fig. 6A), surgical primary lung cancer samples were obtained from the University Clinic of Navarra (CUN), Spain. Samples were collected at the CUN, after obtaining informed consent from each patient, according to the National and International ethical regulations and the approval of the University of Navarra Research Ethics Committee (number 092.2012.mod1). Patients were all Caucasian, 30 males and 11 females, aged between 45 and 76 years old and were diagnosed between 2001 and 2021 and meeting the following inclusion criteria: diagnosis of NSCLC, surgical resection of the primary lung tumor, availability of clinical data and chemotherapy treatments that included platinum alone or in combination. Exclusion criteria: presence of another primary tumor in the 5 years before surgery, excluding non-melanoma skin tumor.

PFS was defined as the period between the diagnosis and the first relapse. The Kaplan–Meier method was used for the survival analysis and the log rank test to compare groups. The Cox proportional hazards model was used to determine the hazard ratios with the IBM SPSS Statistics 25 software.

For the western blot analysis (Fig. 6B), primary NSCLC tumors were obtained from Hôpital Arnaud de Villeneuve, Montpellier, France, after obtaining informed consent from each patient, according to the National and International ethical regulations and the approval of the "comité de protection des personnes (CPP) sud méditerranée" (number RO-2016/33). Patients were all Caucasian, 1 male and 7 females, aged between 57 and 75 years old. All patients had received at least three cycles of platinum-based chemotherapy. Patients were classified as responders or non-responders, based on the absence and/or persistence of the initial lymph node metastases, respectively.

## Graphics

The cartoons for the "Synopsis" were created with BioRender.com: https://BioRender.com/4ogrgq5.

## Statistical analysis

The investigators were not blinded to the analyses.

Unless otherwise specified, data are presented as means ± SEM. One-way analysis of variance (ANOVA) followed by Tukey's post hoc test was performed to assess the significance of expression

## The paper explained

### Problem

Platinum chemotherapy plays a key role in the clinical management of patients with non-small cell lung cancer (NSCLC), and is widely used as a monotherapy or in combination across various treatment lines. Although most patients initially respond, resistance invariably develops over time, and identifying the basis of this resistance remains a significant challenge.

### Results

We revealed a DNA damage-ATM-MDM2-NICD axis that is activated by platinum treatment. NSCLC patient-derived xenografts with intrinsic platinum resistance regained sensitivity to platinum when NICD generation was hindered. Our preclinical findings were further confirmed by the observation that NSCLC patients with high MDM2 expression respond worst to platinum.

### Impact

Our work highlights the potential of targeting the generation of NICD as a strategy for counteracting platinum resistance in this subset of lung cancer patients.

levels in IHC. For tumor growth, a repeated measures two-way ANOVA followed by Tukey's multiple comparison test was used. Kaplan–Meier survival curves were analyzed with the log rank test. $*P \leq 0.05$; $**P \leq 0.01$; $***P \leq 0.001$, $****P \leq 0.0001$. To assess differences between protein degradation curves, a modified Chi-square-based method was used (Hristova and Wimley, 2023). Sample sizes for comparisons between groups followed Mead's recommendations. In particular, we calculate the degrees of freedom of the error component (E) by compiling the total number of samples analyzed (N) for a given comparison minus the number of groups or treatments (T) (cell type, treatments, etc.) and as recommended it was between 10 and 20 or sometimes higher.

## Data availability

No data required deposition in a public database.

The source data of this paper are collected in the following database record: biostudies:S-SCDT-10_1038-S44321-025-00354-9.

## Peer review information

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

## Acknowledgements

We thank Eric Chevet and David Santamaría for helpful discussion and critical reading of the manuscript. Elisabetta Andermarcher professionally edited the manuscript. We thank the IRCM's animal facility unit members (BioCampus RAM-PEFO, IRCM, Montpellier, France) for animal care and for in vivo studies. We acknowledge the "Réseau d'Histologie Expérimentale de Montpellier" for processing our animal tissues, histology techniques and expertise. We acknowledge the ICM/IRCM experimental irradiation facility supported by SIRIC, granted by "INCa-DGOS-INSERM-ITMO Cancer_18004", for the irradiator. SB was supported by a French Ministry of Education and Research fellowship. Work in AM's laboratory is supported by the Institut National du Cancer (INCa_9257, INCa_16696 and INCa_11554) and by SIRIC Montpellier (INCa_Inserm_DGOS_12553). This work was supported by the COALA network (Cure Oncogene-Addicted Lung Adenocarcinoma), LABREXCMP24-001 –INCa_18791. This project received also support from Ligue Nationale contre le cancer (Comité de l'hérault, R20028FF) and from Cancéropôle GSO

(R20056FF). The lab is additionally funded with the TAONAS-LUAD (101058055) from HORIZON-EIC-2021-TRANSITIONOPEN-01. The funders had no role in the study design, data collection and analysis, decision to publish, or preparation of the manuscript.

## Author contributions

**Sara Bernardo**: Data curation; Methodology; Writing—original draft. **Lisa Brunet**: Data curation; Writing—review and editing. **Quentin Dominique Thomas**: Data curation; Writing—review and editing. **David Bracquemond**: Data curation. **Céline Bouclier**: Data curation. **Marie Colomb**: Data curation. **Maicol Mancini**: Data curation; Formal analysis. **Eric Fabbrizio**: Data curation; Formal analysis. **Alba Santos**: Data curation. **Sylvia-Fenosoa Rasamizafy**: Data curation. **Amina-Milissa Maacha**: Data curation. **Anais Giry**: Data curation. **Emilie Bousquet-Mur**: Data curation. **Laura Papon**: Data curation. **Marion Goussard**: Data curation. **Christophe Fremin**: Data curation. **Andrea Pasquier**: Data curation. **María Rodríguez**: Data curation. **Camille Travert**: Conceptualization; Data curation. **Jean-Louis Pujol**: Conceptualization; Resources. **Laetitia K Linares**: Conceptualization; Resources; Data curation. **Lisa Heron-Milhavet**: Conceptualization; Data curation; Supervision. **Alexandre Djiane**: Conceptualization; Supervision. **Irene Ferrer**: Conceptualization; Supervision. **Luis Paz-Ares**: Conceptualization; Supervision. **Xavier Quantin**: Conceptualization; Supervision; Writing—review and editing. **Luis M Montuenga**: Conceptualization; Data curation; Supervision; Funding acquisition; Validation; Investigation; Methodology; Writing—original draft; Writing—review and editing. **Hélène Tourriere**: Conceptualization; Data curation; Supervision; Funding acquisition; Validation; Investigation; Methodology; Writing—original draft; Writing—review and editing. **Antonio Maraver**: Conceptualization; Data curation; Formal analysis; Supervision; Funding acquisition; Validation; Investigation; Methodology; Writing—original draft; Writing—review and editing.

Source data underlying figure panels in this paper may have individual authorship assigned. Where available, figure panel/source data authorship is listed in the following database record: biostudies:S-SCDT-10_1038-S44321-025-00354-9.

## Disclosure and competing interests statement

The authors declare no competing interests.

# Expanded View Figures

**Figure EV1.   DNA damage induces NICD stabilization.**

(**A**) Western blotting of the indicated proteins in A549 cells exposed to 100 μM Carboplatin ($n = 2$), 25 μM irinotecan ($n = 2$), or γ-irradiation (4 Gy) ($n = 1$) for the indicated time. P-ATM, phosphorylated (activated) ATM. (**B**) Immunofluorescence analysis of γH2AX (in red) and 53BP1 (in green) in A549 cells after 24 or 48 h of incubation with 100 μM Carboplatin ($n = 1$). Nuclei were stained with DAPI (in blue). Scale bar: 10 μm. (**C**) Western blotting of the indicated proteins after cellular fractionation of A549 cells incubated ($+$) or not ($-$) with 100 μM Carboplatin for 48 h ($n = 2$). S1, cytoplasmic fraction; S2, nucleoplasmic fraction; P2, chromatin fraction; TUBULIN, loading control for S1; TBP, loading control for P2. (**D**) Western blotting of the indicated proteins in A549 cells incubated with 100 μM Carboplatin for the indicated time ($n = 1$). (**E**) Western blotting of the indicated proteins in A549 cells incubated or not with 100 μM Carboplatin for 48 h and with 50 μg/μl Cycloheximide (CHX) for the indicated time before the end of Carboplatin treatment ($n = 2$). Upper panel: Schematic representation of the experimental design. In the legend, $n$ indicates the number of biological replicates. Source data are available online for this figure.

▶

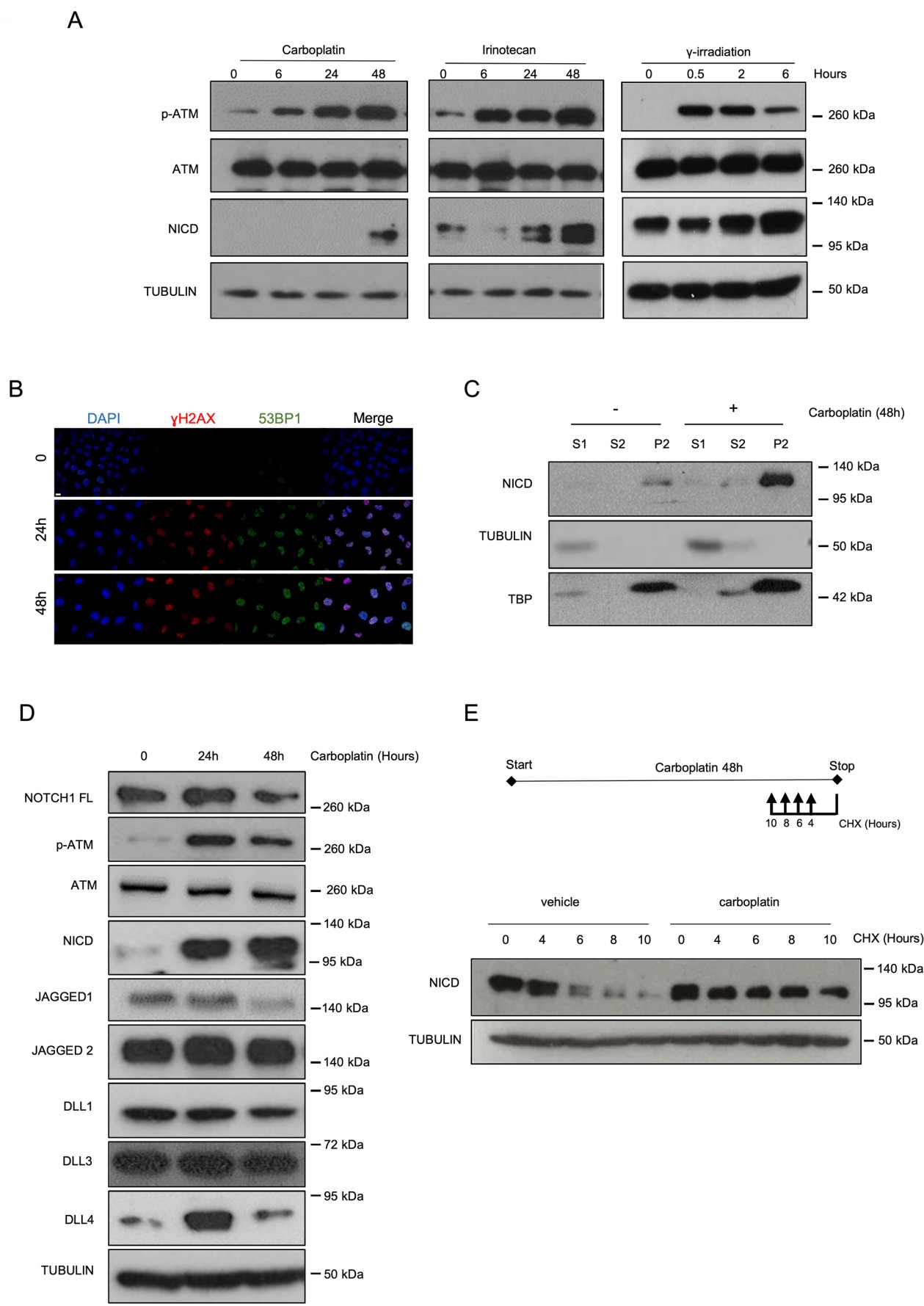

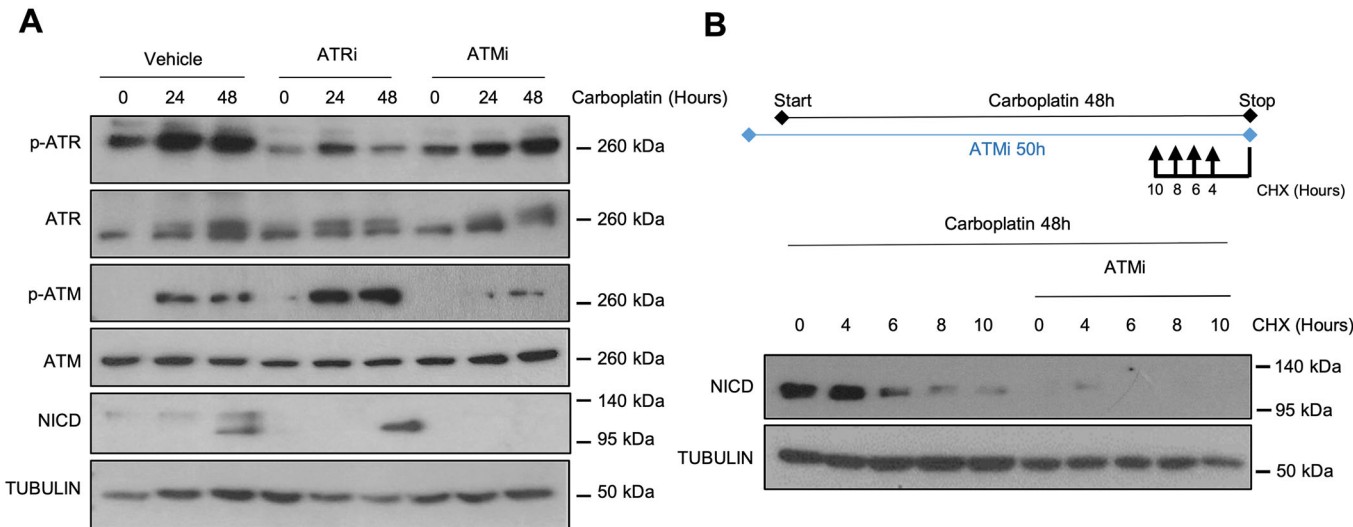

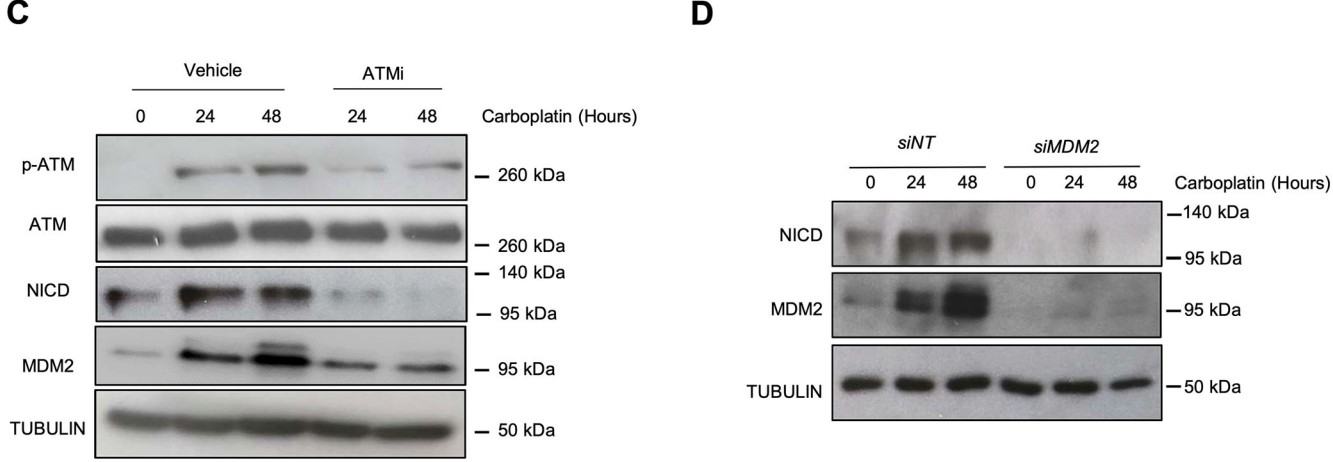

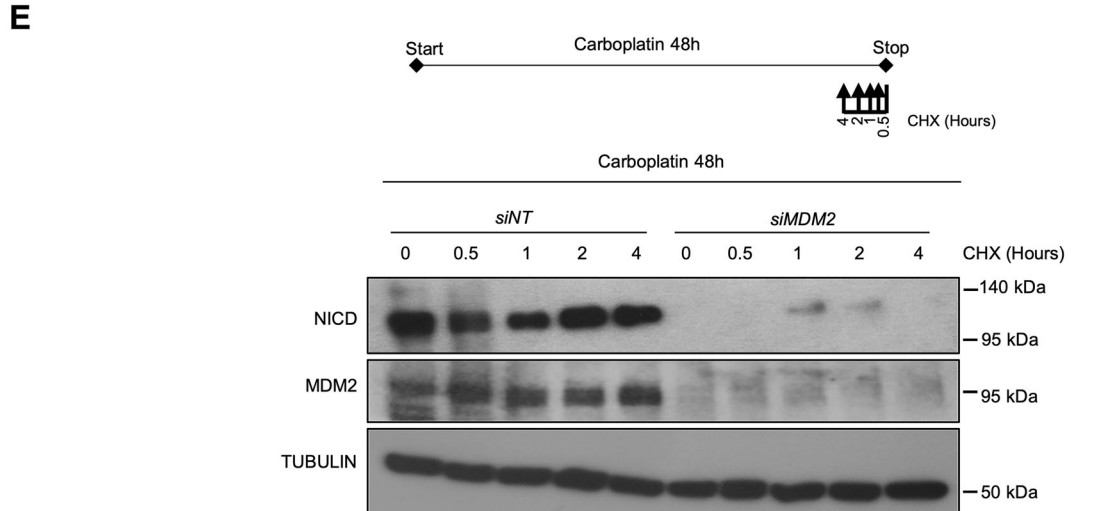

◀ **Figure EV2.** **NICD stabilization is ATM- and MDM2-dependent.**

(A) A549 cells were incubated with 100 μM Carboplatin and with/without 10 μM KU-55933 (ATM inhibitor) or Ceralasertib (ATR inhibitor) for the indicated time followed by immunoblotting to detect the expression of the indicated proteins ($n = 2$). p, phosphorylated. (B) A549 cells were incubated with 100 μM Carboplatin for 48 h and with or without 10 μM KU-55933 (ATM inhibitor; ATMi). 50 μg/μl Cycloheximide (CHX) was added for the indicated time before the end of Carboplatin treatment ($n = 2$). Upper panel: Schematic representation of the experimental design. (C) Western blotting of the indicated proteins in A549 cells incubated with 100 μM Carboplatin for the indicated time, with or without 10 μM KU-55933 (ATM inhibitor; ATMi) ($n = 2$). (D) Western blotting of the indicated proteins in A549 cells transfected with a non-targeting *siRNA* (*siNT*) or with a *siRNA* against MDM2 (*siMDM2*) for 6 h, followed by addition of 100 μM Carboplatin for the indicated time ($n = 3$). (E) At 6 h post-transfection, A549 cells transfected with *siMDM2* or *siNT* were incubated with 100 μM Carboplatin for 48 h and with 50 μg/μl of Cycloheximide (CHX) for the indicated time before the end of Carboplatin treatment ($n = 2$). Upper panel: schematic representation of the experimental design. In the legend, *n* indicates the number of biological replicates. Source data are available online for this figure.

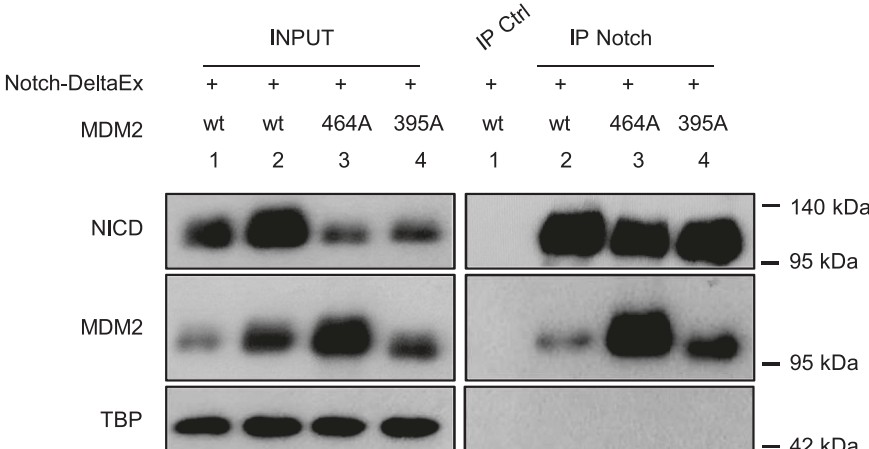

**Figure EV3.   NICD and MDM2 interaction.**

293T cells were transfected with NOTCH1-DeltaEx, empty vector (pcDNA), MDM2 WT, MDM2 464 A or MDM2 395 A for 48 h. Chromatin extracts were used for NOTCH1 immunoprecipitation and the levels of the indicated proteins were measured by western blotting. INPUT: total cell extracts. IP ctrl: immunoprecipitation control with an unspecific mouse antibody, IP Notch: immunoprecipitation with an anti-NOTCH1 antibody. TBP was used as loading control in INPUT. Source data are available online for this figure.

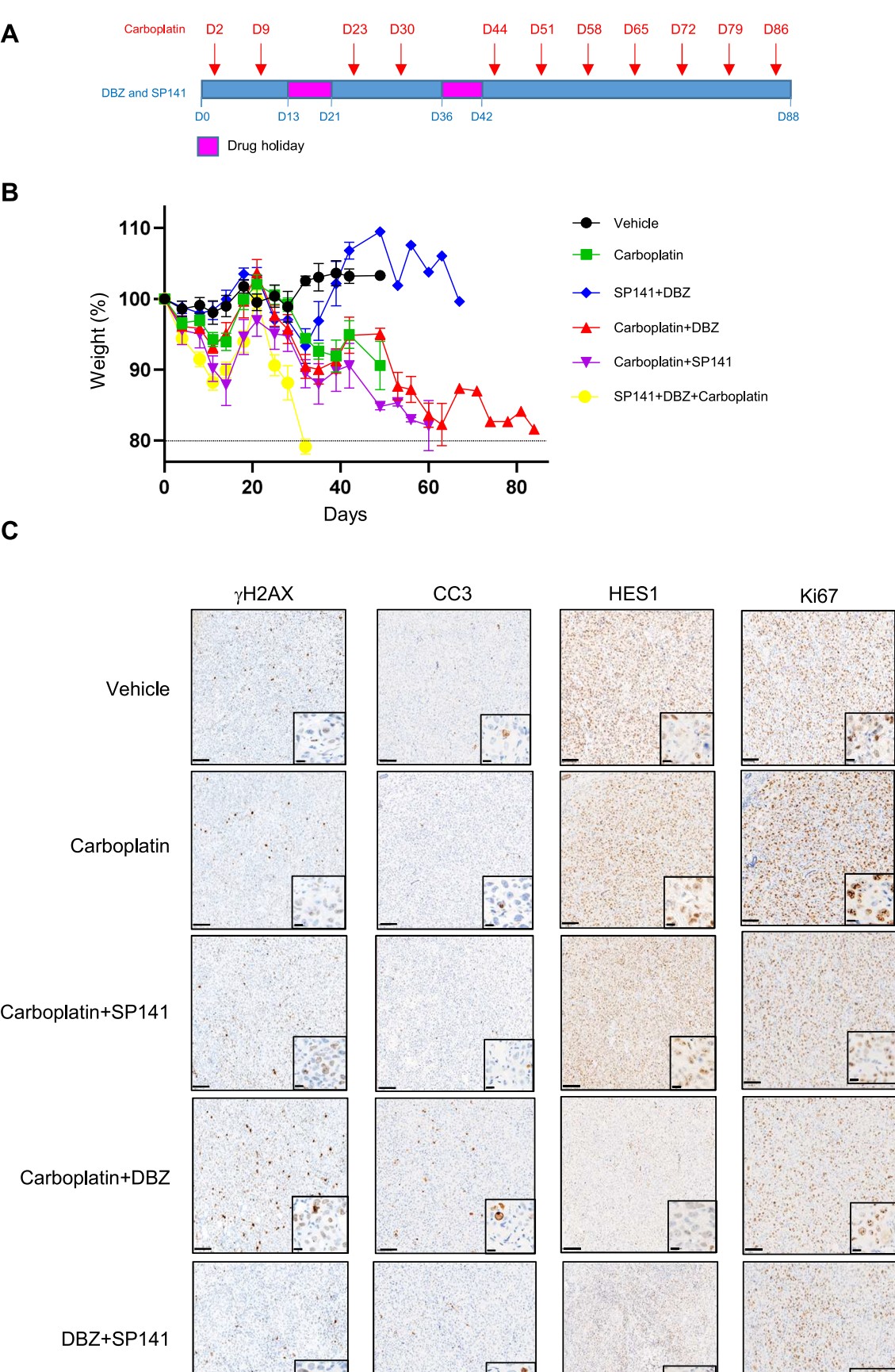

◄ **Figure EV4. In vivo treatments.**

(A) Schedule of the in vivo treatments. Carboplatin (50 mg/kg/day) was administered once per week (red arrow on top). SP141 (30 mg/kg/day) and DBZ (2.2 mg/kg/day) were administered 5 days per week. Pink color depicts the drug holidays. (B) Weight change curves in the different treatment groups as in Fig. 5B throughout the treatment period. Weights were normalized at each time point to day 0 and then represented as an average of all mice in each group. (C) Representative images of the IHC performed in PDX tumor samples from the indicated groups (data quantification in Figs. 5D and EV5A). Scale bars: 100 μm; Inset scale bars: 20 μm. Source data are available online for this figure.

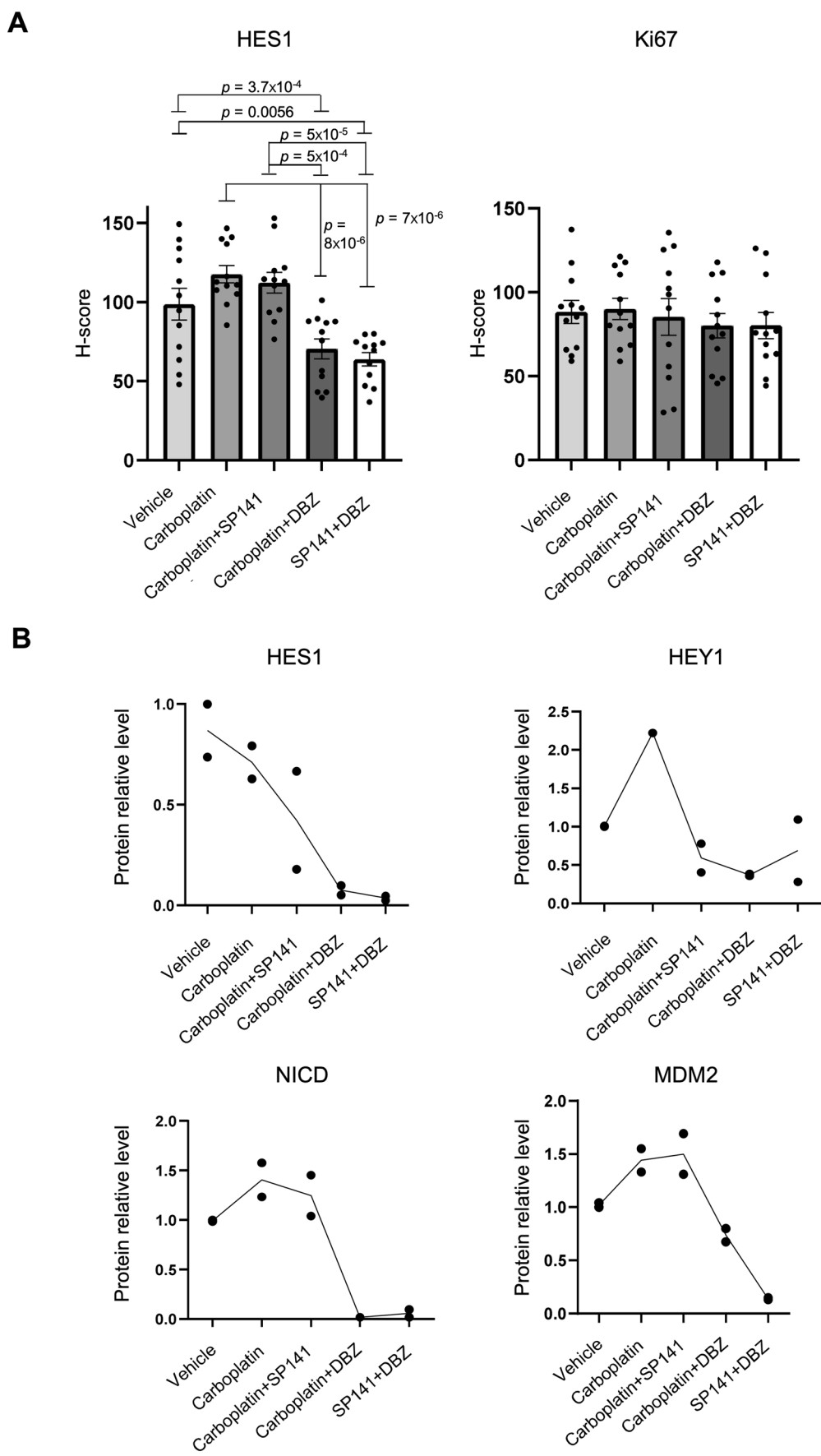

◄ **Figure EV5.  IHC and western blot quantifications.**

(**A**) IHC analysis of HES1 and Ki67 expression in PDX (TP57) xenografts from nude mice treated with vehicle, Carboplatin alone, Carboplatin + SP141, Carboplatin + DBZ, and SP141 + DBZ ($n = 12$ measures from 3 different tumors for all groups). Data are the mean ± SEM and they correspond to the analysis of 4 fields (original magnification, ×20) per tumour. $P$ values were obtained by 1-way ANOVA followed by Tukey's multiple comparison test). (**B**) Quantification of the western blots in Fig. 5E. The level of each protein was normalized to GAPDH in the same condition. Then, protein signal in each lane was normalized to that in lane 1 (vehicle) set at 1. In the legend, $n$ indicates the number of biological replicates. Source data are available online for this figure.

