## [Peer Review File · EMBO Molecular Medicine]

NOTCH1 intracellular domain stabilization by MDM2 plays a major role in NSCLC response to platinum

Sara Bernardo, Lisa Brunet, Quentin Thomas, David Bracquemond, Celine Bouclier, Marie Colomb, Maicol Mancini, Eric Fabbri, Alba Santos, Sylvia Rasamizafy, Amina Maacha, Anais Giry, emilie Bousquet-Mur, Laura Papon, Marion Goussard, Christophe Fremin, Andrea Pasquier, Maria Rodriguez, Camille Traver, Jean-Louis Pujol, Laetitia Linares, Lisa Héron-Milhavet, Alexandre Djiane, Irene Ferrer, Luis Paz-Ares, Xavier Quantin, Luis Montuenga, Helene Tourriere, and Antonio Maraver

Corresponding author(s): Antonio Maraver (antonio.maraver@inserm.fr), Helene Tourriere (helene.tourriere@inserm.fr)

Review Timeline:

Transfer Date:	25th Jul 24
Editorial Decision:	29th Jul 24
Revision Received:	9th May 25
Editorial Decision:	3rd Jun 25
Revision Received:	24th Oct 25
Editorial Decision:	6th Nov 25
Revision Received:	20th Nov 25
Accepted:	25th Nov 25

Editor: Lise Roth

Transaction Report: This manuscript was transferred to EMBO Molecular Medicine following peer review at Review Commons.

**Review
COMMONS**

Review #1

1. Evidence, reproducibility and clarity:

Evidence, reproducibility and clarity (Required)

****Summary:****

This manuscript from Maraver and co-authors investigates the putative resistance mechanisms that hinder the efficacy of platinum-based therapies (e.g., carboplatin) against non-small cell lung carcinoma (NSCLC). Using in vitro lung cancer cell lines, shRNA-based knockdown, and exogenous overexpression systems, the research describes a DNA damage-induced resistance mechanism involving the NOTCH signaling pathway and the E3 ligase MDM2. The authors show that carboplatin treatment induces DNA damage and promotes ATM activation, which in turn activates the NOTCH signaling pathway via ubiquitination and stabilization of the Notch Intracellular Domain (NICD). New findings include the MDM2-mediated ubiquitination and stabilization of NICD. Using in vivo NSCLC-PDX models, they demonstrate that combining carboplatin with Notch and MDM2 inhibitors can enhance tumor killing, suggesting that targeting the MDM2/NICD axis in conjunction with carboplatin may be a viable therapeutic alternative. Furthermore, they show that NICD and MDM2 levels are elevated among tumor samples from chemo-resistant patients. Consistent with these findings, high MDM2 levels correlate with poor progression-free survival (PFS) in NSCLC patients.

****Major comments:****

Some of the key conclusions may not be convincing.

1. One significant weakness of the manuscript is the lack of exploration into the underlying mechanism of how MDM2 mediates the stabilization of NICD. While the observation of MDM2-mediated NICD stabilization is intriguing, it is important to provide a more convincing explanation for the reviewers. This could be achieved by offering a detailed molecular mechanism, especially considering that MDM2 typically targets proteins for degradation.

2. Another weakness lies in the unclear role and the underlying mechanism of ATM in the MDM2-mediated NICD stabilization. While the data presented (Fig. 3B, 3C) suggest that carboplatin could elevate MDM2 levels for NICD stabilization, a more precise method to induce MDM2 overexpression specifically for targeting NICD is required. It appears that ATM plays a crucial role in this regulatory process. The following questions must be

addressed: Does ATM induce the phosphorylation of MDM2 for its protein stabilization and/or E3 ligase activity?

3. The combination therapy of carboplatin with MDM2 and NICD inhibitors may lack compelling rationale (see below).

4. In lines 275-276, the authors stated that their preclinical data establish the enhancement of carboplatin's therapeutic effect in NSCLC in vivo through MDM2-NICD axis inhibition. However, it's important to note that this finding remains preliminary at this stage.

****Minor comments:****

1. The observed loss of NICD during ATMi + carboplatin treatment in Figures 2A and 2B raises the question of whether ATM regulates the gene transcription of NOTCH. In addition to the CHX assay conducted in Figures 2C and 2D, quantifying NOTCH mRNA upon ATM inhibition could provide further insights. Alternatively, referencing relevant studies on this topic may strengthen the discussion.

2. In Figures 4A and 4B, the noticeable discrepancy between the exogenous expression of wild-type (WT) MDM2 and catalytically inactive MDM2-464A raises concerns. It is essential to consider if the reduced ubiquitination and stability of NICD might be attributed to varying levels of MDM2-464A in the cells rather than its catalytic inactivity. While p53 ubiquitination was utilized as a control, ensuring comparable levels of MDM2 and MDM2-464A expression could enhance the experimental rigor. Compared to the smear poly-ubiquitination bands observed for MDM2 in Figure 4B, the ubiquitination of NICD appears simpler. What distinguishes the feature of MDM2-mediated NICD ubiquitination? Could it potentially involve mono-ubiquitination?

3. In Figure 5A, the authors need to consider conducting additional NOTCH-associated factors to definitively demonstrate the activation of NOTCH signaling beyond HES1. Alternatively, in Figure 5B, the NICD Western blot could be complemented by detecting HES1 or other NOTCH-associated factors.

4. In Figures 5C and 5D, crucial control groups are missing, specifically mice treated solely with SP141+DBZ, carboplatin+SP141, and SP141+DBZ. It is essential to include these groups to demonstrate that the enhanced tumor killing results from the combination of carboplatin with SP141 and/or DBZ, rather than from SP141 and DBZ alone. Furthermore, in addition to the currently used NSCLC-PDX model harboring the p53 (P151R) mutation, it would be informative to include a NSCLC-PDX model expressing WT p53.

5. Though beyond the current study's scope, in the discussion section, the authors may want to propose or hypothesize on how MDM2-mediated NICD stabilization contributes to carboplatin resistance. This could provide valuable insights for future research directions.

6. In the Western blot results, the total ATM and ATR controls were absent.
7. Authors may choose to include a graphical abstract at the end of their study to visually illustrate the mechanisms they have described.

2. Significance:

Significance (Required)

Advance: The authors aim to present a novel perspective on the resistance mechanisms to platinum compounds in NSCLC therapy. They explore platinum compounds-induced DNA damage, ATM activation, and MDM2-mediated stabilization of the active form of NOTCH (NICD). However, to strengthen their claims, they must provide more conclusive results.

Audience: This manuscript will likely engage oncologists who investigate the chemotherapy-resistant mechanisms of platinum compounds in NSCLC treatment, as well as scientists specializing in NOTCH and MDM2 pathways. However, the manuscript's central claims lack robust support from the available data, and the current approaches employed are not sufficiently thoughtful and rigorous; there is room for improvement."

My expertise is molecular medicine, cancer biology, and epigenetics.

3. How much time do you estimate the authors will need to complete the suggested revisions:

Estimated time to Complete Revisions (Required)

(Decision Recommendation)

Between 3 and 6 months

Yes

Review #2

1. Evidence, reproducibility and clarity:

Evidence, reproducibility and clarity (Required)

In this manuscript, Sara Bernardo et al. investigated the molecular mechanisms underlying the activation of the Notch signaling in response to DNA damage induced by platinum-based chemotherapeutic agents in non-small cell lung cancer (NSCLC). They demonstrated that carboplatin treatment induces DNA double-strand breaks (DSBs) and stabilizes NICD, a process dependent on ATM and mediated by MDM2. In vivo experiments in patient-derived xenografts (PDX) showed that inhibition of NICD and MDM2 enhanced platinum effectiveness. Furthermore, clinical analysis revealed a correlation between MDM2 expression and poor prognosis in NSCLC patients treated with platinum compounds, emphasizing the clinical relevance of the MDM2-NICD axis in platinum resistance.

****Major comments:****

Overall, the authors have conducted experiments that sufficiently elucidate their claims, and the description of the experiments is detailed. However, there is still room for the improvement.

1. The finding that MDM2 promoted NICD stability through non degradative ubiquitination is interesting and in line with a previous study. As it is also known that NICD is regulated by various post-translational modifications, including ubiquitination that promotes NICD degradation. It is unclear what's the potential difference between these two types of ubiquitination. For example, do these two differ in specific ubiquitination sites? Can the authors provide some discussion?

2. Could the overexpression of MDM2 or NICD lead to carboplatin resistance in A549 or H358 cells?

3. The trends observed in the western blot data within the manuscript appear inconsistent. While the authors propose that NICD levels increased upon incubation with carboplatin, the discrepancy arises when considering the NICD levels without cycloheximide (CHX) treatment in Figure 1E, where no significant elevation is observed (Lane 6 vs. Lane 1).

4. The quality of western blots needs to be improved, especially Fig. 1C and S1C, also Figure 3B. Moreover, the NICD western blot sometimes appears as one band and sometimes as two bands. Please provide an explanation. If possible, please quantify the bands in western blots.

5. Please provide a necessary discussion on whether the targeted treatment approach towards the MDM2-NICD axis is applicable to all patients or only to those with high expression of MDM2/NICD.
6. How to interpret the significance of the simultaneous increase in NICD ubiquitination and stability mediated by MDM2? Please provide a relevant discussion.
7. In Figure 5B, please also check the level of MDM2. In Figure 5C, carboplatin appears to have little impact on tumor growth. How to explain the increase of Ki-67 in the carboplatin treatment group in Figure 5A?

****Minor comments:****

1. Please include scale bars in Figure 1B and Supplemental Figure 1B.
2. Figure 5D, the P values of the survival curve should be indicated in the figures.
3. The presentation of survival curve data in Figures 5D and 6A should be consistent.
4. It seems that supplemental figure 2 is missing.
5. Please carefully check the spelling of the entire text, for example, on page 20, line 426 it should be 'western'. Also, please spell out the abbreviations DDR and ATM.
6. The abbreviation for Cleaved caspase 3 should be CC3.

2. Significance:

Significance (Required)

Notch signaling is associated with the occurrence and development of non-small cell lung cancer (NSCLC). Previous study indicates that the expression of Notch protein is significantly higher in NSCLC tissues compared to normal tissues (PMID: 31170211). Additionally, the upregulation of Notch1 is correlated with higher tumor grades, lymph node metastasis, tumor-node-metastasis (TNM) staging, and poor prognosis (PMID: 25996086). Abnormal activation of Notch signaling pathway is frequently observed in chemotherapy-resistant NSCLC, and some studies have aimed to address NSCLC drug resistance via modulating Notch signaling (PMID: 30087852, 38301911). This manuscript firstly proposes that MDM2-mediated stabilization of NICD upon DNA damage plays a major role in NSCLC response to platinum chemotherapy. It further suggests that targeting the MDM2-NICD axis could prove to be an effective therapeutic strategy. Overall, this work unveils a novel mechanism for Notch activation in response to platinum chemotherapy, providing a renewed outlook on overcoming chemotherapy resistance in NSCLC. This manuscript will attract those interested in the mechanisms of chemotherapy resistance and novel treatment approaches.

3. How much time do you estimate the authors will need to complete the suggested revisions:

Estimated time to Complete Revisions (Required)

(Decision Recommendation)

Between 1 and 3 months

Yes

Revision Plan

Manuscript number: RC- 2024-02497

Corresponding author(s): Tourriere, H elene and Maraver, Antonio

1. General Statements [optional]

We sincerely thank the Editors and Reviewers for the time devoted to our manuscript. We found their critiques interesting and very helpful. After careful examination and thanks to a large collaborative effort, we will be able to answer to all the reviewers' comments by adding significantly new experimental data.

We are also encouraged by the positive comments of the Reviewers:

“This manuscript will likely engage oncologists who investigate the chemotherapy-resistant mechanisms of platinum compounds in NSCLC treatment” (Reviewer 1);

“Overall, the authors have conducted experiments that sufficiently elucidate their claims, and the description of the experiments is detailed.”; and “Overall, this work unveils a novel mechanism for Notch activation in response to platinum chemotherapy, providing a renewed outlook on overcoming chemotherapy resistance in NSCLC” (Reviewer 2).

We are also aware that both reviewers agreed that there is room for improvement, and we are sure that upon accomplishment of all proposed experiments both reviewers will be fully satisfied.

Please bear in mind that although it was known that platinum-based chemotherapy induced the Notch pathway in lung cancer cells, the underlying molecular mechanism was largely unknown. Thanks to the molecular dissection we performed in our study, we propose an innovative treatment for patients with lung cancer, the main cause of death by cancer in the world. Hence, we agree with both reviewers that our study will be appealing for a large number of cancer researchers, and we feel it will be also the case for those interested in DNA damage, Notch and MDM2 pathways.

2. Description of the planned revisions

Reviewer #1 (Evidence, reproducibility and clarity (Required)):

Summary:

This manuscript from Maraver and co-authors investigates the putative resistance mechanisms that hinder the efficacy of platinum-based therapies (e.g., carboplatin)

Revision Plan

against non-small cell lung carcinoma (NSCLC). Using in vitro lung cancer cell lines, shRNA-based knockdown, and exogenous overexpression systems, the research describes a DNA damage-induced resistance mechanism involving the NOTCH signaling pathway and the E3 ligase MDM2. The authors show that carboplatin treatment induces DNA damage and promotes ATM activation, which in turn activates the NOTCH signaling pathway via ubiquitination and stabilization of the Notch Intracellular Domain (NICD). New findings include the MDM2-mediated ubiquitination and stabilization of NICD. Using in vivo NSCLC-PDX models, they demonstrate that combining carboplatin with Notch and MDM2 inhibitors can enhance tumor killing, suggesting that targeting the MDM2/NICD axis in conjunction with carboplatin may be a viable therapeutic alternative. Furthermore, they show that NICD and MDM2 levels are elevated among tumor samples from chemo-resistant patients. Consistent with these findings, high MDM2 levels correlate with poor progression-free survival (PFS) in NSCLC patients.

[Authors] We thank this reviewer for her/his fair summary of our work that highlights our new findings.

Major comments:

Some of the key conclusions may not be convincing.

[Authors] We understand the concerns that reviewer might have and we are sure that upon accomplishment of all experiments detailed below, she/he will be convinced that the manuscript will be ready for publication.

1. One significant weakness of the manuscript is the lack of exploration into the underlying mechanism of how MDM2 mediates the stabilization of NICD. While the observation of MDM2-mediated NICD stabilization is intriguing, it is important to provide a more convincing explanation for the reviewers. This could be achieved by offering a detailed molecular mechanism, especially considering that MDM2 typically targets proteins for degradation.

[Authors] After reading this reviewer's comment, we realize we did a poor job discussing better the previous study demonstrating that MDM2 induced ubiquitination on NICD but not for degradative purposes (Pettersson et al., 2013). In particular, they performed it using a mutated form of ubiquitin in lysine 48, i.e., the K48R mutant. Like this, the authors of this seminal study demonstrated that MDM2 was still able to induce ubiquitination in NICD, and hence it was not degradative.

Still, and to confirm that this is the case also upon DNA damage, we will perform experiments using same K48R mutant to formally prove that MDM2 upon DNA damage does not ubiquitinate NICD via lysine 48-linked polymers, and hence it is not degradative.

Revision Plan

Even more, upon discussion with Laetitia Linares, author of our study and long-lasting expert in ubiquitination (for instance see (Riscal et al., 2016) and (Arena et al., 2018)), we will use another ubiquitin mutant in lysine 63. This different type of ubiquitination does not mark proteins for degradation but promote an association of the targeted protein with DNA helping for DNA repair (Liu et al., 2018). Using a ubiquitin mutated in this lysine, i.e., K63R, this type of ubiquitination cannot occur. Taking into account that we observe NICD increase ubiquitination upon DNA damage, the use of K63R will be very informative. Hence, we will repeat experiments of current Figure 3A with the same WT ubiquitin as before, and now also with K48R and K63R mutants. Even more, we will also include mutant forms of ubiquitin which can only form ubiquitin chains on lysine 48 (K48 only) or lysine 63 (K63 only) and we anticipate that in the presence of K48 only mutant, NICD will not be ubiquitinated upon DNA damage, while the use of K63 only mutant will be very useful. All these data will be part of the **new Figure 3A**.

Of note, Dr Linares has all tools required to perform these experiments and hence we will start them soon.

2. Another weakness lies in the unclear role and the underlying mechanism of ATM in the MDM2-mediated NICD stabilization. While the data presented (Fig. 3B, 3C) suggest that carboplatin could elevate MDM2 levels for NICD stabilization, a more precise method to induce MDM2 overexpression specifically for targeting NICD is required. It appears that ATM plays a crucial role in this regulatory process. The following questions must be addressed: Does ATM induce the phosphorylation of MDM2 for its protein stabilization and/or E3 ligase activity?

[Authors] There are several points here.

For the first one, the use of a more precise method to induce MDM2 overexpression, it is exactly what we did in Figure 4A, i.e., ectopic expression of MDM2 to demonstrate that MDM2 is sufficient to increase NICD levels.

For the second one, i.e., the phosphorylation status of MDM2 by ATM in our system, we will perform different experiments. There are up to six proposed residues in MDM2 to be phosphorylated by ATM upon DNA damage: S386, S395, S407, T419, S425, and S429 (Cheng et al., 2011). Among all of them, S395 is the most well-known and again Dr Linares has interesting tools we will use to answer to this specific reviewer's point. We will use an MDM2 mutant harboring an aspartate instead of the serine in this position, i.e., S395D, that mimics the serine 395 phosphorylation induced by ATM upon DNA damage. We will use this mutant together with the WT and 464A MDM2 proteins already used, and if this residue is important in our phenotype, total levels of NICD will be even higher and/or localize more in the nuclei when compared with WT MDM2. All these new data will appear as the **new Figure 4A** and **new Figure 4B**.

Revision Plan

Furthermore, we will also use an antibody that recognizes this phosphorylation site by WB after carboplatin treatment and it will be part of the **new Figure 3B**.

Finally, we will also express WT MDM2 and purify it by immunoprecipitation in different experimental conditions: steady state, upon carboplatin treatment and also in combination of carboplatin and ATM inhibitor, to perform phospho-proteomics analysis upon all these conditions. Of note, and to show the feasibility of this approach, the proteomic platform at Biocampus in Montpellier has experience using this technique (Kassouf et al., 2019).

3. The combination therapy of carboplatin with MDM2 and NICD inhibitors may lack compelling rationale (see below).

[Authors] This is a very important point but we discuss it below, where more information is provided by the reviewer. Still, we anticipate we will perform a new in vivo experiment to answer to this point.

4. In lines 275-276, the authors stated that their preclinical data establish the enhancement of carboplatin's therapeutic effect in NSCLC in vivo through MDM2-NICD axis inhibition. However, it's important to note that this finding remains preliminary at this stage.

[Authors] We consider that our statement is not exaggerated, but we will tone down the message as proposed by the reviewer in the next submission.

Minor comments:

1. The observed loss of NICD during ATMi + carboplatin treatment in Figures 2A and 2B raises the question of whether ATM regulates the gene transcription of NOTCH. In addition to the CHX assay conducted in Figures 2C and 2D, quantifying NOTCH mRNA upon ATM inhibition could provide further insights. Alternatively, referencing relevant studies on this topic may strengthen the discussion.

[Authors] This is an interesting experiment and we will perform it.

2. In Figures 4A and 4B, the noticeable discrepancy between the exogenous expression of wild-type (WT) MDM2 and catalytically inactive MDM2-464A raises concerns. It is essential to consider if the reduced ubiquitination and stability of NICD might be

Revision Plan

attributed to varying levels of MDM2-464A in the cells rather than its catalytic inactivity. While p53 ubiquitination was utilized as a control, ensuring comparable levels of MDM2 and MDM2-464A expression could enhance the experimental rigor. Compared to the smear poly-ubiquitination bands observed for MDM2 in Figure 4B, the ubiquitination of NICD appears simpler. What distinguishes the feature of MDM2-mediated NICD ubiquitination? Could it potentially involve mono-ubiquitination?

[Authors] The point of the reviewer is well taken, and importantly, as mentioned above in main point 2, we will repeat these experiments and will appear as **new Figure 4A** and **new Figure 4B**.

Regarding the type of ubiquitination, as explained in detail in major point 1 to same reviewer, we will fully characterize the type of ubiquitination on NICD induced by DNA damage, and we will confirm that MDM2 is required for this specific ubiquitination in future **new Figure 4C** where we will overexpress the required ubiquitin forms and WT MDM2.

3. In Figure 5A, the authors need to consider conducting additional NOTCH-associated factors to definitively demonstrate the activation of NOTCH signaling beyond HES1. Alternatively, in Figure 5B, the NICD Western blot could be complemented by detecting HES1 or other NOTCH-associated factors.

[Authors] To answer to this particular point, we will test for other downstream targets of Notch as NRARP and it will appear as part of **new Figure 5C**.

4. In Figures 5C and 5D, crucial control groups are missing, specifically mice treated solely with SP141+DBZ, carboplatin+SP141, and SP141+DBZ. It is essential to include these groups to demonstrate that the enhanced tumor killing results from the combination of carboplatin with SP141 and/or DBZ, rather than from SP141 and DBZ alone. Furthermore, in addition to the currently used NSCLC-PDX model harboring the p53 (P151R) mutation, it would be informative to include a NSCLC-PDX model expressing WT p53.

[Authors] This is a crucial point in this rebuttal as mentioned before in major point 3 and we detail it in here.

We did only 3 groups because preliminary data indicated that SP141 in combination with carboplatin was not showing any benefit compared to carboplatin alone while upon combination of carboplatin with Notch inhibition there was only a slight increase in therapeutic carboplatin benefit but otherwise not very potent, and for simplicity we

Revision Plan

preferred to don't show these data. But, after reading this point from Reviewer 1, even if we will propose later only the triple combination for patients, we clearly need to demonstrate that the other combinations are not potent enough or not at all.

The reviewer asked to include: "SP141+DBZ, carboplatin+SP141, and SP141+DBZ". We imagine that she/he meant: SP141+DBZ, carboplatin+SP141, and carboplatin +DBZ, that together with the vehicle, carboplatin and carboplatin+SP141+DBZ makes 6 groups of treatments. Putting together the 8 mice devoted for tumor growth and survival, plus 4 mice for the acute treatment for IHC and WB purposes (for current Figures 5A and 5B) makes a total of 72, that is a substantial number of mice. Of note, since we performed the in vivo experiment presented in the current manuscript, a new Notch inhibitor called nirogacestat, appear in the market being the first in class Notch inhibitor to treat solid cancer patients (desmoid tumors) after demonstrating a significant therapeutic effect in clinical trials (Gounder et al., 2023).

Hence, we will take advantage of the repetition of this experiment to substitute this new molecule instead of DBZ, that is an interesting molecule for preclinical research, but without any clinical relevance. **Therefore, the use of nirogacestat will further increase the medical impact of our data.** Importantly, nirogacestat is better tolerated than DBZ, meaning that mice can be treated for longer periods of time and we propose in here to treat up to 12 weeks. Finally, after discussion with Quentin Thomas, author of the manuscript and clinical researcher in the lab, we will provide 4 carboplatin cycles as it is proposed today to NSCLC patients in an attempt of getting closer to the clinical setting. In particular we will provide carboplatin to mice on weeks 1, 4, 7 and 10, while treating with MDM2 inhibitor (SP141) and Notch inhibitor (nirogacestat) from Monday to Friday for the 12 weeks.

This experiment will be long and will require an important use of resources both human and financial, but we are sure that the effect in tumor growth and survival will be more dramatic than the one presented now.

On the contrary and as explained in the 4th subheading part of this "revision plan", including another 72 mice to treat a p53 proficient NSCLC PDX, when we already demonstrated in vitro that p53 is not required for the phenotype described in this study, for us it is totally unfeasible by ethical reasons, i.e., the use of animals in research (please see below for further details).

All the new data will appear as **new Figure 5 (B to E)**. For **new Figure 5A** please see below the major comment 2 of Reviewer 2.

5. Though beyond the current study's scope, in the discussion section, the authors may want to propose or hypothesize on how MDM2-mediated NICD stabilization

Revision Plan

contributes to carboplatin resistance. This could provide valuable insights for future research directions.

[Authors] We will discuss this part as proposed by the reviewer.

6. In the Western blot results, the total ATM and ATR controls were absent.

[Authors] The reviewer is totally right and we will repeat experiments to include all the totals as requested.

7. Authors may choose to include a graphical abstract at the end of their study to visually illustrate the mechanisms they have described.

[Authors] Very good idea thanks, we will do it.

Reviewer #1 (Significance (Required)):

Advance: The authors aim to present a novel perspective on the resistance mechanisms to platinum compounds in NSCLC therapy. They explore platinum compounds-induced DNA damage, ATM activation, and MDM2-mediated stabilization of the active form of NOTCH (NICD). However, to strengthen their claims, they must provide more conclusive results.

Audience: This manuscript will likely engage oncologists who investigate the chemotherapy-resistant mechanisms of platinum compounds in NSCLC treatment, as well as scientists specializing in NOTCH and MDM2 pathways. However, the manuscript's central claims lack robust support from the available data, and the current approaches employed are not sufficiently thoughtful and rigorous; there is room for improvement.

My expertise is molecular medicine, cancer biology, and epigenetics.

[Authors] We want to thank again this reviewer for her/his helpful comments that will increase the impact and the relevance of our study while keeping the original message.

Revision Plan

We are also very satisfied when she/he said: "This manuscript will likely engage oncologists who investigate the chemotherapy-resistant mechanisms of platinum compounds in NSCLC treatment".

Reviewer #2 (Evidence, reproducibility and clarity (Required)):

In this manuscript, Sara Bernardo et al. investigated the molecular mechanisms underlying the activation of the Notch signaling in response to DNA damage induced by platinum-based chemotherapeutic agents in non-small cell lung cancer (NSCLC). They demonstrated that carboplatin treatment induces DNA double-strand breaks (DSBs) and stabilizes NICD, a process dependent on ATM and mediated by MDM2. In vivo experiments in patient-derived xenografts (PDX) showed that inhibition of NICD and MDM2 enhanced platinum effectiveness. Furthermore, clinical analysis revealed a correlation between MDM2 expression and poor prognosis in NSCLC patients treated with platinum compounds, emphasizing the clinical relevance of the MDM2-NICD axis in platinum resistance.

[Authors] We thank this reviewer for her/his nice synopsis of our study.

Major comments:

Overall, the authors have conducted experiments that sufficiently elucidate their claims, and the description of the experiments is detailed. However, there is still room for the improvement.

[Authors] We are very pleased that reviewer finds our experimental work "...sufficiently elucidate their claims, and the description of the experiments is detailed." And we are sure that after all the new experiments we are proposing in here, she/he will be fully satisfied.

1.The finding that MDM2 promoted NICD stability through non degradative ubiquitination is interesting and in line with a previous study. As it is also known that NICD is regulated by various post-translational modifications, including ubiquitination that promotes NICD degradation. It is unclear what's the potential difference between these two types of ubiquitination. For example, do these two differ in specific ubiquitination sites? Can the authors provide some discussion?

Revision Plan

[Authors] We agree with the reviewer and hence we will perform a new set of experiments to determine the role of 2 key lysine residues in the ubiquitin protein promoting either degradation or DNA binding. As explained in detail in major point 1 from reviewer 1, we will determine if DNA damage promotes ubiquitination in position 48, i.e., to degrade, or in position 63, i.e., to facilitate the binding to DNA for repairing upon DNA damage, or in any of these 2 positions. And as mentioned above, we will then confirm that MDM2 is responsible of the specific ubiquitination type we will uncover. We are sure that the reviewer will be satisfied by these new data once is generated.

As for the specific ubiquitination sites in NICD, there are up to 17 lysine residues susceptible of being ubiquitinated. Hence unveiling what residues are targeted by MDM2 and if they differ from others inducing degradation as those promoted by the E3 ligase FBXW7, we feel is out of the scope of the current manuscript. Still, we will discuss all this part as kindly proposed by the reviewer.

2. Could the overexpression of MDM2 or NICD lead to carboplatin resistance in A549 or H358 cells?

[Authors] This is a very interesting experiment and prompted by the reviewer's comment we started the subcloning of inducible NICD into lentiviral vectors to generate stable cells and test the carboplatin sensitivity in presence of different levels of NICD. These new data will be the **new Figure 5A**.

3. The trends observed in the western blot data within the manuscript appear inconsistent. While the authors propose that NICD levels increased upon incubation with carboplatin, the discrepancy arises when considering the NICD levels without cycloheximide (CHX) treatment in Figure 1E, where no significant elevation is observed (Lane 6 vs. Lane 1).

[Authors] The point of the reviewer is well taken. Please bear in mind that in here we are handling several signaling pathways that interact among them while having each one different kinetics. Our finding of increased NICD upon carboplatin treatment is highly consistent in vitro and in vivo, but it is true that in the experiment mentioned by the reviewer is not obvious, probably due to some kinetic issue. We are repeating this experiment to have the increased in NICD upon carboplatin as it is in the rest of the manuscript (up to 9 times only in main figures).

4. The quality of western blots needs to be improved, especially Fig. 1C and S1C, also Figure 3B. Moreover, the NICD western blot sometimes appears as one band and

Revision Plan

sometimes as two bands. Please provide an explanation. If possible, please quantify the bands in western blots.

[Authors] We agree with the reviewers that not all WB have the same quality and we will repeat some of them to homogenize the quality all over the manuscript, and particularly, we will repeat the ones kindly pointed out by the reviewer.

The two bands it is something we also noticed and we will pay attention while reproducing the WB, since it might be related to discrepancies in the percentage of acrylamide. If this is not the case, i.e., upon repetition we still observe in some conditions and not in others, we will provide explanations for this in the new submission as kindly proposed by the reviewer.

Finally, and also as proposed by the reviewer we will quantify the WB bands.

5. Please provide a necessary discussion on whether the targeted treatment approach towards the MDM2-NICD axis is applicable to all patients or only to those with high expression of MDM2/NICD.

[Authors] In the discussion of the current manuscript, we focused into the MDM2 high expression subset of patients for this issue, but in the next submission we will enlarge to patients with high levels of NICD also.

6. How to interpret the significance of the simultaneous increase in NICD ubiquitination and stability mediated by MDM2? Please provide a relevant discussion.

[Authors] We will provide strong experimental data to go beyond discussion (please see above the experiments with ubiquitin mutants), but we will also provide discussion of this particular point.

7. In Figure 5B, please also check the level of MDM2. In Figure 5C, carboplatin appears to have little impact on tumor growth. How to explain the increase of Ki-67 in the carboplatin treatment group in Figure 5A?

[Authors] We will measure also levels of MDM2 in the future **new Figure 5C** as requested by the reviewer.

Revision Plan

As for the interesting observation of the Ki67, since we will repeat the whole experiment, we will pay special attention to this point if ever it is repeated. Should be this the case, we will elaborate an explanation.

Minor comments:

1. Please include scale bars in Figure 1B and Supplemental Figure 1B.

[Authors] We thank the reviewer for this comment. We will include the scale bars where required.

2. Figure 5D, the P values of the survival curve should be indicated in the figures.

[Authors] We will include the P values in the future **new Figure 5E**.

3. The presentation of survival curve data in Figures 5D and 6A should be consistent.

[Authors] The point of the reviewer is well taken and we will use Prism to draw the PFS for patients in Figure 6A as we did for the mice in current Figure 5D.

4. It seems that supplemental figure 2 is missing.

[Authors] We actually jumped from supplemental figure 1 to 3 because we do not have any associated supplemental figure to main Figure 2. We will clarify this point in the next submission.

5. Please carefully check the spelling of the entire text, for example, on page 20, line 426 it should be 'western'. Also, please spell out the abbreviations DDR and ATM.

[Authors] We will double check all spelling and provide the abbreviations kindly suggested by the reviewer.

6. The abbreviation for Cleaved caspase 3 should be CC3.

Revision Plan

[Authors] We thank the reviewer for this information, we will use CC3 in the next submission.

Reviewer #2 (Significance (Required)):

Notch signaling is associated with the occurrence and development of non-small cell lung cancer (NSCLC). Previous study indicates that the expression of Notch protein is significantly higher in NSCLC tissues compared to normal tissues (PMID: 31170211). Additionally, the upregulation of Notch1 is correlated with higher tumor grades, lymph node metastasis, tumor-node-metastasis (TNM) staging, and poor prognosis (PMID: 25996086). Abnormal activation of Notch signaling pathway is frequently observed in chemotherapy-resistant NSCLC, and some studies have aimed to address NSCLC drug resistance via modulating Notch signaling (PMID: 30087852, 38301911). This manuscript firstly proposes that MDM2-mediated stabilization of NICD upon DNA damage plays a major role in NSCLC response to platinum chemotherapy. It further suggests that targeting the MDM2-NICD axis could prove to be an effective therapeutic strategy. Overall, this work unveils a novel mechanism for Notch activation in response to platinum chemotherapy, providing a renewed outlook on overcoming chemotherapy resistance in NSCLC. This manuscript will attract those interested in the mechanisms of chemotherapy resistance and novel treatment approaches.

[Authors] We sincerely thank the reviewer for finding that our "...work unveils a novel mechanism for Notch activation in response to platinum chemotherapy, providing a renewed outlook on overcoming chemotherapy resistance in NSCLC". We are also very satisfied when she/he says: "This manuscript will attract those interested in the mechanisms of chemotherapy resistance and novel treatment approaches."

Finally, we are convinced that the reviewer will appreciate all the new proposed experimental data, and also that upon finishing all experiments, she/he will think that the manuscript will be suitable for publication.

Revision Plan

3. Description of the revisions that have already been incorporated in the transferred manuscript

For simplicity, we decided to introduce all changes in next submission upon conclusion of all experimental approaches proposed above.

4. Description of analyses that authors prefer not to carry out

While we will perform almost all experiments proposed by reviewers, there is one we feel is not possible to do due to ethical reasons. Reviewer 1 wanted us to perform a new in vivo experiment with the same PDX using up to 6 treatment groups. We use 8 mice per condition (for tumor growth and survival) plus 4 for the “acute” treatment for WB and IHC purposes, hence 12 mice x 6 groups = 72 mice, and we will perform this experiment as indicated above and proposed by the reviewer.

On the contrary, the reviewer asked us also to repeat the same experiment with a PDX p53 proficient. While we understand the possible interest, since we demonstrated in vitro that p53 is not required for the protective phenotype of MDM2 and Notch upon DNA damage, we honestly believe that using another 72 mice to confirm this aspect in vivo, is against the rational use of animals in research going against the 3Rs rule. Hence, we will not perform this experiment unless Editors believe is strictly required.

REFERENCES

- Arena, G., Cisse, M. Y., Pyrdziak, S., Chatre, L., Riscal, R., Fuentes, M., Arnold, J. J., Kastner, M., Gayte, L., Bertrand-Gaday, C., *et al.* (2018). Mitochondrial MDM2 Regulates Respiratory Complex I Activity Independently of p53. *Mol Cell* 69, 594-609 e598.
- Cheng, Q., Cross, B., Li, B., Chen, L., Li, Z., and Chen, J. (2011). Regulation of MDM2 E3 ligase activity by phosphorylation after DNA damage. *Mol Cell Biol* 31, 4951-4963.
- Gounder, M., Ratan, R., Alcindor, T., Schoffski, P., van der Graaf, W. T., Wilky, B. A., Riedel, R. F., Lim, A., Smith, L. M., Moody, S., *et al.* (2023). Nirogacestat, a gamma-Secretase Inhibitor for Desmoid Tumors. *N Engl J Med* 388, 898-912.
- Kassouf, T., Larive, R. M., Morel, A., Urbach, S., Bettache, N., Marcial Medina, M. C., Merezegue, F., Freiss, G., Peter, M., Boissiere-Michot, F., *et al.* (2019). The Syk Kinase Promotes Mammary Epithelial Integrity and Inhibits Breast Cancer Invasion by Stabilizing the E-Cadherin/Catenin Complex. *Cancers (Basel)* 11.
- Liu, P., Gan, W., Su, S., Hauenstein, A. V., Fu, T. M., Brasher, B., Schwerdtfeger, C., Liang, A. C., Xu, M., and Wei, W. (2018). K63-linked polyubiquitin chains bind to DNA to facilitate DNA damage repair. *Sci Signal* 11.

Revision Plan

Pettersson, S., Sczaniecka, M., McLaren, L., Russell, F., Gladstone, K., Hupp, T., and Wallace, M. (2013). Non-degradative ubiquitination of the Notch1 receptor by the E3 ligase MDM2 activates the Notch signalling pathway. *Biochem J* 450, 523-536.

Riscal, R., Schrepfer, E., Arena, G., Cisse, M. Y., Bellvert, F., Heuillet, M., Rambow, F., Bonneil, E., Sabourdy, F., Vincent, C., *et al.* (2016). Chromatin-Bound MDM2 Regulates Serine Metabolism and Redox Homeostasis Independently of p53. *Mol Cell* 62, 890-902.

29th Jul 2024

Dear Dr. Maraver,

Thank you for the submission of your manuscript to our editorial offices. I have now had the opportunity to read it, together with the referees' reports and your rebuttal letter, and to discuss them with the other members of our editorial team.

We agree that the study fits the scope of the journal, and we appreciate that you are willing to address most of the referees' concerns, with the exception of the additional PDX experiment with p53 proficient cells. We thus encourage you to submit a revised version of your manuscript, including the modifications and revisions described in your point-by-point letter.

Acceptance of the manuscript will entail a second round of review. EMBO Molecular Medicine encourages a single round of revision only and therefore, acceptance or rejection of the manuscript will depend on the completeness of your responses included in the next, final version of the manuscript. For this reason, and to save you from any frustrations in the end, I would strongly advise against returning an incomplete revision.

We require:

- 1) A .docx formatted version of the manuscript text (including legends for main figures, EV figures and tables). Please make sure that the changes are highlighted to be clearly visible.
- 2) Individual production quality figure files as .eps, .tif, .jpg (one file per figure). For guidance, download the 'Figure Guide PDF' (<https://www.embopress.org/page/journal/17574684/authorguide#figureformat>).
- 3) At EMBO Press we ask authors to provide source data for the main figures. Our source data coordinator will contact you to discuss which figure panels we would need source data for and will also provide you with helpful tips on how to upload and organize the files.
- 4) A .docx formatted letter INCLUDING the reviewers' reports and your detailed point-by-point responses to their comments. As part of the EMBO Press transparent editorial process, the point-by-point response is part of the Review Process File (RPF), which will be published alongside your paper.
- 5) A complete author checklist, which you can download from our author guidelines (<https://www.embopress.org/page/journal/17574684/authorguide#submissionofrevisions>). Please insert information in the checklist that is also reflected in the manuscript. The completed author checklist will also be part of the RPF.
- 6) All Materials and Methods need to be described in the main text using our 'Structured Methods' format, which is required for all research articles. According to this format, the Methods section includes a Reagents and Tools Table (listing key reagents, experimental models, software and relevant equipment and including their sources and relevant identifiers) followed by a Methods and Protocols section describing the methods using a step-by-step protocol format. More information on how to adhere to this format as well as a downloadable template (.docx) for the Reagents and Tools Table can be found in our author guidelines:
<https://www.embopress.org/page/journal/17574684/authorguide#structuredmethods>
- 7) Please note that all corresponding authors are required to supply an ORCID ID for their name upon submission of a revised manuscript.
- 8) It is mandatory to include a 'Data Availability' section after the Materials and Methods. Before submitting your revision, primary datasets produced in this study need to be deposited in an appropriate public database, and the accession numbers and database listed under 'Data Availability'. Please remember to provide a reviewer password if the datasets are not yet public (see <https://www.embopress.org/page/journal/17574684/authorguide#dataavailability>). In case you have no data that requires deposition in a public database, please state so in this section. Note that the Data Availability Section is restricted to new primary data that are part of this study.

9) For data quantification: please specify the name of the statistical test used to generate error bars and P values, the number (n) of independent experiments (specify technical or biological replicates) underlying each data point and the test used to calculate p-values in each figure legend. The figure legends should contain a basic description of n, P and the test applied. Graphs must include a description of the bars and the error bars (s.d., s.e.m.). Please provide exact p values.

10) Our journal encourages inclusion of *data citations in the reference list* to directly cite datasets that were re-used and obtained from public databases. Data citations in the article text are distinct from normal bibliographical citations and should directly link to the database records from which the data can be accessed. In the main text, data citations are formatted as follows: "Data ref: Smith et al, 2001" or "Data ref: NCBI Sequence Read Archive PRJNA342805, 2017". In the Reference list, data citations must be labeled with "[DATASET]". A data reference must provide the database name, accession number/identifiers and a resolvable link to the landing page from which the data can be accessed at the end of the reference. Further instructions are available at .

11) We replaced Supplementary Information with Expanded View (EV) Figures and Tables that are collapsible/expandable online. A maximum of 5 EV Figures can be typeset. EV Figures should be cited as 'Figure EV1, Figure EV2' etc... in the text and their respective legends should be included in the main text after the legends of regular figures.

12) The paper explained: EMBO Molecular Medicine articles are accompanied by a summary of the articles to emphasize the major findings in the paper and their medical implications for the non-specialist reader. Please provide a draft summary of your article highlighting

13) For more information: There is space at the end of each article to list relevant web links for further consultation by our readers. Could you identify some relevant ones and provide such information as well? Some examples are patient associations, relevant databases, OMIM/proteins/genes links, author's websites, etc...

14) Author contributions: CRediT has replaced the traditional author contributions section because it offers a systematic machine readable author contributions format that allows for more effective research assessment. Please remove the Authors Contributions from the manuscript and use the free text boxes beneath each contributing author's name in our system to add specific details on the author's contribution. More information is available in our guide to authors.

15) Disclosure statement and competing interests: We updated our journal's competing interests policy in January 2022 and request authors to consider both actual and perceived competing interests. Please review the policy <https://www.embopress.org/competing-interests> and update your competing interests if necessary.

16) Every published paper now includes a 'Synopsis' to further enhance discoverability. Synopses are displayed on the journal webpage and are freely accessible to all readers. They include a short stand first (maximum of 300 characters, including space) as well as 2-5 one-sentences bullet points that summarizes the paper. Please write the bullet points to summarize the key NEW findings. They should be designed to be complementary to the abstract - i.e. not repeat the same text. We encourage inclusion of key acronyms and quantitative information (maximum of 30 words / bullet point). Please use the passive voice. Please attach these in a separate file or send them by email, we will incorporate them accordingly.

17) As part of the EMBO Publications transparent editorial process initiative (see our Editorial at <http://embomolmed.embopress.org/content/2/9/329>), EMBO Molecular Medicine will publish online a Review Process File (RPF) to accompany accepted manuscripts.

In the event of acceptance, this file will be published in conjunction with your paper and will include the anonymous referee reports, your point-by-point response and all pertinent correspondence relating to the manuscript. Let us know whether you agree with the publication of the RPF and as here, if you want to remove or not any figures from it prior to publication. Please note that the Authors checklist will be published at the end of the RPF.

I look forward to receiving your revised manuscript.

Yours sincerely,

Lise Roth

Rev_Com_number: RC-2024-02497

New_manu_number: EMM-2024-20361-T

Corr_author: Maraver

Title: NOTCH1 intracellular domain stabilization by MDM2 plays a major role in NSCLC platinum response

We sincerely thank the Editors and the Reviewers for the time devoted to our manuscript. We found their comments interesting and very helpful. Thanks to a massive collaborative effort, we answered to all the reviewers' comments by adding significantly new experimental data including new *in vivo* experiments. In total we included 23 new panels in the manuscript and 1 only for reviewers.

We are also very encouraged by the positive comments of the Reviewers:

"This manuscript will likely engage oncologists who investigate the chemotherapy-resistant mechanisms of platinum compounds in NSCLC treatment" (Reviewer 1);

"Overall, the authors have conducted experiments that sufficiently elucidate their claims, and the description of the experiments is detailed."; and *"Overall, this work unveils a novel mechanism for Notch activation in response to platinum chemotherapy, providing a renewed outlook on overcoming chemotherapy resistance in NSCLC"* (Reviewer 2).

We are also aware that both reviewers agreed that there was room for improvement. Conversely, upon answering to all their concerns by performing a substantial number of new experiments, we are sure that reviewers will concur with us that the study greatly improved its quality and impact and find that now is ready for publication.

Reviewer #1 (Evidence, reproducibility and clarity (Required)):

Summary:

This manuscript from Maraver and co-authors investigates the putative resistance mechanisms that hinder the efficacy of platinum-based therapies (e.g., carboplatin) against non-small cell lung carcinoma (NSCLC). Using *in vitro* lung cancer cell lines, shRNA-based knockdown, and exogenous overexpression systems, the research describes a DNA damage-induced resistance mechanism involving the NOTCH signaling pathway and the E3 ligase MDM2. The authors show that carboplatin treatment induces DNA damage and promotes ATM activation, which in turn activates the NOTCH signaling pathway via ubiquitination and stabilization of the Notch Intracellular Domain (NICD). New findings include the MDM2-mediated ubiquitination and stabilization of NICD. Using *in vivo* NSCLC-PDX models, they demonstrate that combining carboplatin with Notch and MDM2 inhibitors can enhance tumor killing, suggesting that targeting the MDM2/NICD axis in conjunction with carboplatin may be a viable therapeutic alternative. Furthermore, they show that NICD and MDM2 levels are elevated among tumor samples from chemo-resistant patients. Consistent with these findings, high MDM2 levels correlate with poor progression-free survival (PFS) in NSCLC patients.

[Authors] We thank this reviewer for her/his fair summary of our work that highlights our new findings.

Major comments:

Some of the key conclusions may not be convincing.

[Authors] We fully understand the concerns that reviewer had but we are sure that our new version will fully satisfy her/him and will agree with us that the study is ready for publication.

1. One significant weakness of the manuscript is the lack of exploration into the underlying mechanism of how MDM2 mediates the stabilization of NICD. While the observation of MDM2-mediated NICD stabilization is intriguing, it is important to provide a more convincing explanation for the reviewers. This could be achieved by offering a detailed molecular mechanism, especially considering that MDM2 typically targets proteins for degradation.

[Authors] We mentioned in our previous version the seminal work demonstrating that MDM2 induced ubiquitination on NICD but not for degradative purposes (Pettersson *et al*, 2013). Still and after reading this reviewer's comment, we realized that we might not discussed it with enough detail and now improved it in the current version. In this particular study, they used a mutated form of ubiquitin in lysine 48 that changes to an arginine, i.e., the K48R mutant. This lysine is responsible for the ubiquitin chains generation and recognition by the proteasome (Petroski & Deshaies, 2005), and hence when this mutant is used, proteins that normally follow this path cannot be degraded. Hence, the authors of this preceding study, not related to DNA damage, demonstrated that in the presence of K48R mutant, MDM2 was still able to induce ubiquitination in NICD and therefore it was not degradative.

To answer the reviewer and to confirm that in our experimental setting MDM2 does ubiquitinate but not degrade NICD, we performed a similar experiment using same K48R mutant to formally prove that MDM2 in former Fig. 4B does not ubiquitinate NICD via lysine 48-linked polymers, and hence it is not degradative. We had access to this tool thanks to Laetitia Linares, author of our study and long-lasting expert in MDM2 (for instance see (Riscal *et al*, 2016) and (Arena *et al*, 2018)) and its ubiquitination activity (Linares *et al*, 2003).

In **New Fig. 4C** we observed that as we showed in our previous version, MDM2 ectopic expression induces the ubiquitination of NICD. It is known that contrary to MDM2, several E3 ligases including FBX7, Itch or Sel-10, to name only a few, induce NICD protein degradation to fine tune Notch activity (Lai, 2002) (Welcker & Clurman, 2008). In accordance with this, we observed that NICD was highly accumulated in the presence of the K48R mutant, as well as it was its ubiquitinated form (**New Fig. 4C** right panel). Importantly, this ubiquitination was further enhanced in the presence of ectopic expression of MDM2, demonstrating that indeed **in our experimental system the MDM2 mediated NICD ubiquitination does not promote degradation**. To help readers, we quantified

the levels of ubiquitination normalizing with the levels of ectopic His-ubiquitin following another comment from reviewer 2 (see below).

2. Another weakness lies in the unclear role and the underlying mechanism of ATM in the MDM2-mediated NICD stabilization. While the data presented (Fig. 3B, 3C) suggest that carboplatin could elevate MDM2 levels for NICD stabilization, a more precise method to induce MDM2 overexpression specifically for targeting NICD is required. It appears that ATM plays a crucial role in this regulatory process. The following questions must be addressed: Does ATM induce the phosphorylation of MDM2 for its protein stabilization and/or E3 ligase activity?

[Authors] There are several points here.

We are not sure we understand the first one, i.e., regarding the use of a more precise method to induce MDM2 overexpression since it is exactly what we did in former Fig. 4A, i.e., ectopic expression of MDM2 to demonstrate that MDM2 is sufficient to increase NICD levels, and hence we agree with the reviewer that this experiment was required. Even more, now we further expanded our knowledge by adding another MDM2 mutant (see below).

For the second one, i.e., the role of ATM kinase activity on MDM2 in our system, we performed a dedicated experiment to answer the reviewer. There are up to six proposed residues in MDM2 to be phosphorylated by ATM upon DNA damage: S386, S395, S407, T419, S425, and S429 (Cheng *et al*, 2011), with S395 being the most important one. We benefit again from the expertise of Dr Linares, and we used a MDM2 S395A mutant, in which serine at position 395 is substituted with an alanine to abolish the phosphorylation induced by ATM in this residue (Maya *et al*, 2001). We used this mutant together with the WT and the ligase death 464A MDM2 proteins already used in our previous version generating **New Fig. 4A, 4B** and **New Fig. EV2D**.

As before, in **New Fig. 4A**, we observed that ectopic expression of MDM2 increased NICD levels, but also that the E3 ligase death mutant of MDM2 (464A) could not. Now and thanks to the reviewer's comment, we observed that the S395A mutant also fail to increase NICD levels. To the best of our knowledge the serine 395 in MDM2 is only phosphorylated by ATM and hence our new data further reinforce our original message, i.e., ATM is required to increase NICD levels by acting on MDM2. Of note, both MDM2 mutants were expressed to a higher extension than the WT one in this experiment, and hence the high NICD levels cannot be due to a higher protein amount of WT MDM2 compared to the mutants (**New Fig. 4A**).

In our previous version we also showed that MDM2 WT and E3 ligase death mutant were able to interact with NICD. Now since we are using also the S395A one, we repeated the co-immunoprecipitation assay. In **New Fig. EV2D** we observed that all the MDM2 forms were able to interact with NICD confirming and

expanding our previous observation, i.e., the activity of MDM2 is not required for the interaction with NICD.

The reviewer also asked if ATM was required for MDM2 E3 ligase activity. To answer to this specific point, we performed another experiment. We used again overexpression of His-ubiquitin but this time using only the WT form. As in our previous version we observed that MDM2 induces ubiquitination of NICD while the E3 ligase death version of MDM2 could not, confirming our previous observations. Of note, now we observed that the S395A mutant could not either achieves the ubiquitination levels on NICD promoted by the WT MDM2 form. Furthermore, the ectopic expression of this mutant could not either accomplish the ubiquitination levels promoted by the endogenous MDM2 (**New Fig. 4B**). Finally, the ectopic expression levels of the different MDM2 forms, could not explain the differences observed in NICD ubiquitination implying that **MDM2 serine 395 and, by extension the kinase responsible for its phosphorylation, i.e., ATM, play a critical role in NICD ubiquitination.**

Taking together our new data further enhanced our original link with ATM and we are in debt with the reviewer for her/his comment that strengthened our message.

3. The combination therapy of carboplatin with MDM2 and NICD inhibitors may lack compelling rationale (see below).

[Authors] This is a very important point but we discuss it below, where more information is provided by the reviewer. Still, we anticipate we performed a new *in vivo* experiment to answer to this relevant point.

4. In lines 275-276, the authors stated that their preclinical data establish the enhancement of carboplatin's therapeutic effect in NSCLC *in vivo* through MDM2-NICD axis inhibition. However, it's important to note that this finding remains preliminary at this stage.

[Authors] We adapted the message and now in line 411 it reads: "*All in all, our data suggest that targeting the MDM2-NICD axis could enhance the carboplatin therapeutic effect on NSCLC patients. An important question arises regarding what subset of patients could benefit from Notch inhibition, and to answer it in our local hospital ICM, we are currently conducting a clinical trial aiming to monitor Notch activity in NSCLC patients: Alcina 2 cohort 11 (NCT04025541). Hence our clinical research should pave the way for future clinical trials aiming to treat NSCLC patients using Notch inhibitors*".

Minor comments:

1. The observed loss of NICD during ATMi + carboplatin treatment in Figures

2A and 2B raises the question of whether ATM regulates the gene transcription of NOTCH. In addition to the CHX assay conducted in Figs 2C and 2D, quantifying NOTCH mRNA upon ATM inhibition could provide further insights. Alternatively, referencing relevant studies on this topic may strengthen the discussion.

[Authors] In our previous version we already showed that full length NOTCH1 protein expression was not mirroring the NICD protein levels upon carboplatin treatment, indicating that DNA damage most probably was not affecting massively mRNA transcription and hence at the time we favored the increased NICD through post translational regulation. Now, and to answer the reviewer, we performed qPCR analysis on H358 and A549 cells treated or not with carboplatin and/or ATM inhibitor.

In this Figure only for reviewers, we observed a trend in A549 cells that could indicate that carboplatin is inducing *NOTCH1* mRNA expression. Since *NOTCH1* mRNA is a direct target of NICD (Weng *et al*, 2006), it is difficult to know if this small trend is associated to the DNA damage directly or because NICD accumulates and then induces *NOTCH1* mRNA. Conversely, we did not find same trend in H358 cells. Furthermore, in both cell lines there was a trend that could indicate that *NOTCH1* mRNA is increased upon carboplatin and inhibition of ATM. We believe this is because since the carboplatin-induced NICD protein expression is hindered upon ATM inhibition, cells try to react with more *NOTCH1* mRNA. All in all, we prefer to don't publish these data that would complicate the message of our study. Still, if the reviewer feels that is required, please let us know and we will include it in the supplemental data.

2. In Figs 4A and 4B, the noticeable discrepancy between the exogenous expression of wild-type (WT) MDM2 and catalytically inactive MDM2-464A raises concerns. It is essential to consider if the reduced ubiquitination and stability of NICD might be attributed to varying levels of MDM2-464A in the cells rather than its catalytic inactivity. While p53 ubiquitination was utilized

as a control, ensuring comparable levels of MDM2 and MDM2-464A expression could enhance the experimental rigor. Compared to the smear poly-ubiquitination bands observed for MDM2 in Fig. 4B, the ubiquitination of NICD appears simpler. What distinguishes the feature of MDM2-mediated NICD ubiquitination? Could it potentially involve mono-ubiquitination?

[Authors] The point of the reviewer is well taken. Importantly, as mentioned above in main point 2, we repeated both experiments and in **New Fig. 4A, 4B** and **New Fig. EV2D**, the 2 MDM2 mutants were not expressed less than their WT counterpart, and hence the possibility proposed by the reviewer is no longer possible.

Regarding the type of ubiquitination, as explained in detail in major point 1 above, we demonstrated that it is not degradative. As for the comment about if MDM2 promotes mono- or poly-ubiquitination in NICD we agree with the reviewer that it seems it could be the first one. This is very interesting and we want to explore it in a follow up study, in other words, we respectfully feel it is out of the scope of the current work and in the discussion section we wrote in line 379: *“Likewise, our data is in accordance with a seminal study also showing that MDM2 ubiquitinates NICD but not for degradation (Pettersson et al., 2013). Conversely, at this stage we cannot know if MDM2 promotes mono- or poly-ubiquitination on NICD. We do not know either among the 17 lysins presented in NICD which is (are) targeted by MDM2, and if they differ from those recognized by E3 ligases that ubiquitinate NICD for degradation (Lai, 2002) (Welcker & Clurman, 2008). All these questions are under study currently in the laboratory”.*

3. In Fig. 5A, the authors need to consider conducting additional NOTCH-associated factors to definitively demonstrate the activation of NOTCH signaling beyond HES1. Alternatively, in Fig. 5B, the NICD Western blot could be complemented by detecting HES1 or other NOTCH-associated factors.

[Authors] To answer to this particular point, we tested all antibodies we found for NOTCH1 targets, and we identified one that worked in WB but sadly not in IHC. This antibody recognizes HEY1, that as HES1, is a bHLH transcription factor downstream of the Notch pathway (Weber *et al*, 2014). In **New Fig. 5E** we observed that carboplatin treatment induced HEY1 expression *in vivo* concomitantly with NICD, while as it happened in IHC, HES1 was not. We are in debt with the reviewer for this nice proposition that reinforce one of the main messages of this study, i.e., **platinum induction of DNA damage increases the Notch pathway**.

4. In Fig.s 5C and 5D, crucial control groups are missing, specifically mice treated solely with SP141+DBZ, carboplatin+SP141, and SP141+DBZ. It is essential to include these groups to demonstrate that the enhanced tumor killing results from the combination of carboplatin with SP141 and/or DBZ,

rather than from SP141 and DBZ alone. Furthermore, in addition to the currently used NSCLC-PDX model harboring the p53 (P151R) mutation, it would be informative to include a NSCLC-PDX model expressing WT p53.

[Authors] This is a crucial point in this rebuttal as mentioned before in major point 3 and we detail it in here.

We did only 3 groups because we wanted to perform a proof for the concept of the *in vivo* feasibility of the triple combination. But, after reading this point from Reviewer 1 we agree that it was an important point and made a massive effort to answer to her/him.

The reviewer specifically asked us to include: “SP141+DBZ, carboplatin+SP141, and SP141+DBZ”. We imagine that she/he meant: SP141+DBZ, carboplatin+SP141, and carboplatin+DBZ, that together with the vehicle, carboplatin, and carboplatin+SP141+DBZ makes 6 groups of treatments. Putting together the 8-9 mice devoted for tumor growth and survival, plus 3 mice for the acute treatment for IHC and WB purposes (to also repeat former Fig. 5A and 5B) makes a total of more than 70 mice, that is a substantial number of animals. We decided to extend the treatments to increase the impact of the study and hence, the experiment was long and required an important use of resources both human and financial, but we agree with the reviewer that this information was important to include in the study. On the contrary, we feel that doubling the number of mice to treat a p53 proficient NSCLC PDX, when we already demonstrated *in vitro* that p53 is not required for the phenotype described in this study, it goes against the 3Rs rule for the use of animals in research, and hence we decided to do not perform the experiment with a second PDX with WT p53 due to ethical reasons. The new experiment provided four panels of the **new Fig. 5 (B to E)**.

One of the first things we learned upon performing this new experiment it was that combining SP141 and DBZ had deleterious effects on mice when treating more than two weeks as we did in our first version. Specifically, we observed a strong weight loss that forced us to euthanize some animals due to human end point. Of note, we gave drug holidays to all groups when noticing strong weight loss in several mice and some recovered, while other they never did. In particular we lost 3 out 9 mice in the SP141 and DBZ combination group, i.e., in the absence of carboplatin. Not surprisingly, the addition of carboplatin, i.e., the triple combination, further enhanced the weight loss and we needed to kill 8 out 9 mice. Hence, with the molecules currently available we cannot propose the triple combination.

Still, and thanks to the reviewer remark, we observed a very strong effect in survival for the treatment of carboplatin with DBZ (Notch inhibitor) when compared to vehicle. In fact, the survival in carboplatin with DBZ group was also significantly increased against the carboplatin in monotherapy (**New Fig. 5C**), that as in our previous version, we observed no therapeutic effect in this group proving again that this PDX model displays carboplatin intrinsic resistance (Ferrer *et al*, 2018). Interestingly, the only other treatment group achieving increased survival versus the vehicle was the combination of carboplatin with SP141, i.e.,

the MDM2 inhibitor. In contrast, the combination of SP141 and DBZ did not increase survival, demonstrating that both molecules increase the carboplatin therapeutic effect but do not promote an effect by themselves. Hence, thanks to the reviewer we can affirm now that **both MDM2 and Notch inhibitors increase the potency of platinum treatment in vivo** and we open the possibility that upon development of new inhibitors against these proteins, eventually a triple combination could be proposed.

Finally, since we passed from 3 to 5 treatment groups, and in order to simplify the panels, we decided to analyze only the more important markers. Even more, as mentioned above in minor point 3 from the same reviewer we also included HES1 and HEY1 (both downstream of Notch pathway) in the former Fig. 5B (WB), and finally, following another comment from reviewer 2, we also included in the **New Fig. 5E** also MDM2, all defining proper target inhibition. Hence to homogenize but also again to simplify, we decided to eliminate CC3 and Ki67 from the IHC panel. Following with the same idea of simplifying the panels, we passed from 4 to 2 tumor samples in the WB from previous Fig. 5B. Should the reviewers feel that we need to include CC3 and/or Ki67 in the IHC panel, please let us know and we will perform the stain and quantification accordingly.

Our data now is more solid and we want to thank again the reviewer for her/his comment.

5. Though beyond the current study's scope, in the discussion section, the authors may want to propose or hypothesize on how MDM2-mediated NICD stabilization contributes to carboplatin resistance. This could provide valuable insights for future research directions.

[Authors] As proposed by the reviewer we included a specific paragraph in line 399 of the discussion section that reads: *“Our preclinical data in mice harboring NSCLC PDXs with carboplatin intrinsic resistance, further support this hypothesis because the combination of carboplatin and NICD release inhibition increased DNA damage in tumor cells as well as mice survival. Our work is in accordance with previous studies in ovarian and cervical cancers showing that NICD inhibition enhanced DNA damage upon platinum treatment (Li et al, 2019; McAuliffe et al, 2012) and another one in colorectal cancer where NICD inhibition increased the radiation-induced DNA damage (Zhang et al, 2018), that also fit with our study demonstrating that is the DNA damage response the responsible of NICD stabilization. Interestingly, using a D. melanogaster ovarian tumor model, a previous study showed that NICD ectopic expression promoted the expression of double strand break repair genes (Jevitt et al, 2021), and we are investigating if a similar molecular mechanism could be promoted in mammalian cells”.*

6. In the Western blot results, the total ATM and ATR controls were absent.

[Authors] The reviewer is totally right and we repeated all these experiments and generated a total of **8 new panels** only to answer to this specific point.

7. Authors may choose to include a graphical abstract at the end of their study to visually illustrate the mechanisms they have described.

[Authors] This was a very good idea and we generated a cartoon describing the main messages of our study as part of the “Synopsis”.

Reviewer #1 (Significance (Required)):

Advance: The authors aim to present a novel perspective on the resistance mechanisms to platinum compounds in NSCLC therapy. They explore platinum compounds-induced DNA damage, ATM activation, and MDM2-mediated stabilization of the active form of NOTCH (NICD). However, to strengthen their claims, they must provide more conclusive results.

Audience: This manuscript will likely engage oncologists who investigate the chemotherapy-resistant mechanisms of platinum compounds in NSCLC treatment, as well as scientists specializing in NOTCH and MDM2 pathways. However, the manuscript's central claims lack robust support from the available data, and the current approaches employed are not sufficiently thoughtful and rigorous; there is room for improvement.

My expertise is molecular medicine, cancer biology, and epigenetics.

[Authors] We want to thank again this reviewer for her/his helpful positive comments.

We are also very satisfied when she/he said: “This manuscript will likely engage oncologists who investigate the chemotherapy-resistant mechanisms of platinum compounds in NSCLC treatment”.

Reviewer #2 (Evidence, reproducibility and clarity (Required)):

In this manuscript, Sara Bernardo et al. investigated the molecular mechanisms underlying the activation of the Notch signaling in response to DNA damage induced by platinum-based chemotherapeutic agents in non-

small cell lung cancer (NSCLC). They demonstrated that carboplatin treatment induces DNA double-strand breaks (DSBs) and stabilizes NICD, a process dependent on ATM and mediated by MDM2. In vivo experiments in patient-derived xenografts (PDX) showed that inhibition of NICD and MDM2 enhanced platinum effectiveness. Furthermore, clinical analysis revealed a correlation between MDM2 expression and poor prognosis in NSCLC patients treated with platinum compounds, emphasizing the clinical relevance of the MDM2-NICD axis in platinum resistance.

[Authors] We thank this reviewer for her/his nice synopsis of our study.

Major comments:

Overall, the authors have conducted experiments that sufficiently elucidate their claims, and the description of the experiments is detailed. However, there is still room for the improvement.

[Authors] We are very pleased that reviewer finds our experimental work "...sufficiently elucidate their claims, and the description of the experiments is detailed."

We are also aware of her/his concerns but we are sure that after including all the new data she/he will be fully satisfied.

1.The finding that MDM2 promoted NICD stability through non degradative ubiquitination is interesting and in line with a previous study. As it is also known that NICD is regulated by various post-translational modifications, including ubiquitination that promotes NICD degradation. It is unclear what's the potential difference between these two types of ubiquitination. For example, do these two differ in specific ubiquitination sites? Can the authors provide some discussion?

[Authors] These are interesting points. As explained in detail in major point 1 from reviewer 1, we confirmed that MDM2 could use the ubiquitin mutant K48R for NICD ubiquitination and hence the MDM2-mediated ubiquitination does not promote degradation.

As for the specific ubiquitination sites in NICD, there are up to 17 lysine residues susceptible of being ubiquitinated. Hence unveiling what residues are targeted by MDM2 and if they differ from others inducing degradation as those promoted for instance by the E3 ligase FBXW7, is indeed very interesting but we feel is out of the scope of the current manuscript. Still, and as kindly proposed by the reviewer, we devoted a paragraph in line 382 of the discussion section that reads: "*We do not know either among the 17 lysins presented in NICD which is (are) targeted by MDM2, and if they differ from those recognized by E3 ligases that*

ubiquitinate NICD for degradation (Lai, 2002) (Welcker & Clurman, 2008). All these questions are under study currently in the laboratory”.

2. Could the overexpression of MDM2 or NICD lead to carboplatin resistance in A549 or H358 cells?

[Authors] This is a very interesting experiment and prompted by the reviewer’s comment we generated a new doxycycline inducible NICD lentiviral vector to produce a stable H358 cell line. Using this new cell line and the parental one as control, we tested different concentrations of carboplatin in the presence of doxycycline (that induced ectopic NICD expression) generating the **New Fig. 5A**. We observed that indeed increasing NICD protein levels reduced the sensitivity against carboplatin in the human H358 cell line, providing a rational for the *in vivo* inhibition that appears right after in the new version of our study. We want to acknowledge the reviewer for this experiment that further strength the message of our study, i.e., **increased levels of NICD in lung cancer cells protects them against platinum treatment**.

3. The trends observed in the western blot data within the manuscript appear inconsistent. While the authors propose that NICD levels increased upon incubation with carboplatin, the discrepancy arises when considering the NICD levels without cycloheximide (CHX) treatment in Fig. 1E, where no significant elevation is observed (Lane 6 vs. Lane 1).

[Authors] The point of the reviewer is well taken. Please bear in mind that in here we are handling several signaling pathways: DNA damage response, Notch, and MDM2, that interact among all of them while having each one different kinetics. Our finding of increased NICD upon carboplatin treatment is highly consistent *in vitro* and *in vivo*, but it is true that in the experiment mentioned by the reviewer was not so obvious, probably due to some kinetics issue. We repeated this particular experiment to have the same NICD increased upon carboplatin as it was in the rest of the manuscript (up to 9 times only in main Fig. in previous version) and generated **New Fig. 1E**. We want to thank the reviewer for unveiling us this inconsistency that could compromise the message of our study.

4. The quality of western blots needs to be improved, especially Fig. 1C and S1C, also Fig. 3B. Moreover, the NICD western blot sometimes appears as one band and sometimes as two bands. Please provide an explanation. If possible, please quantify the bands in western blots.

[Authors] There are several points here.

First, we agree with the reviewer that not all WB had the same quality and we repeated the ones kindly pointed out by the reviewer to homogenize the quality

all over the manuscript generating **New Fig. 1C** and **3B** as well as **New Fig. EV 1C**.

Regarding the two bands, NICD is tightly controlled at the post-transcriptional level beyond ubiquitination, including phosphorylation (Braunreiter & Cole, 2019), hydroxylation (Ferrante *et al*, 2022), acetylation (Guarani *et al*, 2011), or methylation (Hein *et al*, 2015) to name only a few, and hence it is not surprising that NICD might show different band sizes. In our experience we observed that the different bands (sometimes more than 2) are not associated to any particular condition, i.e., treatment or cell line, and hence it is something we plan to study in detail in a following study by proteomics. We feel that mentioning this particular aspect in the manuscript would complicate the message and we prefer to not mention it, but if the reviewer believes it is crucial, please let us know and we will do it.

Finally, and also as proposed by the reviewer, we quantified the bands in the experiments we felt it was more relevant, in particular in the Fig. dealing with ubiquitination, i.e., the **New Fig. 4B** and **4C**. Conversely, we think that quantifying all WB all over the manuscript will convolute the Fig. panels and we prefer to avoid it, but if the reviewer still feels it is completely required, we will perform and add them.

5. Please provide a necessary discussion on whether the targeted treatment approach towards the MDM2-NICD axis is applicable to all patients or only to those with high expression of MDM2/NICD.

[Authors] This is an interesting point and we added a comment in the discussion section regarding the need to stratify patients by its Notch activity. In particular we wrote in line 412: *“An important question arises regarding what subset of patients could benefit from Notch inhibition, and to answer it in our local hospital ICM, we are currently conducting a clinical trial aiming to monitor Notch activity in NSCLC patients: Alcina 2 cohort 11 (NCT04025541). Hence our clinical research should pave the way for future clinical trials aiming to treat NSCLC patients using Notch inhibitors”*.

6. How to interpret the significance of the simultaneous increase in NICD ubiquitination and stability mediated by MDM2? Please provide a relevant discussion.

[Authors] We provided strong experimental data going beyond discussion in **New Fig. 4C** using the ubiquitin mutant K48R. Furthermore, as kindly proposed by the reviewer we also provided a new paragraph in the discussion section. In particular we wrote in line 379: *“Likewise, our data is in accordance with a seminal study also showing that MDM2 ubiquitinates NICD but not for degradation (Pettersson et al., 2013). Conversely, at this stage we cannot know if MDM2 promotes mono- or poly-ubiquitination on NICD. We do not know either among the 17 lysins*

presented in NICD which is (are) targeted by MDM2, and if they differ from those recognized by E3 ligases that ubiquitinate NICD for degradation (Lai, 2002) (Welcker & Clurman, 2008). All these questions are under study currently in the laboratory”.

7. In Fig. 5B, please also check the level of MDM2. In Fig. 5C, carboplatin appears to have little impact on tumor growth. How to explain the increase of Ki-67 in the carboplatin treatment group in Fig. 5A?

[Authors] There are several points here.

First and as requested by the reviewer in the new *in vivo* experiment we measured the levels of MDM2 by WB. In **New Fig. 5E** we confirmed the increase in MDM2 upon carboplatin treatment we observed in our previous version by IHC. Conversely, we could not perform the MDM2 IHC stain in the new experiment because we could not find the required antibody. So, we are in debt with the reviewer for this critical point that allow us to maintain our original message that **carboplatin induces MDM2 expression also *in vivo***. If reviewer feels that the IHC data of MDM2 is crucial, please let us know and we will try to find the antibody from collaborators beyond this manuscript and perform the stain and quantification.

Indeed, as the reviewer kindly mentioned, carboplatin has little impact on tumor growth in this particular PDX model as already published by our collaborators and authors of this study (Ferrer *et al.*, 2018). This is why we chose this particular model with intrinsic carboplatin resistance, to proof that the combination of carboplatin with Notch or MDM2 inhibitors regain sensitivity to platinum treatment.

As for the interesting observation of the Ki67, honestly, we cannot provide at this moment any good explanation. Conversely, we previously observed a higher growth in gefitinib resistant EGFR genetic engineered mouse models when treated with this drug (Bousquet Mur *et al.*, 2020), hence we believe that resistant cells can be boosted by the drug against became resistant in some circumstances. In any case and as mentioned above, for panel simplicity and homogenization with the WB data presented in the current version we decided to don't perform Ki67 IHC stain.

Minor comments:

1. Please include scale bars in Fig. 1B and Supplemental Fig. 1B.

[Authors] We thank the reviewer for this comment. We included the scale bars as requested.

2. Fig. 5D, the P values of the survival curve should be indicated in the Figs.

[Authors] Since we have increase the survival curves, we decided to use only asterisks indicating the statistical significance in **new Fig. 5E**. If the reviewer prefers we show the exact p value, we will include it, please let us know.

3.The presentation of survival curve data in Fig.s 5D and 6A should be consistent.

[Authors] The point of the reviewer is well taken and we used Prism to draw the PFS for patients in **New Fig. 6A** to homogenize both Fig. as kindly proposed by the reviewer.

4.It seems that supplemental Fig. 2 is missing.

[Authors] We actually jumped from 1 to 3 because we did not have any associated supplemental Fig. to main Fig. 2. Now and following the editorial policy of EMBO Mol Med, we present from EV1 to 3 and we want to thank the reviewer for highlighting this point.

5.Please carefully check the spelling of the entire text, for example, on page 20, line 426 it should be 'western'. Also, please spell out the abbreviations DDR and ATM.

[Authors] We double checked all spelling and to the best of our knowledge it should be good. We also included the abbreviations kindly suggested by the reviewer the first time they appeared in lines 44 to 46.

6.The abbreviation for Cleaved caspase 3 should be CC3.

[Authors] We thank the reviewer for this information.

Reviewer #2 (Significance (Required)):

Notch signaling is associated with the occurrence and development of non-small cell lung cancer (NSCLC). Previous study indicates that the expression of Notch protein is significantly higher in NSCLC tissues compared to normal

tissues (PMID: 31170211). Additionally, the upregulation of Notch1 is correlated with higher tumor grades, lymph node metastasis, tumor-node-metastasis (TNM) staging, and poor prognosis (PMID: 25996086). Abnormal activation of Notch signaling pathway is frequently observed in chemotherapy-resistant NSCLC, and some studies have aimed to address NSCLC drug resistance via modulating Notch signaling (PMID: 30087852, 38301911). This manuscript firstly proposes that MDM2-mediated stabilization of NICD upon DNA damage plays a major role in NSCLC response to platinum chemotherapy. It further suggests that targeting the MDM2-NICD axis could prove to be an effective therapeutic strategy. Overall, this work unveils a novel mechanism for Notch activation in response to platinum chemotherapy, providing a renewed outlook on overcoming chemotherapy resistance in NSCLC. This manuscript will attract those interested in the mechanisms of chemotherapy resistance and novel treatment approaches.

[Authors] We sincerely thank the reviewer for finding that our "...work unveils a novel mechanism for Notch activation in response to platinum chemotherapy, providing a renewed outlook on overcoming chemotherapy resistance in NSCLC". We are also very satisfied when she/he says: "This manuscript will attract those interested in the mechanisms of chemotherapy resistance and novel treatment approaches."

We want to thank again both reviewers for helping us to make our study stronger and improving the impact and significance while keeping the original message.

REFERENCES

- Arena G, Cisse MY, Pyrdziak S, Chatre L, Riscal R, Fuentes M, Arnold JJ, Kastner M, Gayte L, Bertrand-Gaday C *et al* (2018) Mitochondrial MDM2 Regulates Respiratory Complex I Activity Independently of p53. *Mol Cell* 69: 594-609 e598
- Bousquet Mur E, Bernardo S, Papon L, Mancini M, Fabbriozzi E, Goussard M, Ferrer I, Giry A, Quantin X, Pujol JL *et al* (2020) Notch inhibition overcomes resistance to tyrosine kinase inhibitors in EGFR-driven lung adenocarcinoma. *J Clin Invest* 130: 612-624
- Braunreiter KM, Cole SE (2019) A tale of two clocks: phosphorylation of NICD by CDKs links cell cycle and segmentation clock. *EMBO Rep* 20: e48247
- Cheng Q, Cross B, Li B, Chen L, Li Z, Chen J (2011) Regulation of MDM2 E3 ligase activity by phosphorylation after DNA damage. *Mol Cell Biol* 31: 4951-4963
- Ferrante F, Giaimo BD, Friedrich T, Sugino T, Mertens D, Kugler S, Gahr BM, Just S, Pan L, Bartkuhn M *et al* (2022) Hydroxylation of the NOTCH1 intracellular domain regulates Notch signaling dynamics. *Cell Death Dis* 13: 600
- Ferrer I, Quintanal-Villalonga A, Molina-Pinelo S, Garcia-Heredia JM, Perez M, Suarez R, Ponce-Aix S, Paz-Ares L, Carnero A (2018) MAP17 predicts sensitivity to platinum-based therapy, EGFR inhibitors and the proteasome inhibitor bortezomib in lung adenocarcinoma. *J Exp Clin Cancer Res* 37: 195
- Guarani V, Deflorian G, Franco CA, Kruger M, Phng LK, Bentley K, Toussaint L, Dequiedt F, Mostoslavsky R, Schmidt MHH *et al* (2011) Acetylation-dependent

regulation of endothelial Notch signalling by the SIRT1 deacetylase. *Nature* 473: 234-238

Hein K, Mittler G, Cizelsky W, Kuhl M, Ferrante F, Liefke R, Berger IM, Just S, Strang JE, Kestler HA *et al* (2015) Site-specific methylation of Notch1 controls the amplitude and duration of the Notch1 response. *Sci Signal* 8: ra30

Jevitt A, Huang YC, Zhang SM, Chatterjee D, Wang XF, Xie GQ, Deng WM (2021) Modeling Notch-Induced Tumor Cell Survival in the Drosophila Ovary Identifies Cellular and Transcriptional Response to Nuclear NICD Accumulation. *Cells* 10

Lai EC (2002) Protein degradation: four E3s for the notch pathway. *Curr Biol* 12: R74-78

Li S, Ren B, Shi Y, Gao H, Wang J, Xin Y, Huang B, Liao S, Yang Y, Xu Z *et al* (2019) Notch1 inhibition enhances DNA damage induced by cisplatin in cervical cancer. *Exp Cell Res* 376: 27-38

Linares LK, Hengstermann A, Ciechanover A, Muller S, Scheffner M (2003) HdmX stimulates Hdm2-mediated ubiquitination and degradation of p53. *Proc Natl Acad Sci U S A* 100: 12009-12014

Maya R, Balass M, Kim ST, Shkedy D, Leal JF, Shifman O, Moas M, Buschmann T, Ronai Z, Shiloh Y *et al* (2001) ATM-dependent phosphorylation of Mdm2 on serine 395: role in p53 activation by DNA damage. *Genes Dev* 15: 1067-1077

McAuliffe SM, Morgan SL, Wyant GA, Tran LT, Muto KW, Chen YS, Chin KT, Partridge JC, Poole BB, Cheng KH *et al* (2012) Targeting Notch, a key pathway for ovarian cancer stem cells, sensitizes tumors to platinum therapy. *Proc Natl Acad Sci U S A* 109: E2939-2948

Petroski MD, Deshaies RJ (2005) Mechanism of lysine 48-linked ubiquitin-chain synthesis by the cullin-RING ubiquitin-ligase complex SCF-Cdc34. *Cell* 123: 1107-1120

Pettersson S, Sczaniecka M, McLaren L, Russell F, Gladstone K, Hupp T, Wallace M (2013) Non-degradative ubiquitination of the Notch1 receptor by the E3 ligase MDM2 activates the Notch signalling pathway. *Biochem J* 450: 523-536

Riscal R, Schrepfer E, Arena G, Cisse MY, Bellvert F, Heuillet M, Rambow F, Bonneil E, Sabourdy F, Vincent C *et al* (2016) Chromatin-Bound MDM2 Regulates Serine Metabolism and Redox Homeostasis Independently of p53. *Mol Cell* 62: 890-902

Weber D, Wiese C, Gessler M (2014) Hey bHLH transcription factors. *Curr Top Dev Biol* 110: 285-315

Welcker M, Clurman BE (2008) FBW7 ubiquitin ligase: a tumour suppressor at the crossroads of cell division, growth and differentiation. *Nat Rev Cancer* 8: 83-93

Weng AP, Millholland JM, Yashiro-Ohtani Y, Arcangeli ML, Lau A, Wai C, Del Bianco C, Rodriguez CG, Sai H, Tobias J *et al* (2006) c-Myc is an important direct target of Notch1 in T-cell acute lymphoblastic leukemia/lymphoma. *Genes Dev* 20: 2096-2109

Zhang H, Jiang H, Chen L, Liu J, Hu X, Zhang H (2018) Inhibition of Notch1/Hes1 signaling pathway improves radiosensitivity of colorectal cancer cells. *Eur J Pharmacol* 818: 364-370

3rd Jun 2025

Dear Dr. Maraver,

Thank you for submitting your manuscript to EMBO Molecular Medicine.

We have contacted the referees who reviewed your manuscript for Review Commons. Unfortunately, referee #2 was unavailable to evaluate your revised manuscript; however, referee #1 has agreed to review your responses to their comments as well.

As you will see below, while referee #1 appreciates the work that has been done, s/he also has a few remaining concerns that should be addressed for your manuscript to be considered further by EMM.

As EMBO Press usually only allows one round of revisions, please be aware that this will be your last opportunity to address these issues. The revised manuscript will be reviewed again, and we cannot guarantee a positive outcome at this stage.

Moreover, please address the following editorial requests:

1. Please note that an email bounced for Maicol Mancini (maicol.mancini@weizmann.ac.il). There is a name discrepancy between Celine Bouclier (submission system) vs. Céline Bouclier (manuscript title page)
2. Please provide up to 5 keywords.
3. The figures should be removed from the manuscript text and uploaded as individual high resolution figure files. The legends should be compiled at the end of the manuscript text, after the References. The EV figures should also be uploaded as individual figure files and the legends added after the main figure legends, under the heading "Expanded View Figure Legends".
4. It is mandatory to include a 'Data Availability' section after the Materials and Methods. Before submitting your revision, primary datasets produced in this study need to be deposited in an appropriate public database, and the accession numbers and database listed under 'Data Availability'. Please remember to provide a reviewer password if the datasets are not yet public (see <https://www.embopress.org/page/journal/17574684/authorguide#dataavailability>). In case you have no data that requires deposition in a public database, please state so in this section. Note that the Data Availability Section is restricted to new primary data that are part of this study.
5. Please make sure that the funding information provided in the manuscript is identical to the information provided in the submission system (the French Ministry of Education and Research fellowship, FIMA, CIBERONC (CB16/12/00443), Spanish Ministry of Economy and Innovation and Fondo de Investigación Sanitaria Fondo Europeo de Desarrollo Regional (PI22/00451) are not entered in the submission system).
6. Disclosure statement and competing interests: We updated our journal's competing interests policy in January 2022 and request authors to consider both actual and perceived competing interests. Please review the policy <https://www.embopress.org/competing-interests> and update your competing interests if necessary.
7. The paper explained: EMBO Molecular Medicine articles are accompanied by a summary of the articles to emphasize the major findings in the paper and their medical implications for the non-specialist reader. Please provide a draft summary of your article highlighting
 - the medical issue you are addressing,
 - the results obtained and
 - their clinical impact.

8. Please upload the source data as one ZIP file per figure. Fig 1 and EV1 should be uploaded separately. We don't accept powerpoint files, please upload them in a different format.

9. All Materials and Methods need to be described in the main text using our 'Structured Methods' format. According to this format, the Methods section includes a Reagents and Tools Table (listing key reagents, experimental models, software and relevant equipment and including their sources and relevant identifiers) followed by a Methods and Protocols section describing the methods, ideally using a step-by-step protocol format. The aim is to facilitate adoption of the methodologies across labs. Please download and fill our Reagents and Tools Table template (.docx), which you can find in our author guidelines: <https://www.embopress.org/page/journal/14693178/authorguide#structuredmethods>.

10. The synopsis image should be uploaded as a separate file.

11. Please make sure that all figures/figure panels are referenced in the manuscript. Currently, a callout is missing for Fig 5D; "Fig." should be added for Fig EV3A; a callout for Fig EV3B is missing.

12. The reference to BIORENDER should be removed from the acknowledgements and moved to Methods using the following format:

Graphics:

(some of the... OR Figure #... OR synopsis) Graphics were created with BioRender.com.

13. Figure reuse must be mentioned in the figure legends (Figure 1D and Figure EV1D).

14. Please provide exact p values (missing for figures 5A, D; EV3 C).

As part of the EMBO Publications transparent editorial process initiative (see our Editorial at <http://embomolmed.embopress.org/content/2/9/329>), EMBO Molecular Medicine will publish online a Review Process File (RPF) to accompany accepted manuscripts.

This file will be published in conjunction with your paper and will include the anonymous referee reports, your point-by-point response and all pertinent correspondence relating to the manuscript. Let us know whether you agree with the publication of the RPF and as here, if you want to remove or not any figures from it prior to publication.

I look forward to receiving your revised manuscript.

Sincerely,

Lise Roth

***** Reviewer's comments *****

Referee #1 (Remarks for Author):

Overall comment:

The reviewer appreciates the authors' efforts to revise and improve the manuscript in response to the previous suggestions. Most concerns have been adequately addressed, and the manuscript is close to being suitable for publication. However, to meet the standards of EMBO Molecular Medicine, we request that the authors address the additional comments and perform the suggested experiments listed below to further strengthen the manuscript.

Specific comments:

1. The authors mentioned monitoring NICD half-life multiple times throughout the manuscript, including under carboplatin treatment with and without MDM2 knockdown. While the effects of both carboplatin and MDM2 knockdown on NICD protein stability are evident, a single experiment may not sufficiently validate changes in NICD half-life across different conditions. We recommend quantifying the data presented in Figures 1E, 2C, 2D, 3D, EV1E, and EV2C using at least three biological replicates for statistical robustness. Additionally, NICD half-life should be measured by fitting degradation curves, as described in PMID: 34626566. Figure 1E should include the 4-hour time point to maintain consistency with Figures 2C and 2D.
2. Figures EV1E and EV2B should be repeated and replaced due to the appearance of unexpected NICD bands-specifically at the 10-hour vehicle treatment in Fig. EV1E and at the 0-hour siMDM2 condition in Fig. EV2B. Interestingly, these results suggest the presence of two NICD isoforms, with carboplatin potentially affecting their migration or modification state. Similar lower- and higher-migrating NICD forms are also observed under the 0-hour ATMi condition in Fig. 2C and the MDM2-WT expression condition in Fig. 4A, respectively. Collectively, these observations support the hypothesis that the higher-migrating NICD form is associated with NICD protein stabilization and may be closely linked to the carboplatin- ATM-MDM2 regulatory axis. These observations may be valuable to discuss in the discussion section.
3. Since this manuscript focuses on protein stability regulation, we strongly recommend including molecular weight markers in all immunoblots to enhance interpretability.
4. The authors added a sentence referencing an ongoing clinical trial based on their current findings: "An important question arises regarding what subset of patients could benefit from Notch inhibition, and to answer it in our local hospital ICM, we are currently conducting a clinical trial aiming to monitor Notch activity in NSCLC patients: Alcina 2 cohort 11 (NCT04025541). Hence our clinical research should pave the way for future clinical trials aiming to treat NSCLC patients using Notch inhibitors". It would strengthen the revised manuscript if the authors provided more detail on how Notch activity is being monitored in NSCLC patients as part of this clinical trial.
5. There are a few suggestions regarding the revised Figure 5. Since the authors mentioned significant weight loss in mice during treatment, it would be helpful to include mouse weight data (as supplementary material) to assess the potential toxicity of the drug combinations. Although there was a statistically significant increase in the γ H2AX H-score in the carboplatin + DBZ group compared to the vehicle group, the difference appears modest. Moreover, there seems to be no significant difference between the carboplatin monotherapy group and either the carboplatin + DBZ or the carboplatin + SP141 combination groups.

These findings suggest that the therapeutic efficacy of the combination treatments may not be fully attributable to DNA damage. Interestingly, the SP141 + DBZ combination significantly reduced the γ H2AX Hscore. Please consider explaining these observations and discussing their implications in the discussion section. Additionally, the H-scores of CC3 and Ki67 could provide valuable insights to explain these discrepancies and may be worth including in the revised manuscript.

6. It would also be beneficial to quantify the bands in the western blot shown in Figures 3B and 5E. In Fig. 5E, MDM2 levels appear elevated in both the carboplatin monotherapy and the carboplatin + SP141 groups, but decreased in the SP141 + DBZ group compared to the vehicle. Please explain the observed changes in MDM2 expression across the different treatment groups.

7. Although the current manuscript is generally understandable, there is still room to enhance its clarity and appeal to readers prior to publication. Additionally, some grammatical errors remain. We recommend that the authors improve the language of the manuscript.

Minor comments:

1. In line 201, it is recommended to replace "required" with "critical," as the S356A-MDM2-mediated ubiquitination of NICD shown in Figure 4A is significantly reduced but not completely abolished.
2. In lines 279-281, the sentence "We measured tumor growth in all mice until the first one in each group reached 279 the human end point (tumor volume of 1500 mm³), that it was 25 for vehicle, 42 for the 280 carboplatin-DBZ group and 32 for the rest" may be confusing to readers, as it does not specify that the numbers refer to days. Please clarify the sentence by explicitly indicating that these numbers represent the number of days required to reach the endpoint in each group.
3. The graphical abstract is clear and suitable for inclusion in the main manuscript. It is suggested to present it as Figure 6C.

We sincerely thank the Editors and the Reviewer for the time dedicated to our manuscript in this second round of revision.

We answered to all the new comments and added new experimental data. In total we included 14 new panels in this second revision, that together with the previous 23 from the first round, give a total of 37 new panels, witnessing the enormous work we devoted to the current study.

Reviewer #1 Overall comment:

The reviewer appreciates the authors' efforts to revise and improve the manuscript in response to the previous suggestions. Most concerns have been adequately addressed, and the manuscript is close to being suitable for publication. However, to meet the standards of EMBO Molecular Medicine, we request that the authors address the additional comments and perform the suggested experiments listed below to further strengthen the manuscript.

[Authors]

We appreciate that the Reviewer acknowledged our efforts in revising the study. Furthermore, by addressing all new comments, we sincerely hope that the Reviewer will concur with us that our study is ready for publication in EMBO Mol Med.

Conversely, the inclusion of new experiments not related to any of the Reviewer's comments in the first revision was unusual. We refer here only to comment 1. We discussed with Dr Lisa Roth, senior editor at EMBO Mol Med and arrived to a consensus regarding this particular point (please see below). We would like to thank Dr Roth and all the editorial team for their work and help during the revision of our study.

Specific comments:

1. The authors mentioned monitoring NICD half-life multiple times throughout the manuscript, including under carboplatin treatment with and without MDM2 knockdown. While the effects of both carboplatin and MDM2 knockdown on NICD protein stability are evident, a single experiment may not sufficiently validate changes in NICD half-life across different conditions. We recommend quantifying the data presented in Figures 1E, 2C, 2D, 3D, EV1E, and EV2C using at least three biological replicates for statistical robustness. Additionally, NICD half-life should be measured by fitting degradation curves, as described in PMID: 34626566. Figure 1E should include the 4-hour time point to maintain consistency with Figures 2C and 2D.

[Authors]

These are appealing experiments, but as mentioned above in the first revision none of the two Reviewers mentioned anything about this otherwise interesting issue. After discussion with Dr Lise Roth, we arrived to a compromise in which we agreed to perform this set of experiments in triplicate only for the H358 cell line and in duplicate for the A549 cell line (see below). We used the study mentioned by the Reviewer to generate the degradation curves for H358 cells (Li *et al*, 2021), and we followed a second study to statistically compare the curves (Hristova & Wimley, 2023). Furthermore, the Reviewer wanted us to repeat the Figure 1E with the 4-hour time point (also not mentioned in first revision by any of the reviewers), and therefore, we performed three experiments for this panel. As previously mentioned, in A549 cells we repeated the three conditions to show a replicate for all the figures related to this cell line as requested by the Reviewer, and in one condition (previous EV1E), we repeated the experiment twice to answer to comment 2 (see below).

To put our work in context, only for this comment we performed a total of 11 experiments, each one using two different 6-well plates and generated: **new Fig. 1E**, **new Fig. 1F**, **new Fig. 2B**, **new Fig. 2C**, **new Fig. 3D**, **new Fig. 3E**, **new Fig. EV1E**, **new Fig. EV2C** and **new Fig. EV2E**.

Importantly, the main message of our study did not change, i.e., carboplatin increases NICD stability and this is ATM- and MDM2-mediated. However, we agree with the Reviewer that now our message is more robust and we thank the Reviewer for this.

2. Figures EV1E and EV2B should be repeated and replaced due to the appearance of unexpected NICD bands-specifically at the 10-hour vehicle treatment in Fig. EV1E and at the 0-hour siMDM2 condition in Fig. EV2B. Interestingly, these results suggest the presence of two NICD isoforms, with carboplatin potentially affecting their migration or modification state. Similar lower- and higher-migrating NICD forms are also observed under the 0-hour ATMi condition in Fig. 2C and the MDM2-WT expression condition in Fig. 4A, respectively. Collectively, these observations support the hypothesis that the higher-migrating NICD form is associated with NICD protein stabilization and may be closely linked to the carboplatin- ATM-MDM2 regulatory axis. These observations may be valuable to discuss in the discussion section.

[Authors]

There are several points here.

First, the Reviewer asked us to repeat the figures EV1E and EV2B and accordingly, we generated the **new Fig. EV1E** and **new Fig. EV2D**. As mentioned in comment 1, we did it in duplicate for each condition.

Regarding the different NICD sizes, in previous revision it was already mentioned by Reviewers. At this moment, we felt that discussing this issue could confuse the message, and hence we discussed it only to them and not include it in the manuscript. Upon reading this comment again from the Reviewer, we imagine she/he would like we discuss it in the manuscript. Therefore, now we included a sentence in the Discussion section, line 386, that reads: *“Moreover, NICD is tightly controlled at the post-transcriptional level, not only through ubiquitination, but also through, for instance, phosphorylation (Braunreiter & Cole, 2019), hydroxylation (Ferrante et al, 2022), acetylation (Guarani et al, 2011) and methylation (Hein et al, 2015). Therefore, it is not surprising to detect NICD bands of different sizes. In our experience, these different bands (sometimes >2) are not associated with any particular condition (e.g., treatment or cell line), and we plan to study this issue in detail using proteomics.”*

3. Since this manuscript focuses on protein stability regulation, we strongly recommend including molecular weight markers in all immunoblots to enhance interpretability.

[Authors]

The point of the Reviewer is well taken and we included the MWs in all required figures.

4. The authors added a sentence referencing an ongoing clinical trial based on their current findings: "An important question arises regarding what subset of patients could benefit from Notch inhibition, and to answer it in our local hospital ICM, we are currently conducting a clinical trial aiming to monitor Notch activity in NSCLC patients: Alcina 2 cohort 11 (NCT04025541). Hence our clinical research should pave the way for future clinical trials aiming to treat NSCLC patients using Notch inhibitors". It would strengthen the revised manuscript if the authors provided more detail on how Notch activity is being monitored in NSCLC patients as part of this clinical trial.

[Authors]

We thank the Reviewer for this comment. The biomarker used is HES1, and it is working well for the ongoing trials at ICM (our cancer hospital) in patients with EGFR-driven and KRAS-driven lung adenocarcinoma treated with EGFR and KRAS^{G12C} targeted therapies, but it will not for patients treated with carboplatin. As mentioned in the rebuttal letter of our first revision, the anti-HEY1 antibody that nicely works for immunoblotting, does not work for immunohistochemistry in our hands, therefore we will try with other commercial antibodies. In any case, we added another sentence at the end of the sentence mentioned by the Reviewer and now (line 418), the whole paragraph reads: *“An important question now is how to identify the patients who could benefit from Notch inhibition. To this aim,*

at our cancer center (ICM), we are currently conducting a clinical trial to monitor Notch activity in KRAS^{G12C}- and EGFR-mutated patients with NSCLC using HES1 as a read-out for the Notch pathway (Alcina 2 cohort 11; NCT04025541). Therefore, our clinical research could pave the way for future clinical trials to test the effect of Notch inhibitors in patients with NSCLC treated with targeted therapy. By contrast, in patients with NSCLC treated with Carboplatin, we would need to use another surrogate marker of Notch activation, for instance HEY1.”

We are in debt with the Reviewer for this suggestion.

5. There are a few suggestions regarding the revised Figure 5. Since the authors mentioned significant weight loss in mice during treatment, it would be helpful to include mouse weight data (as supplementary material) to assess the potential toxicity of the drug combinations. Although there was a statistically significant increase in the γ H2AX H-score in the carboplatin + DBZ group compared to the vehicle group, the difference appears modest. Moreover, there seems to be no significant difference between the carboplatin monotherapy group and either the carboplatin + DBZ or the carboplatin + SP141 combination groups. These findings suggest that the therapeutic efficacy of the combination treatments may not be fully attributable to DNA damage. Interestingly, the SP141 + DBZ combination significantly reduced the γ H2AX Hscore. Please consider explaining these observations and discussing their implications in the discussion section. Additionally, the H-scores of CC3 and Ki67 could provide valuable insights to explain these discrepancies and may be worth including in the revised manuscript.

[Authors]

There are several points here.

Regarding the weight loss of animals, we generated **new Fig. EV4B** to show weight measurements throughout the treatment period.

For the second point, we agree with the Reviewer that is difficult to know whether the higher γ H2AX H-score is enough to explain the increased survival and decreased PDX tumor growth in the carboplatin + DBZ combination group. We also agree that the γ H2AX H-score of carboplatin + SP141 is not significant, but interestingly, the H-score difference is significant between the vehicle and carboplatin + DBZ combination groups, mirroring what happened with tumor growth: the tumor growth curve of the carboplatin + SP141 group is between those of the vehicle and carboplatin + DBZ combination groups and the difference was not significant. Furthermore, thanks to the Reviewer's comment on the quantification of CC3 and Ki67 signals, we observed that CC3 expression was increased only in the carboplatin + DBZ combination group compared with the other groups, while Ki67 did not change (**new Fig. 5C**, **new Fig. EV4C** and **new Fig. EV5A**). These new data indicate that there is an increase in apoptosis, and

hence, the Reviewer was totally right by saying that CC3 and Ki67 expression could help to understand the phenotypes promoted by the carboplatin + DBZ combination and we are in debt for this comment. Still, we cannot know whether apoptosis is mediated only by DNA damage, and following this Reviewer's comment, we included a sentence in the Discussion before a paragraph already presented in previous version, now in line 404: "*As preclinical in vivo model, we used nude mice xenografted with a NSCLC PDX that displays intrinsic resistance against platinum. Accordingly, in these mice, treatment with Carboplatin alone did not increase γ -H2AX expression in tumor cells compared with vehicle, unlike what observed after treatment with Carboplatin+DBZ (Notch inhibitor). Therefore, although at this stage we cannot rule out that other mechanisms besides DNA damage are applying in our experimental system, our data are in accordance with previous studies on ovarian and cervical cancers showing that NICD inhibition enhances DNA damage upon platinum treatment (Li et al, 2019; McAuliffe et al, 2012). Moreover, a study on colorectal cancer found that NICD inhibition increases radiation-induced DNA damage (Zhang et al, 2018). In addition, using a *D. melanogaster* ovarian tumor model, a previous study showed that NICD ectopic expression promotes the upregulation of double strand break repair genes (Jevitt et al, 2021). We are investigating whether a similar molecular mechanism could occur in mammalian cells.*"

Lastly, the finding that in the SP141 + DBZ group, γ H2AX expression is lower than in the carboplatin-treated groups is not surprising because carboplatin is the source of DNA damage. In agreement, this group did not show any therapeutic benefit. However, thanks to this comment, we realized that the statistical results in the γ H2AX figure were confusing and we remade it highlighting only the significant parts. We thank the Reviewer for helping us to clarify this particular point.

6. It would also be beneficial to quantify the bands in the western blot shown in Figures 3B and 5E. In Fig. 5E, MDM2 levels appear elevated in both the carboplatin monotherapy and the carboplatin + SP141 groups, but decreased in the SP141 + DBZ group compared to the vehicle. Please explain the observed changes in MDM2 expression across the different treatment groups.

[Authors]

We thank the reviewer for these suggestions.

We think that including the quantification for Fig 3B would require the quantification of all western blots of the study. This would complicate massively all the figures because it will require to decrease the size of panels in EV figures, thus for clarity we will not do it.

Conversely, as the western blot in Fig. 5B displays data from *in vivo* tumors and we have immunohistochemistry quantification in the same figure, it is easier to justify the quantification of the bands in this blot. Furthermore, there is an internal

duplicate, i.e., 2 lines per treatment, and we quantified all of them generating the new **Fig. EV5B**.

As for MDM2 expression changes in the different treatment groups, we included a sentence at the end of an existing sentence in Results, and now it reads in line 314: *“Lastly, Carboplatin increased MDM2 protein levels (Figs. 5E and EV5B), confirming our in vitro data, whereas Notch inhibition (DBZ) decreased this effect, indicating that a feed forward loop could exist between NICD and MDM2.”*

7. Although the current manuscript is generally understandable, there is still room to enhance its clarity and appeal to readers prior to publication. Additionally, some grammatical errors remain. We recommend that the authors improve the language of the manuscript.

[Authors]

The new version has been fully revised by Elisabetta Andermarcher, that professionally edit scientific articles. Although majority were minor changes, the number was high and in order to help both the reviewer and editors we decided to don't highlight them, and only mark in red the scientific comments in the new “manuscript” file.

Minor comments:

1. In line 201, it is recommended to replace "required" with "critical," as the S356A-MDM2mediated ubiquitination of NICD shown in Figure 4A is significantly reduced but not completely abolished.

[Authors]

The point of the Reviewer concerning the S395 mutant is well taken and we changed, in current line 197, the word “required” to “plays a critical role”, which we prefer than using “critical” alone.

2. In lines 279-281, the sentence "We measured tumor growth in all mice until the first one in each group reached the human end point (tumor volume of 1500 mm³), that it was 25 for vehicle, 42 for the carboplatin-DBZ group and 32 for the rest" may be confusing to readers, as it does not specify that the numbers refer to days. Please clarify the sentence by explicitly indicating that these numbers represent the number of days required to reach the endpoint in each group.

[Authors]

As requested by the Reviewer we changed the sentence, line 268, as follows: *“We measured tumor growth in all mice until the first animal in each group*

reached the end point (tumor volume of 1500 mm³): day 25 for the vehicle group, day 42 for the carboplatin+DBZ group, and day 32 for the other groups.”

3. The graphical abstract is clear and suitable for inclusion in the main manuscript. It is suggested to present it as Figure 6C.

[Authors]

We thank the Reviewer by acknowledging the clarity and suitability of our graphical abstract. The editorial policy of EMBO Mol Med requires that this part goes separately and we will follow their rules.

We want to thank again the Reviewer for helping us to make our study more solid while keeping the original message.

REFERENCES

- Braunreiter KM, Cole SE (2019) A tale of two clocks: phosphorylation of NICD by CDKs links cell cycle and segmentation clock. *EMBO Rep* 20: e48247
- Ferrante F, Giaimo BD, Friedrich T, Sugino T, Mertens D, Kugler S, Gahr BM, Just S, Pan L, Bartkuhn M *et al* (2022) Hydroxylation of the NOTCH1 intracellular domain regulates Notch signaling dynamics. *Cell Death Dis* 13: 600
- Guarani V, Deflorian G, Franco CA, Kruger M, Phng LK, Bentley K, Toussaint L, Dequiedt F, Mostoslavsky R, Schmidt MHH *et al* (2011) Acetylation-dependent regulation of endothelial Notch signalling by the SIRT1 deacetylase. *Nature* 473: 234-238
- Hein K, Mittler G, Cizelsky W, Kuhl M, Ferrante F, Liefke R, Berger IM, Just S, Strang JE, Kestler HA *et al* (2015) Site-specific methylation of Notch1 controls the amplitude and duration of the Notch1 response. *Sci Signal* 8: ra30
- Hristova K, Wimley WC (2023) Determining the statistical significance of the difference between arbitrary curves: A spreadsheet method. *PLoS One* 18: e0289619
- Jevitt A, Huang YC, Zhang SM, Chatterjee D, Wang XF, Xie GQ, Deng WM (2021) Modeling Notch-Induced Tumor Cell Survival in the Drosophila Ovary Identifies Cellular and Transcriptional Response to Nuclear NICD Accumulation. *Cells* 10
- Li J, Cai Z, Vaites LP, Shen N, Mitchell DC, Huttlin EL, Paulo JA, Harry BL, Gygi SP (2021) Proteome-wide mapping of short-lived proteins in human cells. *Mol Cell* 81: 4722-4735 e4725
- Li S, Ren B, Shi Y, Gao H, Wang J, Xin Y, Huang B, Liao S, Yang Y, Xu Z *et al* (2019) Notch1 inhibition enhances DNA damage induced by cisplatin in cervical cancer. *Exp Cell Res* 376: 27-38
- McAuliffe SM, Morgan SL, Wyant GA, Tran LT, Muto KW, Chen YS, Chin KT, Partridge JC, Poole BB, Cheng KH *et al* (2012) Targeting Notch, a key pathway for ovarian cancer stem cells, sensitizes tumors to platinum therapy. *Proc Natl Acad Sci U S A* 109: E2939-2948
- Zhang H, Jiang H, Chen L, Liu J, Hu X, Zhang H (2018) Inhibition of Notch1/Hes1 signaling pathway improves radiosensitivity of colorectal cancer cells. *Eur J Pharmacol* 818: 364-370

6th Nov 2025

Dear Dr. Maraver, Dear Antonio,

Thank you for submitting your revised study. As you will see below, referee #1 is satisfied with the revisions, and I will therefore be able to accept your manuscript once the following editorial concerns are addressed:

1/ Manuscript text:

- Please remove the red font text and indicate in track changes mode any new modification in the text.
- I introduced minor modifications in your abstract, please carefully check and let me know if you agree or amend as you see fit: "Despite major advances in the clinical management of non-small cell lung carcinoma (NSCLC), most patients treated with first-line platinum-based chemotherapy combined with immune checkpoint inhibitors will relapse, which constitutes an unmet medical need. Here, we found that various DNA damage inducers increase the levels of Notch Intracellular Domain (NICD), the active form of NOTCH1. Mechanistically, we revealed that, upon platinum treatment, the expression levels of both MDM2 and NICD were increased and that MDM2 stabilised NICD through ubiquitination. Using NSCLC patient-derived xenografts displaying intrinsic carboplatin resistance, we demonstrated that combining carboplatin with a γ -secretase inhibitor, which hinders NICD generation, significantly improves survival and reduces tumour growth compared with carboplatin monotherapy. Furthermore, in patients with NSCLC who received platinum-based chemotherapy, the level of MDM2 expression in the tumour correlated with poor progression-free survival, which further validates the key role of MDM2 in response to platinum compounds. Our findings present a new therapeutic opportunity for patients with NSCLC, the most common form of lung cancer and the leading cause of cancer-related death worldwide."
- Materials and protocols should be renamed Methods:
 - o Cells: please indicate whether the cells were authenticated.
 - o Mice: please provide a statement on housing and husbandry conditions.
 - o Human samples: for each cohort, please include a statement confirming that the experiments conformed to the principles set out in the WMA Declaration of Helsinki and the Department of Health and Human Services Belmont Report. Please provide details of authority granting ethics approval (IRB or equivalent committee(s), provide reference number for approval. If collected and within the bounds of privacy constraints report on age, sex and gender or ethnicity for all study participants.
 - o Statistics: please provide a statement on sample size and randomization.

2/ Figures:

- Please provide individual data points in the graphs, as well as exact p values.
- Please indicate the number of replicate (n) for each figure panel. It appears that in some instances, n=1 (i.e. fig 1A, D, ..). While we usually ask for n=3 (in particular for Western blots), we note that you have performed the experiments in different cell lines and replicated several of them. We also consulted with the referee, who agreed not to ask for further experiments at this stage. We would however ask you to please provide any replicate that you might already have in the Source Data files.
- For composite images comprising several pieces of blots (for instance Fig. 1A), please indicate whether the membranes were stripped, and/or include all loading controls in the figures.

3/ Checklist: the right column should mention where the information indicated in the left column can be found (i.e methods, figure legends, etc).

4/ I introduced minor changes in your Paper Explained, please let me know if you agree or amend as you see fit:

Problem

Platinum chemotherapy plays a key role in the clinical management of patients with non-small cell lung cancer (NSCLC), and is widely used as a monotherapy or in combination across various treatment lines. Although most patients initially respond, resistance invariably develops over time, and identifying the basis of this resistance remains a significant challenge.

Results

We revealed a DNA damage-ATM-MDM2-NICD axis that is activated by platinum treatment. NSCLC patient-derived xenografts with intrinsic platinum resistance regained sensitivity to platinum when NICD generation was hindered. Our preclinical findings were further confirmed by the observation that NSCLC patients with high MDM2 expression respond worst to platinum.

Impact

Our work highlights the potential of targeting the generation of NICD as a strategy for counteracting platinum resistance in this subset of NSCLC patients..

5/ I included minor edits in your synopsis, please let me know if you agree or amend as you see fit:

Most patients with NSCLC undergo platinum-based chemotherapy at some point during their treatment. Some patients never respond, and the majority of those who initially respond will eventually relapse, highlighting the need to understand the mechanisms of platinum resistance.

-Carboplatin-induced DNA damage increases NOTCH1 intracellular domain (NICD) expression levels.

- Upon DNA damage, ATM activates MDM2, which then ubiquitinates NICD and stabilises it. This unveils an ATM-MDM2-NICD axis that decreases the effectiveness of platinum therapy.
- Abolishing NICD generation in vivo makes NSCLC patient-derived xenografts displaying intrinsic resistance to carboplatin more susceptible to platinum-based chemotherapy.
- Patients with NSCLC who express higher levels of MDM2 have shorter progression-free survival after platinum therapy.

Thank you for providing a nice visual abstract. I have cropped a small portion of this image to serve as thumbnail for the table of content on our webpage (attached). Please let us know if you agree or provide an alternative image at the same dimensions (115px x 70px).

6/ As part of the EMBO Publications transparent editorial process initiative (see our Editorial at <http://embomolmed.embopress.org/content/2/9/329>), EMBO Molecular Medicine will publish online a Review Process File (RPF) to accompany accepted manuscripts.

This file will be published in conjunction with your paper and will include the anonymous referee reports, your point-by-point response and all pertinent correspondence relating to the manuscript. Let us know whether you agree with the publication of the RPF and as here, if you want to remove or not any figures from it prior to publication.

I look forward to receiving your revised manuscript.

With kind regards,

Lise

***** Reviewer's comments *****

Referee #1 (Comments on Novelty/Model System for Author):

This work used human lung cancer cell lines, CDX, and PDX models, which are adequate for the study.

Referee #1 (Remarks for Author):

The reviewer appreciates the authors' efforts to revise and improve the manuscript in response to the comments and suggestions. The authors have thoroughly addressed the previous questions and concerns. The current version is in good shape and suitable for publication in EMBO Molecular Medicine.

The authors have addressed the remaining editorial requests.

25th Nov 2025

Dear Dr. Maraver, Dear Antonio,

Thank you for submitting your revised files. I am pleased to inform you that your manuscript is accepted for publication and is now being sent to our publisher to be included in the next available issue of EMBO Molecular Medicine.

With kind regards,

Lise
